# CalmAn an open source tool for scalable calcium imaging data analysis

**Andrea Giovannucci[1]\*, Johannes Friedrich[1,2,3], Pat Gunn[1], Jérémie Kalfon[4†], Brandon L Brown[5], Sue Ann Koay[6], Jiannis Taxidis[7], Farzaneh Najafi[8], Jeffrey L Gauthier[6], Pengcheng Zhou[2,3], Baljit S Khakh[5,9], David W Tank[6], Dmitri B Chklovskii[1], Eftychios A Pnevmatikakis[1]\***

[1]Center for Computational Biology, Flatiron Institute, Simons Foundation, New York, United States; [2]Department of Statistics, Columbia University, New York, United States; [3]Center for Theoretical Neuroscience, Columbia University, New York, United States; [4]ECE Paris, Paris, France; [5]Department of Physiology, University of California, Los Angeles, Los Angeles, United States; [6]Princeton Neuroscience Institute, Princeton University, Princeton, United States; [7]Department of Neurology, University of California, Los Angeles, Los Angeles, United States; [8]Cold Spring Harbor Laboratory, New York, United States; [9]Department of Neurobiology, University of California, Los Angeles, Los Angeles, United States

**\*For correspondence:**
agiovann@email.unc.edu (AG);
epnevmatikakis@flatironinstitute.org (EAP)

†JK contributed to this work during an internship at the Flatiron Institute

**Competing interests:** The authors declare that no competing interests exist.

**Abstract** Advances in fluorescence microscopy enable monitoring larger brain areas in-vivo with finer time resolution. The resulting data rates require reproducible analysis pipelines that are reliable, fully automated, and scalable to datasets generated over the course of months. We present CAIMAN, an open-source library for calcium imaging data analysis. CAIMAN provides automatic and scalable methods to address problems common to pre-processing, including motion correction, neural activity identification, and registration across different sessions of data collection. It does this while requiring minimal user intervention, with good scalability on computers ranging from laptops to high-performance computing clusters. CAIMAN is suitable for two-photon and one-photon imaging, and also enables real-time analysis on streaming data. To benchmark the performance of CAIMAN we collected and combined a corpus of manual annotations from multiple labelers on nine mouse two-photon datasets. We demonstrate that CAIMAN achieves near-human performance in detecting locations of active neurons.
DOI: https://doi.org/10.7554/eLife.38173.001

## Introduction

Understanding the function of neural circuits is contingent on the ability to accurately record and modulate the activity of large neural populations. Optical methods based on the fluorescence activity of genetically encoded calcium binding indicators (*Chen et al., 2013*) have become a standard tool for this task, due to their ability to monitor in vivo targeted neural populations from many different brain areas over extended periods of time (weeks or months). Advances in microscopy techniques facilitate imaging larger brain areas with finer time resolution, producing an ever-increasing amount of data. A typical resonant scanning two-photon microscope produces data at a rate greater than 50 GB/Hr (calculation performed on a $512 \times 512$ Field of View imaged at 30 Hz producing an unsigned 16-bit integer for each measurement), a number that can be significantly higher (up to more than 1TB/Hour) with other custom recording technologies (*Sofroniew et al., 2016*; *Ahrens et al., 2013*; *Flusberg et al., 2008*; *Cai et al., 2016*; *Prevedel et al., 2014*; *Grosenick et al., 2017*; *Bouchard et al., 2015*).

**eLife digest** The human brain contains billions of cells called neurons that rapidly carry information from one part of the brain to another. Progress in medical research and healthcare is hindered by the difficulty in understanding precisely which neurons are active at any given time. New brain imaging techniques and genetic tools allow researchers to track the activity of thousands of neurons in living animals over many months. However, these experiments produce large volumes of data that researchers currently have to analyze manually, which can take a long time and generate irreproducible results.

There is a need to develop new computational tools to analyze such data. The new tools should be able to operate on standard computers rather than just specialist equipment as this would limit the use of the solutions to particularly well-funded research teams. Ideally, the tools should also be able to operate in real-time as several experimental and therapeutic scenarios, like the control of robotic limbs, require this. To address this need, Giovannucci et al. developed a new software package called CaImAn to analyze brain images on a large scale.

Firstly, the team developed algorithms that are suitable to analyze large sets of data on laptops and other standard computing equipment. These algorithms were then adapted to operate online in real-time. To test how well the new software performs against manual analysis by human researchers, Giovannucci et al. asked several trained human annotators to identify active neurons that were round or donut-shaped in several sets of imaging data from mouse brains. Each set of data was independently analyzed by three or four researchers who then discussed any neurons they disagreed on to generate a 'consensus annotation'. Giovannucci et al. then used CaImAn to analyze the same sets of data and compared the results to the consensus annotations. This demonstrated that CaImAn is nearly as good as human researchers at identifying active neurons in brain images.

CaImAn provides a quicker method to analyze large sets of brain imaging data and is currently used by over a hundred laboratories across the world. The software is open source, meaning that it is freely-available and that users are encouraged to customize it and collaborate with other users to develop it further.

DOI: https://doi.org/10.7554/eLife.38173.002

This increasing availability and volume of calcium imaging data calls for automated analysis methods and reproducible pipelines to extract the relevant information from the recorded movies, that is the locations of neurons in the imaged Field of View (FOV) and their activity in terms of raw fluorescence and/or neural activity (spikes). The typical steps arising in the processing pipelines are the following (*Figure 1a*): (i) Motion correction, where the FOV at each data frame (image or volume) is registered against a template to correct for motion artifacts due to the finite scanning rate and existing brain motion, (ii) source extraction where the different active and possibly overlapping sources are extracted and their signals are demixed from each other and from the background neuropil signals (*Figure 1b*), and (iii) activity deconvolution, where the neural activity of each identified source is deconvolved from the dynamics of the calcium indicator.

## Related work

### Source extraction

Some source extraction methods attempt the detection of neurons in static images using supervised or unsupervised learning methods. Examples of unsupervised methods on summary images include graph-cut approaches applied to the correlation image (*Kaifosh et al., 2014*; *Spaen et al., 2017*), and dictionary learning (*Pachitariu et al., 2013*). Supervised learning methods based on boosting (*Valmianski et al., 2010*), or, more recently, deep neural networks have also been applied to the problem of neuron detection (*Apthorpe et al., 2016*; *Klibisz et al., 2017*). While these methods can be efficient in detecting the locations of neurons, they cannot infer the underlying activity nor do they readily offer ways to deal with the spatial overlap of different components.

To extract temporal traces jointly with the spatial footprints of the components one can use methods that directly represent the full spatio-temporal data using matrix factorization approaches for example independent component analysis (ICA) (*Mukamel et al., 2009*), constrained

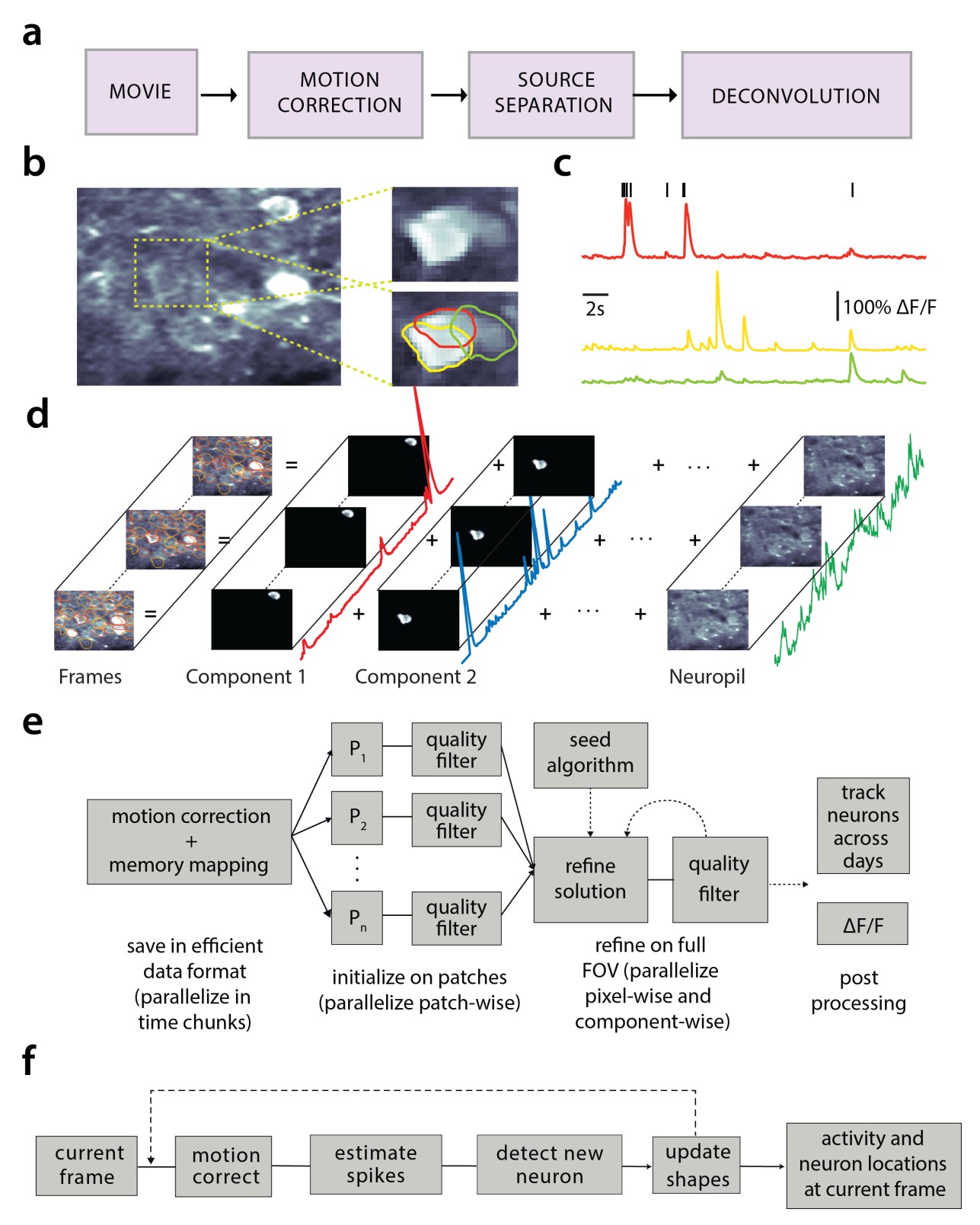

**Figure 1.** Processing pipeline of CAIMAN for calcium imaging data. (**a**) The typical pre-processing steps include (i) correction for motion artifacts, (ii) extraction of the spatial footprints and fluorescence traces of the imaged components, and (iii) deconvolution of the neural activity from the fluorescence traces. (**b**) Time average of 2000 frames from a two-photon microscopy dataset (left) and magnified illustration of three overlapping neurons (right), as detected by the CNMF algorithm. (**c**) Denoised temporal components of the three neurons in (**b**) as extracted by CNMF and

*Figure 1 continued on next page*

*Figure 1 continued*

matched by color (in relative fluorescence change, $\Delta F/F$). (**d**) Intuitive depiction of CNMF. The algorithm represents the movie as the sum of spatially localized rank-one spatio-temporal components capturing neurons and processes, plus additional non-sparse low-rank terms for the background fluorescence and neuropil activity. (**e**) Flow-chart of the CAIMAN BATCH processing pipeline. From left to right: Motion correction and generation of a memory efficient data format. Initial estimate of somatic locations in parallel over FOV patches using CNMF. Refinement and merging of extracted components via seeded CNMF. Removal of low quality components. Final domain dependent processing stages. (**f**) Flow-chart of the CAIMAN ONLINE algorithm. After a brief mini-batch initialization phase, each frame is processed in a streaming fashion as it becomes available. From left to right: Correction for motion artifacts. Estimation of activity from existing neurons, identification and incorporation of new neurons. The spatial footprints of inferred neurons are also updated periodically (dashed lines).

DOI: https://doi.org/10.7554/eLife.38173.003

nonnegative matrix factorization (CNMF) (*Pnevmatikakis et al., 2016*) (and its adaptation to one-photon data (*Zhou et al., 2018*)), clustering based approaches (*Pachitariu et al., 2017*), dictionary learning (*Petersen et al., 2017*), or active contour models (*Reynolds et al., 2017*). Such spatio-temporal methods are unsupervised, and focus on detecting active neurons by considering the spatio-temporal activity of a component as a contiguous set of pixels within the FOV that are correlated in time. While such methods tend to offer a direct decomposition of the data in a set of sources with activity traces in an unsupervised way, in principle they require processing of the full dataset, and thus are quickly rendered intractable. Possible approaches to deal with the data size include distributed processing in High Performance Computing (HPC) clusters (*Freeman et al., 2014*), spatio-temporal decimation (*Friedrich et al., 2017a*), and dimensionality reduction (*Pachitariu et al., 2017*). Recently, *Giovannucci et al., 2017* prototyped an online algorithm (ONACID), by adapting matrix factorization setups (*Pnevmatikakis et al., 2016*; *Mairal et al., 2010*), to operate on calcium imaging streaming data and thus natively deal with large data rates. For a full review see (*Pnevmatikakis, 2018*).

## Deconvolution

For the problem of predicting spikes from fluorescence traces, both supervised and unsupervised methods have been explored. Supervised methods rely on the use of labeled data to train or fit bio-physical or neural network models (*Theis et al., 2016*), although semi-supervised that jointly learn a generative model for fluorescence traces have also been proposed (*Speiser et al., 2017*). Unsupervised methods can be either deterministic, such as sparse non-negative deconvolution (*Vogelstein et al., 2010*; *Pnevmatikakis et al., 2016*) that give a single estimate of the deconvolved neural activity, or probabilistic, that aim to also characterize the uncertainty around these estimates (e.g., (*Pnevmatikakis et al., 2013*; *Deneux et al., 2016*)). A recent community benchmarking effort (*Berens et al., 2017*) characterizes the similarities and differences of various available methods.

## CAIMAN

Here we present CAIMAN, an open source pipeline for the analysis of both two-photon and one-photon calcium imaging data. CAIMAN includes algorithms for both offline analysis (CAIMAN BATCH) where all the data is processed at once at the end of each experiment, and online analysis on streaming data (CAIMAN ONLINE). Moreover, CAIMAN requires very moderate computing infrastructure (e.g., a personal laptop or workstation), thus providing automated, efficient, and reproducible large-scale analysis on commodity hardware.

## Contributions

Our contributions can be roughly grouped in three different directions:

**Methods**: CAIMAN BATCH improves on the scalability of the source extraction problem by employing a MapReduce framework for parallel processing and memory mapping which allows the analysis of datasets larger than would fit in RAM on most computer systems. It also improves on the qualitative performance by introducing automated routines for component evaluation and classification, better handling of neuropil contamination, and better initialization methods. While these benefits are here presented in the context of the widely used CNMF algorithm of *Pnevmatikakis et al. (2016)*, they are in principle applicable to any matrix factorization approach.

CᴀɪᴍAɴ ᴏɴʟɪɴᴇ improves and extends the OɴACID prototype algorithm (*Giovannucci et al., 2017*) by introducing, among other advances, new initialization methods and a convolutional neural network (CNN) based approach for detecting new neurons on streaming data. Our analysis on in vivo two-photon and light-sheet imaging datasets shows that CᴀɪᴍAɴ ᴏɴʟɪɴᴇ approaches human-level performance and enables novel types of closed-loop experiments. Apart from these significant algorithmic improvements CᴀɪᴍAɴ includes several useful analysis tools such as, a MapReduce and memory-mapping compatible implementation of the CNMF-E algorithm for one-photon microendoscopic data (*Zhou et al., 2018*), a novel efficient algorithm for registration of components across multiple days, and routines for segmentation of structural (static) channel information which can be used for component seeding.

**Software**: CᴀɪᴍAɴ is a complete open source software suite implemented primarily in Python, and is already widely used by, and has received contributions from, its community. It contains efficient implementations of the standard analysis pipeline steps (motion correction - source extraction - deconvolution - registration across different sessions), as well as numerous other features. Much of the functionality is also available in a separate MATLAB implementation.

**Data**: We benchmark the performance of CᴀɪᴍAɴ against a previously unreleased corpus of manually annotated data. The corpus consists of 9 mouse in vivo two-photon datasets. Each dataset is manually annotated by 3–4 independent labelers that were instructed to select active neurons in a principled and consistent way. In a subsequent stage, the annotations were combined to create a 'consensus' annotation, that is used to benchmark CᴀɪᴍAɴ, to train supervised learning based classifiers, and to quantify the limits of human performance. The manual annotations are released to the community, providing a valuable tool for benchmarking and training purposes.

## Paper organization

The paper is organized as follows: We first give a brief presentation of the analysis methods and features provided by CᴀɪᴍAɴ. In the *Results* section we benchmark CᴀɪᴍAɴ ʙᴀᴛᴄʜ and CᴀɪᴍAɴ ᴏɴʟɪɴᴇ against a corpus of manually annotated data. We apply CᴀɪᴍAɴ ᴏɴʟɪɴᴇ to a zebrafish whole brain lightsheet imaging recording, and demonstrate how such large datasets can be processed efficiently in real time. We also present applications of CᴀɪᴍAɴ ʙᴀᴛᴄʜ to one-photon data, as well as examples of component registration across multiple days. We conclude by discussing the utility of our tools, the relationship between CᴀɪᴍAɴ ʙᴀᴛᴄʜ and CᴀɪᴍAɴ ᴏɴʟɪɴᴇ and outline future directions. Detailed descriptions of the introduced methods are presented in Materials and methods.

## Methods

Before presenting the new analysis features introduced with this work, we overview the analysis pipeline that CᴀɪᴍAɴ uses and builds upon.

### Overview of analysis pipeline

The standard analysis pipeline for calcium imaging data used in CᴀɪᴍAɴ is depicted in *Figure 1a*. The data is first processed to remove motion artifacts. Subsequently the active components (neurons and background) are extracted as individual pairs of a spatial footprint that describes the shape of each component projected to the imaged FOV, and a temporal trace that captures its fluorescence activity (*Figure 1b–d*). Finally, the neural activity of each fluorescence trace is deconvolved from the dynamics of the calcium indicator. These operations can be challenging because of limited axial resolution of 2-photon microscopy (or the much larger integration volume in one-photon imaging). This results in spatially overlapping fluorescence from different sources and neuropil activity. Before presenting the new features of CᴀɪᴍAɴ in more detail, we briefly review how it incorporates existing tools in the pipeline.

### Motion correction

CᴀɪᴍAɴ uses the NᴏRMCᴏʀʀᴇ algorithm (*Pnevmatikakis and Giovannucci, 2017*) that corrects non-rigid motion artifacts by estimating motion vectors with subpixel resolution over a set of overlapping patches within the FOV. These estimates are used to infer a smooth motion field within the FOV for each frame. For two-photon imaging data this approach is directly applicable, whereas for one-photon micro-endoscopic data the motion is estimated on high pass spatially filtered data, a necessary

operation to remove the smooth background signal and create enhanced spatial landmarks. The inferred motion fields are then applied to the original data frames.

### Source extraction

Source extraction is performed using the constrained non-negative matrix factorization (CNMF) framework of *Pnevmatikakis et al. (2016)* which can extract components with overlapping spatial footprints (*Figure 1b*). After motion correction the spatio-temporal activity of each source can be expressed as a rank one matrix given by the outer product of two components: a component in space that describes the spatial footprint (location and shape) of each source, and a component in time that describes the activity trace of the source (*Figure 1c*). The data can be described by the sum of all the resulting rank one matrices together with an appropriate term for the background and neuropil signal and a noise term (*Figure 1d*). For two-photon data the neuropil signal can be modeled as a low rank matrix (*Pnevmatikakis et al., 2016*). For microendoscopic data the larger integration volume leads to more complex background contamination (*Zhou et al., 2018*). Therefore, a more descriptive model is required (see Materials and methods (Mathemathical model of the CNMF framework) for a mathematical description). CAIMAN BATCH embeds these approaches into a general algorithmic framework that enables scalable automated processing with improved results versus the original CNMF and other popular algorithms, in terms of quality and processing speed.

### Deconvolution

Neural activity deconvolution is performed using sparse non-negative deconvolution (*Vogelstein et al., 2010*; *Pnevmatikakis et al., 2016*) and implemented using the near-online OASIS algorithm (*Friedrich et al., 2017b*). The algorithm is competitive to the state of the art according to recent benchmarking studies (*Berens et al., 2017*). Prior to deconvolution, the traces are detrended to remove non-stationary effects, for example photo-bleaching.

### Online processing

The three processing steps described above can be implemented in an online fashion using the ONACID algorithm (*Giovannucci et al., 2017*). The method extends the online dictionary learning framework presented in *Mairal et al. (2010)* for source extraction, by introducing spatial constraints, adding the capability of finding new components as they appear and also incorporating the steps of motion correction and deconvolution (*Figure 1e*). CAIMAN extends and improves the ONACID prototype algorithm by introducing a number of algorithmic features and a CNN based component detection approach, leading to a major performance improvement.

We now present the new methods introduced by CAIMAN. More details are given in Materials and methods and pseudocode descriptions of the main routines are given in the *Appendix*.

## Batch processing of large scale datasets on standalone machines

The batch processing pipeline mentioned above represents a computational bottleneck. For instance, a naive first step might be to load in-memory the full dataset; this approach is non-scalable as datasets typically exceed available RAM (and extra memory is required by any analysis pipeline). To limit memory usage, as well as computation time, CAIMAN BATCH relies on a MapReduce approach (*Dean and Ghemawat, 2008*). Unlike previous work (*Freeman et al., 2014*), CAIMAN BATCH assumes minimal computational infrastructure (down to a standard laptop computer), is not tied to a particular parallel computation framework, and is compatible with HPC scheduling systems like SLURM (*Yoo et al., 2003*).

Naive implementations of motion correction algorithms need to either load in memory the full dataset or are constrained to process one frame at a time, therefore preventing parallelization. Motion correction is parallelized in CAIMAN BATCH without significant memory overhead by processing temporal chunks of movie data on different CPUs. First, each chunk is registered with its own template and a new template is formed by the registered data of each chunk. CAIMAN BATCH then broadcasts to each CPU a meta-template, obtained as the median between all templates, which is used to align all the frames in each chunk. Each process writes in parallel to the target file containing motion-corrected data, which is stored as a memory mapped array. This allows arithmetic operations to be performed against data stored on the hard drive with minimal memory use, and data slices to

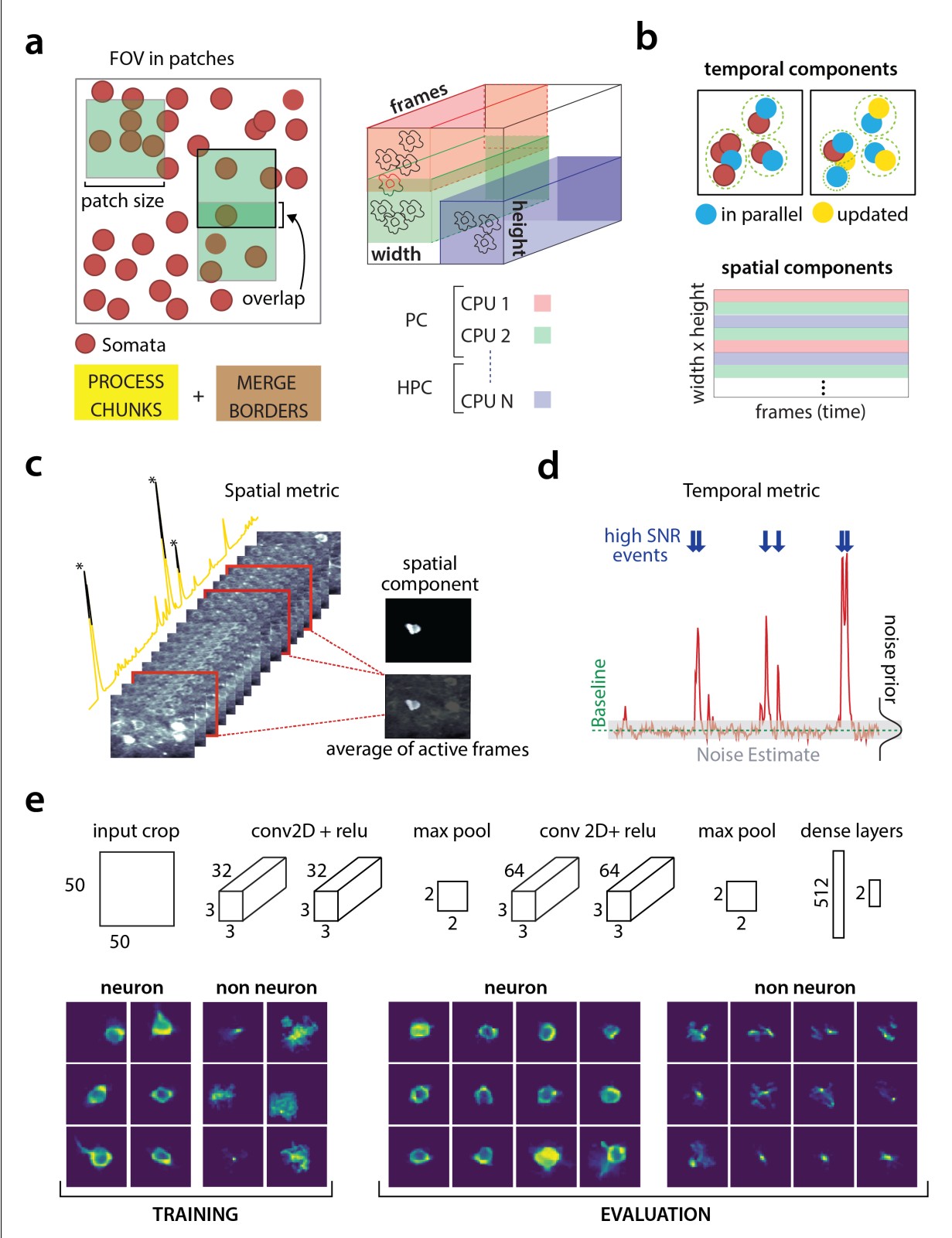

**Figure 2.** Parallelized processing and component quality assessment for CaImAn batch. (**a**) Illustration of the parallelization approach used by CaImAn batch for source extraction. The data movie is partitioned into overlapping sub-tensors, each of which is processed in an embarrassingly parallel fashion using CNMF, either on local cores or across several machines in a HPC. The results are then combined. (**b**) Refinement after combining the results can also be parallelized both in space and in time. Temporal traces of spatially non-overlapping components can be updated in parallel (top) and the

*Figure 2 continued on next page*

*Figure 2 continued*

contribution of the spatial footprints for each pixel can be computed in parallel (bottom). Parallelization in combination with memory mapping enable large scale processing with moderate computing infrastructure. (**c**) Quality assessment in space: The spatial footprint of each real component is correlated with the data averaged over time, after removal of all other activity. (**d**) Quality assessment in time: A high SNR is typically maintained over the course of a calcium transient. (**e**) CNN based assessment. *Top*: A 4-layer CNN based classifier is used to classify the spatial footprint of each component into neurons or not, see Materials and methods (*Classification through CNNs*) for a description. *Bottom*: Positive and negative examples for the CNN classifier, during training (left) and evaluation (right) phase. The CNN classifier can accurately classify shapes and generalizes across datasets from different brain areas.

DOI: https://doi.org/10.7554/eLife.38173.004

be indexed and accessed without loading the full file in memory. More details are given in Materials and methods (Memory mapping).

Similarly, the source extraction problem, especially in the case of detecting cell bodies, is inherently local with a neuron typically appearing in a neighborhood within a small radius from its center of mass (*Figure 2a*). Exploiting this locality, CAIMAN BATCH splits the FOV into a set of spatially overlapping patches which enables the parallelization of the CNMF (or any other) algorithm to extract the corresponding set of local spatial and temporal components. The user specifies the size of the patch, the amount of overlap between neighboring patches and the initialization parameters for each patch (number of components and rank background for CNMF, average size of each neuron, stopping criteria for CNMF-E). Subsequently the patches are processed in parallel by the CNMF/CNMF-E algorithm to extract the components and neuropil signals from each patch.

Apart from harnessing memory and computational benefits due to parallelization, processing in patches intrinsically equalizes dynamic range and enables CAIMAN BATCH to detect neurons across the whole FOV, a feature absent in the original CNMF, where areas with high absolute fluorescence variation tend to be favored. This results in better source extraction performance. After all the patches have been processed, the results are embedded within the FOV (*Figure 2a*), and the overlapping regions between neighboring patches are processed so that components corresponding to the same neuron are merged. The process is summarized in algorithmic format in Algorithm 1 and more details are given in Materials and methods (Combining results from different patches).

## Initialization methods

Due to the non-convex nature of the objective function for matrix factorization, the choice of the initialization method can severely impact the final results. CAIMAN BATCH provides an extension of the GREEDYROI method used in *Pnevmatikakis et al. (2016)*, that detects neurons based on localized spatiotemporal activity. CAIMAN BATCH can also be seeded with binary masks that are obtained from different sources, for example through manual annotation or segmentation of structural channel (SEEDEDINITIALIZATION, Algorithm 3). More details are given in Materials and methods (Initialization strategies).

## Automated component evaluation and classification

A common limitation of matrix factorization algorithms is that the number of components that the algorithm seeks during its initialization must be pre-determined by the user. For example, *Pnevmatikakis et al. (2016)* suggest detecting a large number of components which are then ordered according to their size and activity pattern, with the user deciding on a cut-off threshold. When processing large datasets in patches the target number of components is passed on to every patch implicitly assuming a uniform density of (active) neurons within the entire FOV. This assumption does not hold in the general case and can produce many spurious components. CAIMAN introduces tests, based on unsupervised and supervised learning, to assess the quality of the detected components and eliminate possible false positives. These tests are based on the observation that active components are bound to have a distinct localized spatio-temporal signature within the FOV. In CAIMAN BATCH, these tests are initially applied after the processing of each patch is completed, and additionally as a post-processing step after the results from the patches have been merged and refined, whereas in CAIMAN ONLINE they are used to screen new candidate components. We briefly present these tests below and refer to Materials and methods (Details of quality assessment tests) for more details:

**Spatial footprint consistency**: To test whether a detected component is spurious, we correlate the spatial footprint of this component with the average frame of the data, taken over the intervals when the component, with no other overlapping component, was active (*Figure 2c*). The component is rejected if the correlation coefficient is below a certain threshold $\theta_{sp}$ (e.g., $\theta_{sp} < 0.5$).

**Trace SNR**: For each component we computed the peak SNR of its temporal trace averaged over the duration of a typical transient (*Figure 2d*). The component is rejected if the computed SNR is below a certain threshold $\theta_{SNR}$ (e.g., $\theta_{SNR} = 2$).

**CNN based classification**: We also trained a 4-layer convolutional neural network (CNN) to classify spatial footprints into true or false components (*Figure 2e*), where a true component here corresponds to a spatial footprint that resembles the soma of a neuron. The classifier, which we call batch classifier, was trained on a small corpus of manually annotated datasets (full description given in section *Benchmarking against consensus annotation*) and exhibited similar high classification performance on test samples from different datasets.

While CaImAn uses the CNMF algorithm, the tests described above can be applied to results obtained from any source extraction algorithm, highlighting the modularity of our tools.

## Online analysis with CaImAn online

CaImAn supports online analysis on streaming data building on the core of the prototype algorithm of *Giovannucci et al., 2017*, and extending it in terms of qualitative performance and computational efficiency:

**Initialization:** Apart from initializing CaImAn online with CaImAn batch on a small time interval, CaImAn online can also be initialized in a bare form over an even smaller time interval, where only the background components are estimated and all the components are determined during the online analysis. This process, named BareInitialization, can be achieved by running the CNMF algorithm (*Pnevmatikakis et al., 2016*) over the small interval to estimate the background components and possibly a small number of components. The SeededInitialization of Algorithm 3 can also be used.

**Deconvolution:** Instead of a separate step after demixing as in *Giovannucci et al., 2017*, deconvolution here can be performed simultaneously with the demixing online, leading to more stable traces especially in cases of low-SNR, as also observed in *Pnevmatikakis et al. (2016)*. Online deconvolution can also be performed for models that assume second order calcium dynamics, bringing the full power of *Friedrich et al., 2017b* to processing of streaming data.

**Epochs:** CaImAn online supports multiple passes over the data, a process that can detect early activity of neurons that were not picked up during the initial pass, as well as smooth the activity of components that were detected at late stages during the first epoch.

**New component detection using a CNN:** To search for new components in a streaming setup, OnACID keeps a buffer of the residual frames, computed by subtracting the activity of already found components and background signals. Candidate components are determined by looking for points of maximum energy in this residual signal, after some smoothing and dynamic range equalization. For each such point identified, a candidate shape and trace are constructed using a rank-1 NMF in a local neighborhood around this point. In its original formulation (*Giovannucci et al., 2017*), the shape of the component was evaluated using the space correlation test described above. Here, we use a CNN classifier approach that tests candidate components by examining their spatial footprint as obtained by the average of the residual buffer across time. This online classifier (different from the batch classifier for quality assessment described above), is trained to be strict, minimizing the number of false positive components that enter the online processing pipeline. It can test multiple components in parallel, and it achieves better performance with no hyper-parameter tuning compared to the previous approach. More details on the architecture and training procedure are given in Materials and methods (Classification through CNNs). The identification of candidate components is further improved by performing spatial high pass filtering on the average residual buffer to enhance its contrast. The new process for detecting neurons is described in Algorithm 4 and 5. See *Videos 1* and *2* on a detailed graphic description of the new component detection step.

Distributed update of spatial footprints: A time limiting step in OnACID (*Giovannucci et al., 2017*) is the periodic update of all spatial footprints at given frames. This constraint is lifted with AImAn online that distributes the update of spatial footprints among all frames ensuring a similar

processing speed for each frame. See Materials and methods (Distributed shape update) for more details.

## Component registration across multiple sessions

 CaImAn provides a method to register components from the same FOV across different sessions. The method uses an intersection over union metric to calculate the distance between different cells in different sessions and solves a linear assignment problem to perform the registration in a fully automated way (RegisterPair, Algorithm 7). To register the components between more than two sessions (RegisterMulti, Algorithm 8), we order the sessions chronologically and register the components of the current session against the union of components of all the past sessions aligned to the current FOV. This allows for the tracking of components across multiple sessions without the need of pairwise registration between each pair of sessions. More details as well as discussion of other methods (*Sheintuch et al., 2017*) are given in Materials and methods (Component registration).

## Benchmarking against manual annotations

To quantitatively evaluate CaImAn we benchmarked its results against manual annotations.

### Creating consensus labels through manual annotation

We collected manual annotations from multiple independent labelers who were instructed to find round or donut shaped (since proteins expressing the calcium indicator are confined outside the cell nuclei, neurons will appear as ring shapes, with a dark disk in the center) *active* neurons on nine two-photon in vivo mouse brain datasets. To distinguish between active and inactive neurons, the annotators were given the max-correlation image for each dataset (the value of the correlation image for each pixel represent the average correlation (across time) between the pixel and its neighbors (*Smith and Häusser, 2010*). This summarization can enhance active neurons and suppress neuropil for two photon datasets (*Figure 3—figure supplement 1a*). See Materials and methods (Collection of manual annotations) for more information). In addition, the annotators were given a temporally decimated background subtracted movie of each dataset. The datasets were collected at various labs and from various brain areas (hippocampus, visual cortex, parietal cortex) using several GCaMP variants. A summary of the features of all the annotated datasets is given in Table 2.

To address human variability in manual annotation each dataset was labeled by 3 or 4 independent labelers, and the final consensus annotation dataset was created by having the different labelers reaching a *consensus* over their disagreements (*Figure 3a*). The consensus annotation was taken as 'ground truth' for the purpose of benchmarking CaImAn and each individual labeler (*Figure 3b*). More details are given in Materials and methods (Collection of manual annotations). We believe that the current database, which is publicly available at https://users.flatironinstitute.org/~neuro/caiman_paper, presents an improvement over the existing Neurofinder database (http://neurofinder.code-neuro.org/) in several aspects:

**Consistency**: The datasets are annotated using exactly the same procedure (see Materials and methods), and in all datasets the goal is to detect only active cells. In contrast, the annotation of the various Neurofinder datasets is performed either manually or automatically by segmenting an image of a static (structural) indicator. Even though structural indicators could be used for ground truth extraction, the segmentation of such images is not a straightforward problem in the case of dense expression, and the stochastic expression of indicators can lead to mismatches between functional and structural indicators.

**Uncertainty quantification**: By employing more than one human labeler we discovered a surprising level of disagreement between different annotators (see *Table 1*, *Figure 3b* for details). This result indicates that individual annotations can be unreliable for benchmarking purposes and that unreproducible scientific results might ensue. The combination of the various annotations leads to more reliable set of labels and also quantifies the limits of human performance.

### Comparing CaImAn against manual annotations

To compare CaImAn against the consensus annotation, the manual annotations were used as binary masks to construct the consensus spatial and temporal components, using the SeededInitialization procedure (Algorithm 3) of CaImAn batch. This step is necessary to adapt the manual annotations to

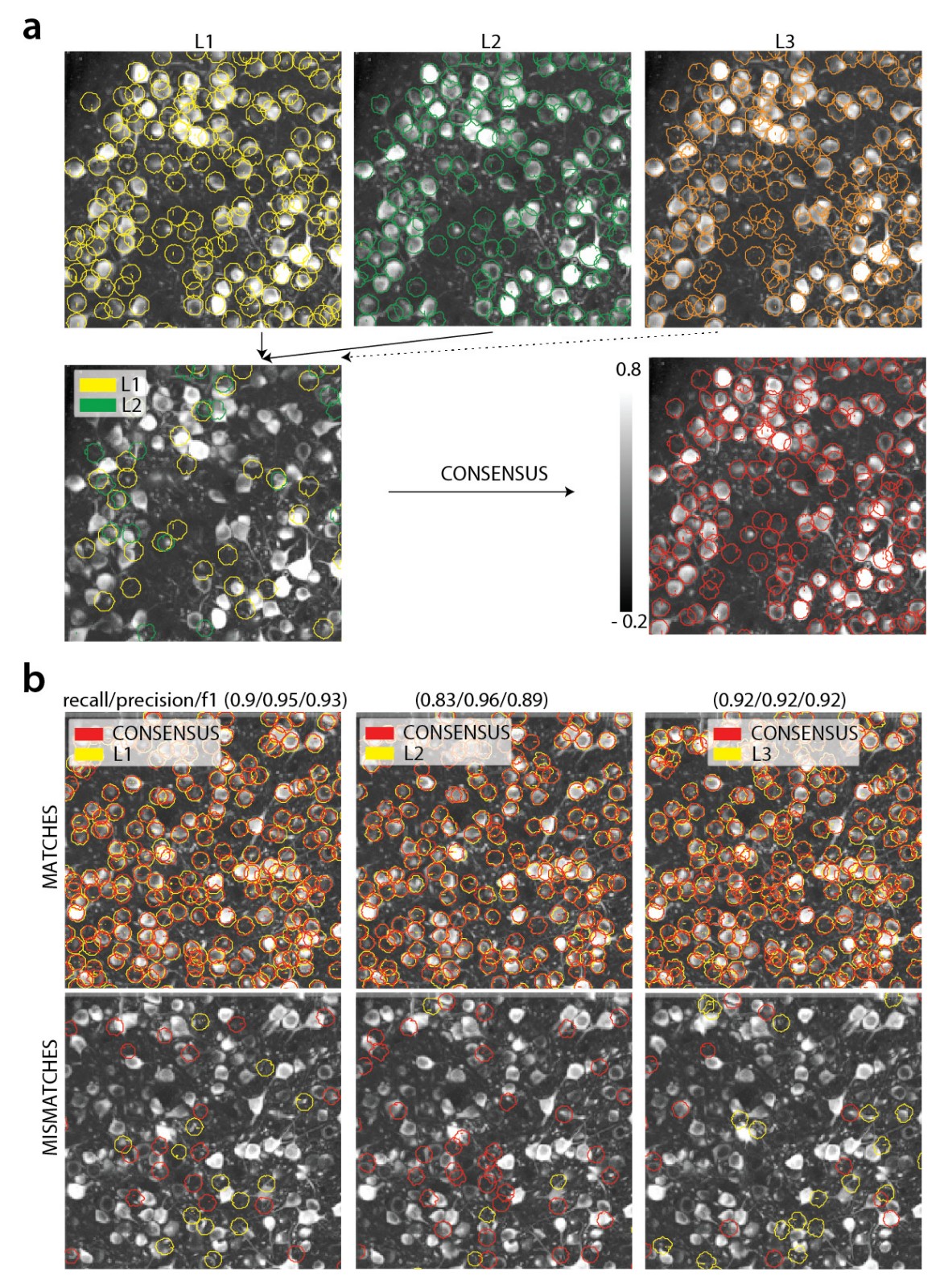

**Figure 3.** Consensus annotation generation. (**a**) *Top:* Individual manual annotations on the dataset K53 (only part of the FOV is shown) for labelers L1 (left), L2 (middle), L3(right). Contour plots are plotted against the max-correlation image of the dataset. *Bottom:* Disagreements between L1 and L2 (left), and consensus labels (right). In this example, consensus considerably reduced the number of initially selected neurons. (**b**) Matches (top) and mismatches (bottom) between each individual labeler and consensus annotation. Red contours on the mismatches panels denote false negative

*Figure 3 continued on next page*

*Figure 3 continued*

contours, that is components in the consensus not selected by the corresponding labeler, whereas yellow contours indicate false positive contours. Performance of each labeler is given in terms of precision/recall and $F_1$ score and indicates an unexpected level of variability between individual labelers.

DOI: https://doi.org/10.7554/eLife.38173.005
The following figure supplement is available for figure 3:

**Figure supplement 1.** Construction of components obtained from consensus annotation.
DOI: https://doi.org/10.7554/eLife.38173.006

the shapes of the actual spatial footprints of each neuron in the FOV (*Figure 3—figure supplement 1*), as manual annotations primarily produced elliptical shapes. The set of spatial footprints obtained from CaImAn is registered against the set of consensus spatial footprints (derived as described above) using our component registration algorithm REGISTERPAIR (Algorithm 7). Performance is then quantified using a precision/recall framework similar to other studies (*Apthorpe et al., 2016*; *Giovannucci et al., 2017*).

## Software

CaImAn is developed by and for the community. Python open source code for the above-described methods is available at https://github.com/flatironinstitute/CaImAn (*Giovannucci et al., 2018*; copy archived at https://github.com/elifesciences-publications/CaImAn). The repository contains documentation, several demos, and Jupyter notebook tutorials, as well as visualization tools, and a message/discussion board. The code, which is compatible with Python 3, uses several open-source libraries, such as OpenCV (*Bradski, 2000*), scikit-learn (*Pedregosa et al., 2011*), and scikit-image (*van der Walt et al., 2014*). Most routines are also available in MATLAB at https://github.com/flatironinstitute/CaImAn-MATLAB (*Pnevmatikakis et al., 2018*; copy archived at https://github.com/elifesciences-publications/CaImAn-MATLAB). We provide tips for efficient data analysis at https://github.com/flatironinstitute/CaImAn/wiki/CaImAn-Tips. All the annotated datasets together with the

**Table 1.** Results of each labeler, CaImAn batch and CaImAn online algorithms against consensus annotation.

Results are given in the form $\frac{F_1 \text{score} \quad \text{\# of active neurons}}{(\text{precision,} \quad \text{recall})}$, and empty entries correspond to datasets not manually annotated by the specific labeler. The number of frames for each dataset, as well as the number of neurons that each labeler and algorithm found are also given. In *italics* the datasets used to train the CNN classifiers.

| Name<br># of frames | L1 | L2 | L3 | L4 | CaImAn batch | CaImAn online |
|---|---|---|---|---|---|---|
| *N.01.01*<br>1825 | 0.80 241<br>(0.95, 0.69) | 0.89 287<br>(0.96, 0.83) | 0.78 386<br>(0.73, 0.84) | 0.75 289<br>(0.80, 0.70) | 0.76 317<br>(0.76, 0.77) | 0.75 298<br>(0.81, 0.70) |
| *N.03.00.t*<br>2250 | X | 0.90 188<br>(0.88, 0.92) | 0.85 215<br>(0.78, 0.93) | 0.78 206<br>(0.73, 0.83) | 0.78 154<br>(0.76, 0.80) | 0.74 150<br>(0.79, 0.70) |
| *N.00.00*<br>2936 | X | 0.92 425<br>(0.93, 0.91) | 0.83 402<br>(0.86, 0.80) | 0.87 358<br>(0.96, 0.80) | 0.72 366<br>(0.79, 0.67) | 0.69 259<br>(0.87, 0.58) |
| YST<br>3000 | 0.78 431<br>(0.76, 0.81) | 0.90 465<br>(0.85, 0.97) | 0.82 505<br>(0.75, 0.92) | 0.79 285<br>(0.96, 0.67) | 0.77 332<br>(0.85, 0.70) | 0.77 330<br>(0.84, 0.70) |
| N.04.00.t<br>3000 | X | 0.69 471<br>(0.54, 0.97) | 0.75 411<br>(0.61, 0.97) | 0.87 326<br>(0.78, 0.98) | 0.69 218<br>(0.69, 0.70) | 0.7 260<br>(0.68, 0.72) |
| N.02.00<br>8000 | 0.89 430<br>(0.86, 0.93) | 0.87 382<br>(0.88, 0.85) | 0.84 332<br>(0.92, 0.77) | 0.82 278<br>(1.00, 0.70) | 0.78 351<br>(0.78, 0.78) | 0.78 334<br>(0.85, 0.73) |
| J123<br>41000 | X | 0.83 241<br>(0.73, 0.96) | 0.90 181<br>(0.91, 0.90) | 0.91 177<br>(0.92, 0.89) | 0.73 157<br>(0.88, 0.63) | 0.82 172<br>(0.85, 0.80) |
| J115<br>90000 | 0.85 708<br>(0.96, 0.76) | 0.93 869<br>(0.94, 0.91) | 0.94 880<br>(0.95, 0.93) | 0.83 635<br>(1.00, 0.71) | 0.78 738<br>(0.87, 0.71) | 0.79 1091<br>(0.71, 0.89) |
| K53<br>116043 | 0.89 795<br>(0.96, 0.83) | 0.92 928<br>(0.92, 0.92) | 0.93 875<br>(0.95, 0.91) | 0.83 664<br>(1.00, 0.72) | 0.76 809<br>(0.80, 0.72) | 0.81 1025<br>(0.77, 0.87) |
| mean ± std | 0.84±0.05<br>(0.9±0.08, 0.8±0.08) | 0.87±0.07<br>(0.85±0.13, 0.92±0.05) | 0.85±0.06<br>(0.83±0.11, 0.88±0.06) | 0.83±0.09<br>(0.91±0.1, 0.78±0.1) | 0.754±0.03<br>(0.8±0.06, 0.72±0.05) | 0.762±0.05<br>(0.82±0.06, 0.73±0.1) |

DOI: https://doi.org/10.7554/eLife.38173.007

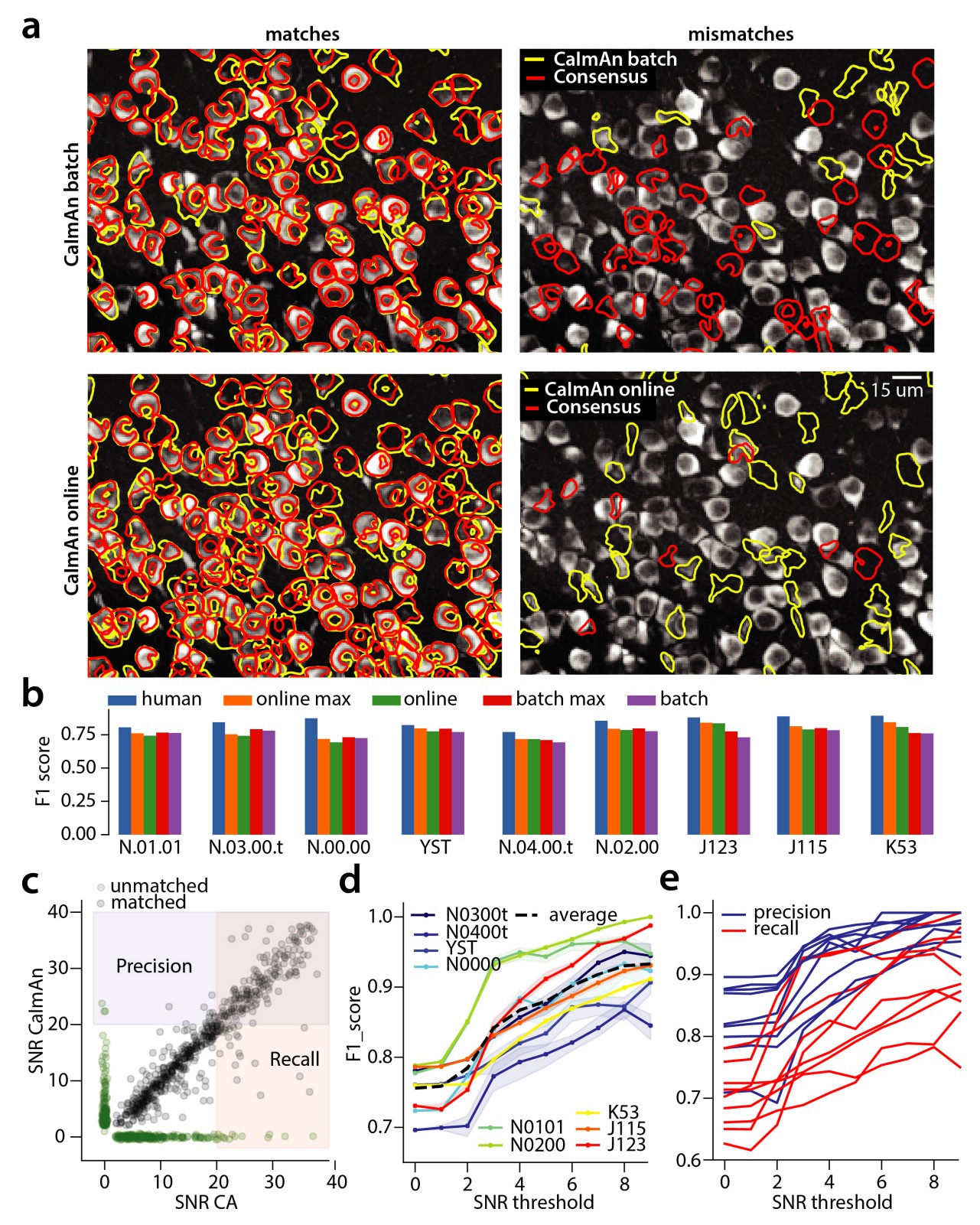

**Figure 4.** Evaluation of CᴀʟᴍAɴ performance against manually annotated data. (a) Comparison of CᴀʟᴍAɴ ʙᴀᴛᴄʜ (top) and CᴀʟᴍAɴ ᴏɴʟɪɴᴇ (bottom) when benchmarked against consensus annotation for dataset K53. For a portion of the FOV, correlation image overlaid with matches (left panels, red: consensus, yellow: CᴀʟᴍAɴ) and mismatches (right panels, red: false negatives, yellow: false positives). (b) Performance of CᴀʟᴍAɴ ʙᴀᴛᴄʜ, and CᴀʟᴍAɴ ᴏɴʟɪɴᴇ vs average human performance (blue). For each algorithm the results with both the same parameters for each dataset and with the optimized per

*Figure 4 continued on next page*

*Figure 4 continued*

dataset parameters are shown. CAIMAN BATCH and CAIMAN ONLINE reach near-human accuracy for neuron detection. Complete results with precision and recall for each dataset are given in *Table 1*. (c–e) Performance of CAIMAN BATCH increases with peak SNR. (c) Example of scatter plot between SNRs of matched traces between CAIMAN BATCH and consensus annotation for dataset K53. False negative/positive pairs are plotted in green along the x- and y-axes respectively, perturbed as a point cloud to illustrate the density. Most false positive/negative predictions occur at low SNR values. Shaded areas represent thresholds above which components are considered for matching (blue for CAIMAN BATCH and red for consensus selected components) (d) $F_1$ score and upper/lower bounds of CAIMAN BATCH for all datasets as a function of various peak SNR thresholds. Performance of CAIMAN BATCH increases significantly for neurons with high peak SNR traces (see text for definition of metrics and the bounds). (e) Precision and recall of CAIMAN BATCH as a function of peak SNR for all datasets. The same trend is observed for both precision and recall.

DOI: https://doi.org/10.7554/eLife.38173.008

The following figure supplements are available for figure 4:

**Figure supplement 1.** Performance of CAIMAN ONLINE over different choices of parameters.
DOI: https://doi.org/10.7554/eLife.38173.009

**Figure supplement 2.** CAIMAN BATCH outperforms the Suite2p algorithm in all datasets when benchmarked against the consensus annotation.
DOI: https://doi.org/10.7554/eLife.38173.010

individual and consensus annotation are available at https://users.flatironinstitute.org/~neuro/caiman_paper. All the material is also available from the Zenodo repository at https://zenodo.org/record/1659149/export/hx#.XC_Rms9Ki9t

# Results

## Manual annotations show a high degree of variability

We compared the performance of each human annotator against a consensus annotation. The performance was quantified with a precision/recall framework and the results of the performance of each individual labeler against the consensus annotation for each dataset is given in *Table 1*. The range of human performance in terms of $F_1$ score was 0.69–0.94. All annotators performed similarly on average (0.84 ± 0.05, 0.87 ± 0.07, 0.85 ± 0.06, 0.83 ± 0.08). We also ensured that the performance of labelers was stable across time (i.e. their learning curve plateaued, data not shown). As shown in *Table 1* (see also *Figure 4b*) the $F_1$ score was never 1, and in most cases it was less or equal to 0.9, demonstrating significant variability between annotators. *Figure 3* (bottom) shows an example of matches and mismatches between individual labelers and consensus annotation for dataset K53, where the level of agreement was relatively high. The high degree of variability between human responses indicates the challenging nature of the source extraction problem and raises reproducibility concerns in studies relying heavily on manual ROI selection.

This process may have generated slightly biased results in favor of each individual annotator as the consensus annotation is always a subset of the union of the individual annotations. We also used an alternative cross-validation approach, where the labels of each annotator were compared with

**Table 2.** Properties of manually annotated datasets.

For each dataset the duration, imaging rate and calcium indicator are given, as well as the number of active neurons selected after consensus between the manual annotators.

| Name | Area brain | Lab | Rate (Hz) | Size (T×X×Y) | Indicator | #Labelers | #Neurons CA |
|------|-----------|-----|-----------|--------------|-----------|-----------|-------------|
| NF.01.01 | Visual Cortex | Hausser | 7 | 1825 × 512 × 512 | GCaMP6s | 4 | 333 |
| NF.03.00.t | Hippocampus | Losonczy | 7 | 2250 × 498 × 467 | GCaMP6f | 3 | 178 |
| NF.00.00 | Cortex | Svoboda | 7 | 2936 × 512 × 512 | GCaMP6s | 3 | 425 |
| YST | Visual Cortex | Yuste | 10 | 3000 × 200 × 256 | GCaMP3 | 4 | 405 |
| NF.04.00.t | Cortex | Harvey | 7 | 3000 × 512 × 512 | GCaMP6s | 3 | 257 |
| NF.02.00 | Cortex | Svoboda | 30 | 8000 × 512 × 512 | GCaMP6s | 4 | 394 |
| J123 | Hippocampus | Tank | 30 | 41000 × 458 × 477 | GCaMP5 | 3 | 183 |
| J115 | Hippocampus | Tank | 30 | 90000 × 463 × 472 | GCaMP5 | 4 | 891 |
| K53 | Parietal Cortex | Tank | 30 | 116043 × 512 × 512 | GCaMP6f | 4 | 920 |

DOI: https://doi.org/10.7554/eLife.38173.011

the combined results of the remaining annotators. The combination was constructed using a majority vote when a dataset was labeled from 4 annotators, or an intersection of selections when a dataset was labeled by 3. The results (see Table 3 in Materials and methods) indicate an even higher level of disagreement between the annotators with lower average $F_1$ score $0.82 \pm 0.06$ (mean $\pm$ STD) and range of values $0.68 - 0.90$. More details are given in Materials and methods (Cross-Validation analysis of manual annotations).

## CaImAn batch and CaImAn online detect neurons with near-human accuracy

We first benchmarked CaImAn batch and CaImAn online against consensus annotation for the task of identifying neurons locations and their spatial footprints, using the same precision recall framework (*Table 1*). *Figure 4a* shows an example dataset (K53) along with neuron-wise matches and mismatches between CaImAn batch vs consensus annotation (top) and CaImAn online vs consensus annotation (bottom).

The results indicate a similar performance between CaImAn batch and CaImAn online; CaImAn batch has $F_1$ scores in the range 0.69–0.78 and average performance $0.75 \pm 0.03$ (mean $\pm$ STD). On the other hand CaImAn online had $F_1$ scores in the range 0.70–0.82 and average performance $0.76 \pm 0.05$. While the two algorithms performed similarly on average, CaImAn online tends to perform better for longer datasets (e.g., datasets J115, J123, K53 that all have more than 40000 frames; see also *Table 2* for characteristics of the various datasets). CaImAn batch operates on the entire dataset at once, representing each spatial footprint with a constant in time vector. In contrast, CaImAn online operates at a local level looking at a short window over time to detect new components, while adaptively changing their spatial footprint based on new data. This enables CaImAn online to adapt to slow non-stationarities that can appear in long experiments.

CaImAn approaches but is in most cases below the accuracy levels of human annotators (*Figure 4b*). We attribute this to two primary factors: First, CNMF detects active components regardless of their shape, and can detect non-somatic structures with significant transients. While non-somatic components can be filtered out to some extent using the CNN classifier, their existence degrades performance compared to the manual annotations that consist only of neurons. Second, to demonstrate the generality and ease of use of our tools, the results presented here are obtained by running CaImAn batch and CaImAn online with *exactly* the same parameters for each dataset (see Materials and methods (Implementation details)): fine-tuning to each individual dataset can significantly increase performance (*Figure 4b*).

To test the later point we measured the performance of CaImAn online on the nine datasets, as a function of 3 parameters: (i) the trace SNR threshold for testing the traces of candidate components, (ii) the CNN threshold for testing the shapes of candidate components, and (iii) the number of candidate components to be tested at each frame (more details can be found in Materials and methods (Implementation details for CaImAn online)). By choosing a parameter combination that maximizes the value for each dataset, the performance generally increases across the datasets with $F_1$ scores in the range 0.72–0.85 and average performance $0.78 \pm 0.05$ (see *Figure 4* (orange) and *Figure 4—figure supplement 1* (magenta)). This analysis also shows that in general a strategy of testing a large number of components per timestep but with stricter criteria, achieves better results than testing fewer components with looser criteria (at the expense of increased computational cost). The results also indicate different strategies for parameter choice depending on the length of a dataset: Lower threshold values and/or larger number of candidate components (*Figure 4—figure supplement 1* (red)), lead to better values for shorter datasets, but can decrease precision and overall performance for longer datasets. The opposite also holds for higher threshold values and/or smaller number of candidate components (*Figure 4—figure supplement 1* (blue)), where CaImAn online for shorter datasets can suffer from lower recall values, whereas in longer datasets CaImAn online can add neurons over a longer period of time while maintaining high precision values and thus achieve better performance. A similar grid search was also performed for the CaImAn batch algorithm where four parameters of the component evaluation step (space correlation, trace SNR, min/max CNN thresholds) were optimized individually to filter out false positives. This procedure led to $F_1$ scores in in the range 0.71–0.81 and average performance $0.774 \pm 0.034$ (*Figure 4* (red)).

We also compared the performance of CaImAn against Suite2p (*Pachitariu et al., 2017*), another popular calcium imaging data analysis package. By using a small grid search around some default

parameters of Suite2p we extracted the set of parameters that worked better in the eight datasets where the algorithm converged (in the dataset J123 Suite2p did not converge). CaImAn outperformed Suite2p in all datasets with the latter obtaining $F_1$ scores in the range 0.41–0.75, with average performance $0.55 \pm 0.12$. More details about the comparison are shown in *Figure 4—figure supplement 2* and Materials and methods (Comparison with Suite2p).

## Neurons with higher SNR transients are detected more accurately

For the parameters that yielded on average the best results (see *Table 1*), both CaImAn batch and CaImAn online exhibited higher precision than recall ($0.8 \pm 0.06$ vs $0.72 \pm 0.05$ for CaImAn batch, and $0.82 \pm 0.06$ vs $0.73 \pm 0.1$ for CaImAn online, respectively). This can be partially explained by the component evaluation steps at the end of patch processing (*Figure 1e*) for CaImAn batch (and the end of each frame for CaImAn online) which aim to filter out false positive components, thus increasing precision while leaving recall intact (or in fact lowering it in case where true positive components are filtered out). To better understand this behavior, we analyzed the CaImAn batch performance as a function of the SNR of the inferred and consensus traces (*Figure 4c–e*). The SNR measure of a trace corresponds to the peak-SNR averaged over the length of a typical trace (see Materials and methods (Detecting fluorescence traces with high SNR)). An example is shown in *Figure 4c* where the scatter plot of SNR between matched consensus and inferred traces is shown (false negative/positive components are shown along the x- and y- axis, respectively). To evaluate the performance we computed a precision metric as the fraction of inferred components above a certain SNR threshold that are matched with a consensus component (*Figure 4c*, shaded blue). Similarly we computed a recall metric as the fraction of consensus components above a SNR threshold that are detected by CaImAn batch (*Figure 4c*, shaded red), and an $F_1$ score as the harmonic mean of the two (*Figure 4d*). The results indicate that the performance significantly improves as a function of the SNR for all datasets considered, improving on average from 0.73 when all neurons are considered to 0.92 when only neurons with traces having SNR $\geq 9$ are considered (*Figure 4d*). This increase in the $F_1$ score resulted increase in both the precision and the recall as a function of the SNR (*Figure 4e*) (these precision and recall metrics are computed on different sets of neurons, and therefore strictly speaking one cannot combine them to form an $F_1$ score. However, they can be bound from above by being evaluated on the set of matched and non-matched components where at least one trace is above the threshold (union of blue and pink zones in *Figure 4c*) or below by considering only matched and non-matched components where both consensus and inferred traces have SNR above the threshold (intersection of blue and pink zones in *Figure 4c*). In practice these bounds were very tight for all but one dataset (*Figure 4d*). More details can be found in Materials and methods (Performance quantification as a function of SNR)). A similar trend is also observed for CaImAn online (data not shown).

## CaImAn reproduces the consensus traces with high fidelity

Testing the quality of the inferred traces is more challenging due to the unavailability of ground truth data in the context of large scale in vivo recordings. As mentioned above, we defined as 'ground truth' the traces obtained by running the CNMF algorithm seeded with the binary masks obtained by consensus annotation procedure. After spatial alignment with the results of CaImAn , the matched traces were compared both for CaImAn batch and for CaImAn online. *Figure 5a*, shows an example of 5 of these traces for the dataset K53, showing very similar behavior of the traces in these three different cases.

To quantify the similarity we computed the correlation coefficients of the traces (consensus vs CaImAn batch, and consensus vs CaImAn online) for all the nine datasets (*Figure 5b–c*). Results indicated that for all but one dataset (*Figure 5b*) CaImAn batch reproduced the traces with higher fidelity, and in all cases the mean correlation coefficients was higher than 0.9, and the empirical histogram of correlation coefficients peaked at the maximum bin 0.99–1 (*Figure 5c*). The results indicate that the batch approach extracts traces closer to the consensus traces. This can be attributed to a number of reasons: By processing all the time points simultaneously, the batch approach can smooth the trace estimation over the entire time interval as opposed to the online approach where at each timestep only the information up to that point is considered. Moreover, CaImAn online might not detect a neuron until it becomes strongly active. This neuron's activity before detection is

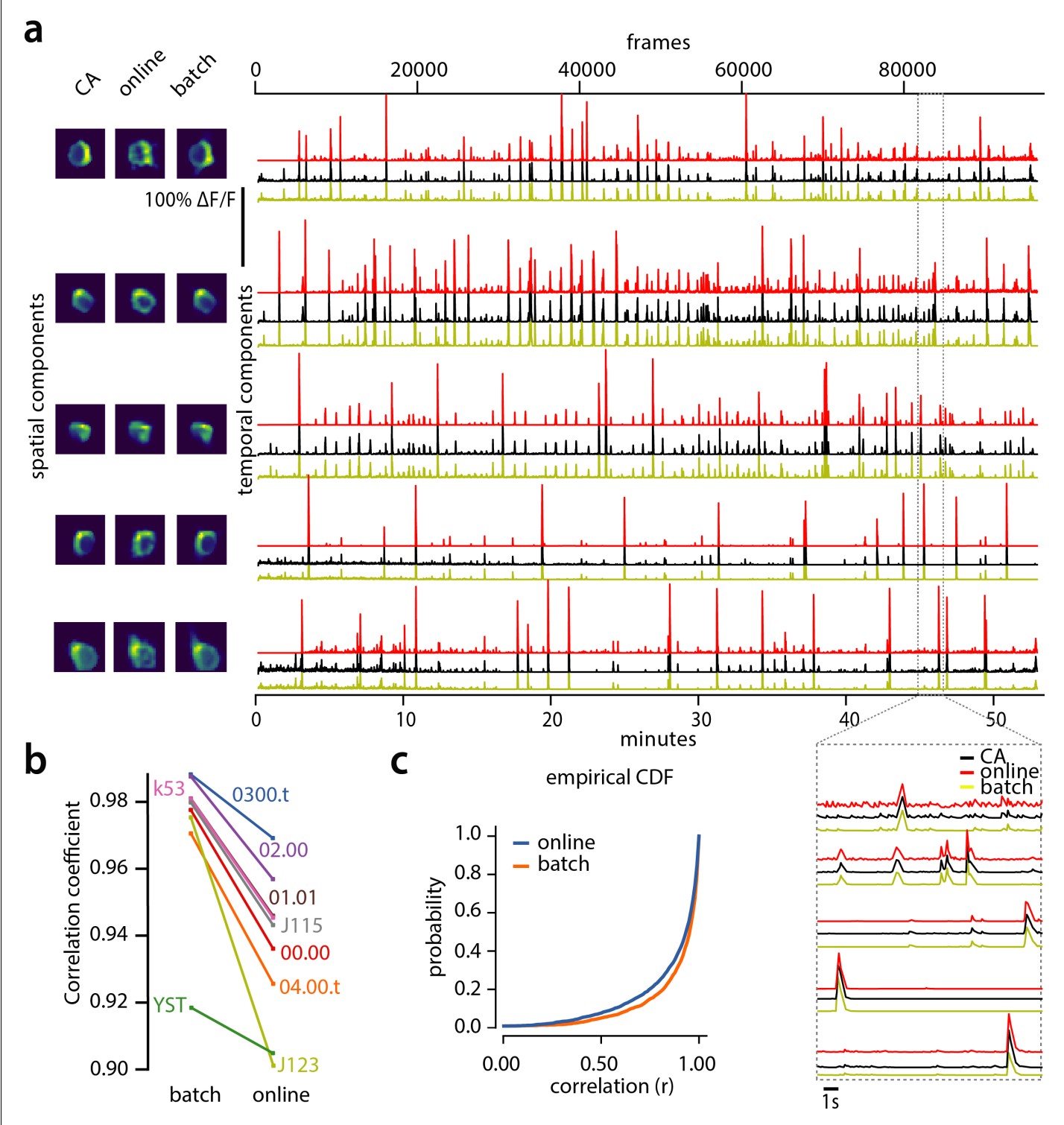

**Figure 5.** Evaluation of CAIMAN extracted traces against traces derived from consensus annotation. (a) Examples of shapes (left) and traces (right) are shown for five matched components extracted from dataset K53 for consensus annotation (CA, black), CAIMAN BATCH (yellow) and CAIMAN ONLINE (red) algorithms. The dashed gray portion of the traces is also shown magnified (bottom-right). Consensus spatial footprints and traces were obtained by seeding CAIMAN with the consensus binary masks. The traces extracted from both versions of CAIMAN match closely the consensus traces. (b) Slope graph for the average correlation coefficient for matches between consensus and CAIMAN BATCH, and between consensus and CAIMAN ONLINE. Batch processing produces traces that match more closely the traces extracted from the consensus labels. (c) Empirical cumulative distribution functions of

*Figure 5 continued on next page*

*Figure 5 continued*

correlation coefficients aggregated over all the tested datasets. Both distributions exhibit a sharp derivative close to 1 (last bin), with the batch approach giving better results.

DOI: https://doi.org/10.7554/eLife.38173.012

unknown and has a default value of zero, resulting in a lower correlation coefficient. While this can be ameliorated to a great extent with additional passes over the data, the results indicate the trade-offs inherent between online and batch algorithms.

## Online analysis of a whole brain zebrafish dataset

We tested CaImAn ONLINE with a 380 GB whole brain dataset of larval zebrafish (*Danio rerio*) acquired with a light-sheet microscope (*Kawashima et al., 2016*). The imaged transgenic fish (Tg(elavl3:H2B-GCaMP6f)jf7) expressed the genetically encoded calcium indicator GCaMP6f in almost all neuronal nuclei. Data from 45 planes (FOV $820 \times 410$ $\mu m^2$, spaced at 5.5 $\mu$m intervals along the dorso-ventral axis) was collected at 1 Hz for 30 min (for details about preparation, equipment and experiment refer to *Kawashima et al. (2016)*). With the goal of simulating real-time analysis of the data, we run all the 45 planes in parallel on a computing cluster with nine nodes (each node is equipped with 24 CPUs and 128–256 GB RAM, Linux CentoOS). Data was not stored locally in each machine but directly accessed from a network drive.

The algorithm was initialized with CaImAn BATCH run on 200 initial frames and looking for 500 components. The small number of frames (1885) and the large FOV size ($2048 \times 1188$ pixels) for this dataset motivated this choice of increased number of components during initialization. In *Figure 6* we report the results of the analysis for plane number 11 of 45. For plane 11, CaImAn ONLINE found 1524 neurons after processing 1685 frames. Since no ground truth was available for this dataset, it was only possible to evaluate the performance of this algorithm by visual inspection. CaImAn ONLINE identified all the neurons with a clear footprint in the underlying correlation image (higher SNR, *Figure 6a*) and missed a small number of the fainter ones (low SNR). By visual inspection of the components the authors could find very few false positives. Given that the parameters were not tuned and that the classifier was not trained on zebrafish neurons, we hypothesize that the algorithm is biased towards a high precision result. Spatial components displayed the expected morphological features of neurons (*Figure 6b–c*). Considering all the planes (*Figure 6e* and *Figure 6—figure supplement 1*) CaImAn ONLINE was able to identify in a single pass of the data a total of 66108 neurons. See *Video 1* for a summary across all planes. The analysis was performed in 21 min, with the first three minutes spent in initialization, and the remaining 18 in processing the data in streaming mode (and in parallel for each plane). This demonstrates the ability of CaImAn ONLINE to process large amounts of data in real-time (see also Figure 8 for a discussion of computational performance).

## Analyzing 1p microendoscopic data using CaImAn

We tested the CNMF-E implementation of CaImAn BATCH on in vivo microendosopic data from mouse dorsal striatum, with neurons expressing GCaMP6f. 6000 frames were acquired at 30 frames per second while the mouse was freely moving in an open field arena (for further details refer to *Zhou et al., 2018*). In *Figure 7* we report the results of the analysis using CaImAn BATCH with patches and compare to the results of the MATLAB implementation of *Zhou et al. (2018)*. Both implementations detect similar components (*Figure 7a*) with an $F_1$-score of 0.89. 573 neurons were found in common by both implementations. 106 and 31 additional components were detected by *Zhou et al. (2018)* and CaImAn BATCH respectively. The median correlation between the temporal traces of neurons detected by both implementations was 0.86. Similar results were also obtained by running CaImAn BATCH without patches. Ten example temporal traces are plotted in *Figure 7b*.

## Computational performance of CaImAn

We examined the performance of CaImAn in terms of processing time for the various analyzed datasets presented above (*Figure 8*). The processing time discussed here excludes motion correction, which is highly efficient and primarily depends on the level of the FOV discretization for non-rigid motion correction (*Pnevmatikakis and Giovannucci, 2017*). For CaImAn BATCH, each dataset was analyzed using three different computing architectures: (i) a single laptop (MacBook Pro) with 8

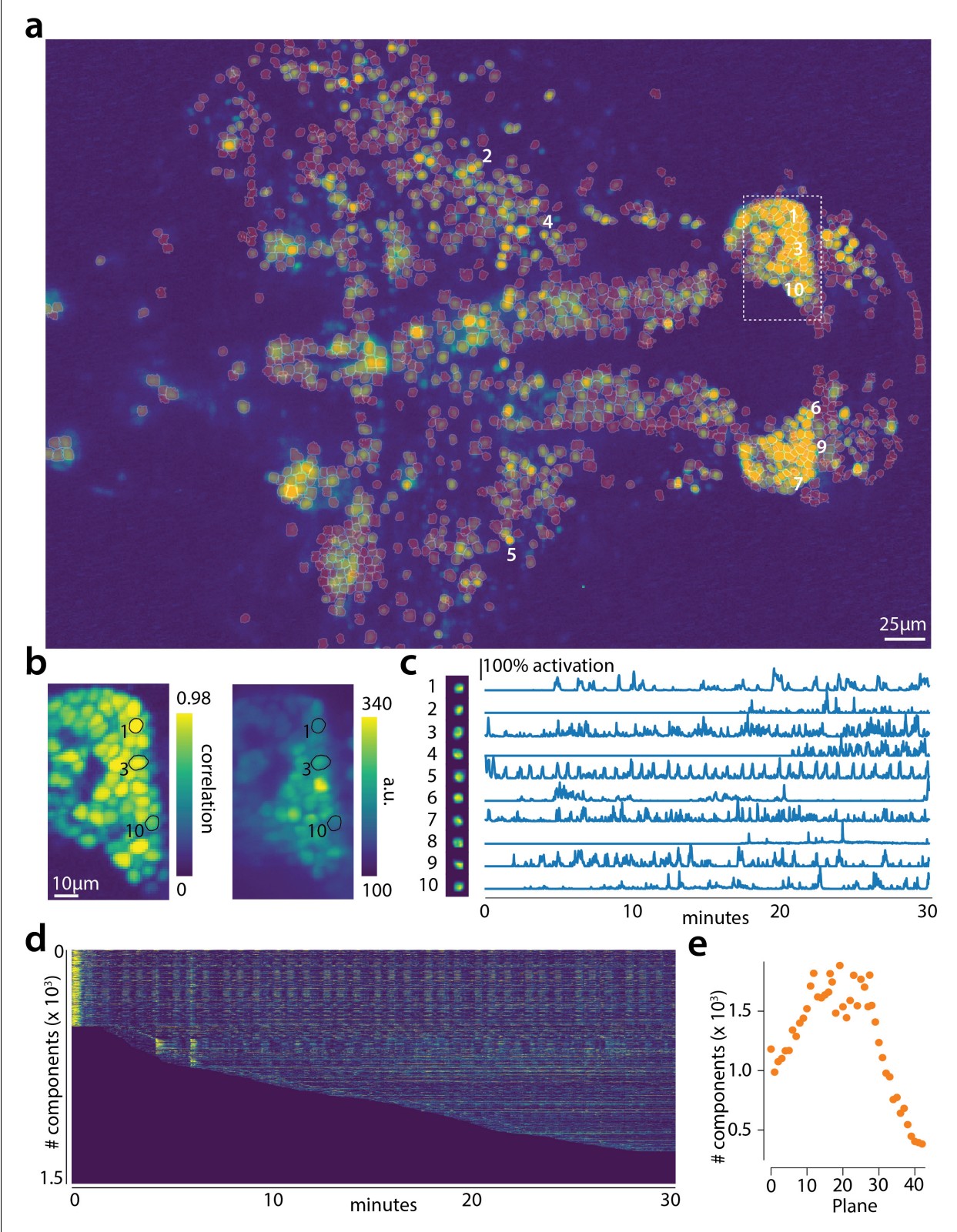

**Figure 6.** Online analysis of a 30 min long whole brain recording of the zebrafish brain. (**a**) Correlation image overlaid with the spatial components (in red) found by the algorithm (portion of plane 11 out of 45 planes in total). (**b**) Correlation image (left) and mean image (right) for the dashed region in panel (**a**) with superimposed the contours of the neurons marked in (**a**). (**c**) Spatial (left) and temporal (right) components associated to the ten example neurons marked in panel (**a**). (**d**) Temporal traces for all the neurons found in the FOV in (**a**); the initialization on the first 200 frames contained 500

*Figure 6 continued on next page*

*Figure 6 continued*

neurons (present since time 0). (**e**) Number of neurons found per plane (See also *Figure 6—figure supplement 1* for a summary of the results from all planes).

DOI: https://doi.org/10.7554/eLife.38173.013

The following video and figure supplement are available for figure 6:

**Figure supplement 1.** Spatial and temporal components for all planes.

DOI: https://doi.org/10.7554/eLife.38173.014

**Figure 6—video 1.** Results of CaImAn online initialized by CaImAn batch on a whole brain zebrafish dataset.

DOI: https://doi.org/10.7554/eLife.38173.015

CPUs (Intel Core i7) and 16 GB of RAM (blue in *Figure 8a*), (ii) a linux-based workstation (CentOS) with 24 CPUs (Intel Xeon CPU E5-263 v3 at 3.40 GHz) and 128 GB of RAM (magenta), and (iii) a linux-based HPC cluster (CentOS) where 112 CPUs (Intel Xeon Gold 6148 at 2.40 GHz, four nodes, 28 CPUs each) were allocated for the processing task (yellow). *Figure 8a* shows the processing of CaImAn batch as a function of dataset size on the four longest datasets, whose size exceeded 8 GB, on log-log plot.

Results show that, as expected, employing more processing power results in faster processing. CaImAn batch on a HPC cluster processes data faster than acquisition time (*Figure 8a*) even for very large datasets. Processing of an hour long dataset was feasible within 3 hr on a single laptop, even though the size of the dataset is several times the available RAM. Here, acquisition time is computed based on the assumption of imaging a FOV discretized over a $512 \times 512$ grid at a 30 Hz rate (a typical two-photon imaging setup with resonant scanning microscopes). Dataset size is computed by representing each measurement using single precision arithmetic, which is the minimum precision required for standard algebraic processing. These assumptions lead to a data rate of $\sim 105$ GB/hr. In general performance scales linearly with the number of frames (and hence, the size of the dataset), but we also observe a dependency on the number of components, which during the solution refinement step can be quadratic. This is expected from the properties of the matrix factorization approach as also noted by past studies (*Pnevmatikakis et al., 2016*). The majority of the time (*Figure 8b*) required for CaImAn batch processing is taken by CNMF algorithmic processing either during the initialization in patches (orange bar) or during merging and refining the results of the individual patches (green bar).

To study the effects of parallelization we ran CaImAn batch several times on the same hardware (linux-based workstation with 24CPUs), limiting the runs to different numbers of CPUs each time (*Figure 8c*). In all cases we saw significant performance gains from parallel processing, with the gains being similar for all stages of processing (patch processing, refinement, and quality testing, data not shown). We saw the most effective scaling with our 50 G dataset (J123). For the largest datasets (J115, $\sim 100$GB), the speedup reaches a plateau due to limited available RAM, suggesting that more RAM can lead to better scaling. For small datasets ($\sim 5$ GB) the speedup factor is limited by increased communications overhead (indicative of *weak scaling* in the language of high performance computing).

The cost of processing 1p data in CaImAn batch using the CNMF-E algorithm (*Zhou et al., 2018*) is shown (*Figure 8d*) for our workstation-

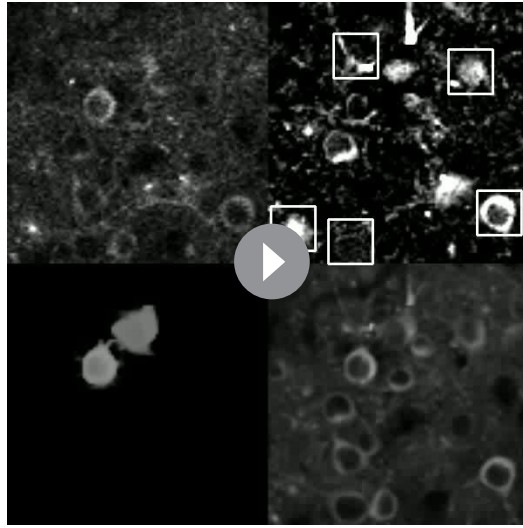

**Video 1.** Depiction of CaImAn online on a small patch of in vivo cortex data. Top left: Raw data. Bottom left: Footprints of identified components. Top right: Mean residual buffer and proposed regions for new components (in white squares). Enclosings of accepted regions are shown in magenta. Several regions are proposed multiple times before getting accepted. This is due to the strict behavior of the classifier to ensure a low number of false positives. Bottom right: Reconstructed activity.

DOI: https://doi.org/10.7554/eLife.38173.016

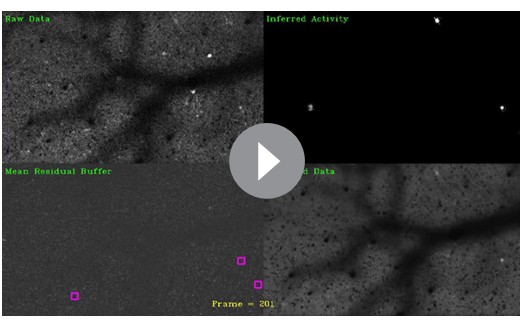

**Video 2.** Depiction of CAIMAN ONLINE on a single plane of mesoscope data courtesy of E. Froudarakis, J. Reimers and A. Tolias (Baylor College of Medicine). Top left: Raw data. Top right: Inferred activity (without neuropil). Bottom left: Mean residual buffer and accepted regions for new components (magenta squares). Bottom right: Reconstructed activity.
DOI: https://doi.org/10.7554/eLife.38173.017

class hardware. Splitting in patches and processing in parallel can lead to computational gains at the expense of increased memory usage. This is because the CNMF-E introduces a background term that has the size of the dataset and needs to be loaded and updated in memory in two copies. This leads to processing times that are slower compared to the standard processing of 2 p datasets, and higher memory requirements. However ( 8 c), memory usage can be controlled enabling scalable inference at the expense of slower processing speeds.

*Figure 8a* also shows the performance of CAIMAN ONLINE (red markers). Because of the low memory requirements of the streaming algorithm, this performance only mildly depends on the computing infrastructure, allowing for near real-time processing speeds on a standard laptop (*Figure 8a*). As discussed in *Giovannucci et al., 2017* processing time of CAIMAN ONLINE depends primarily on (i) the computational cost of tracking the temporal activity of discovered neurons, (ii) the cost of detecting and incorporating new neurons, and (iii) the cost of periodic updates of spatial footprints. *Figure 8e* shows the cost of each of these steps for each frame, for one epoch of processing of the dataset J123. Distributing the spatial footprint update more uniformly among all frames removes the computational bottleneck appearing in *Giovannucci et al., 2017*, where all the footprints where updated periodically at the same frame. The cost of detecting and incorporating new components remains approximately constant across time, and is dependent on the number of candidate components at each timestep. In this example five candidate components were used per frame resulting in a relatively low cost ($\sim 7$ ms per frame). A higher number of candidate components can lead to higher recall in shorter datasets but at a computational cost. This step can benefit by the use of a GPU for running the online CNN on the footprints of the candidate components. Finally, as noted in *Giovannucci et al., 2017*, the cost of tracking components can be kept low, and increases mildly over time as more components are found by the algorithm (the analysis here excludes the cost of motion correction, because the files where motion corrected before hand to ensure that manual annotations and the algorithms where operating on the same FOV. This cost depends on whether rigid or pw-rigid motion correction is being used. Rigid motion correction taking on average 3–5 ms per frame for a $512 \times 512$ pixel FOV, whereas pw-rigid motion correction with patch size $128 \times 128$ pixel is typically 3–4 times slower).

*Figure 8f* shows the overall processing speed (in frames per second) for CAIMAN ONLINE for the nine annotated datasets. Apart from the number of neurons, the processing speed also depends on the size of the imaged FOV and the use of spatial downsampling. Datasets with smaller FOV (e.g., YST) or datasets where spatial downsampling is used can achieve higher processing speeds for the same amount of neurons (blue dots in *Figure 8f*) as opposed to datasets where no spatial downsampling is used (orange dots in *Figure 8f*). In most cases, spatial downsampling can be used to increase processing speed without significantly affecting the quality of the results, an observation consistent with previous studies (*Friedrich et al., 2017a*).

In *Figure 8g* the cost per frame is plotted for the analysis of the whole brain zebrafish recording. The lower imaging rate (1 Hz) allows for tracing of neural activity with computational costs significantly lower than the 1 s between volume imaging time (*Figure 8e*), even in the presence of a large number of components (typically more than 1000 per plane, *Figure 6*) and the significantly larger FOV ($2048 \times 1188$ pixels).

## CAIMAN successfully tracks neurons across multiple days

*Figure 9* shows an example of tracking neurons across six different sessions corresponding to six different days of mouse cortex in vivo data using our multi-day registration algorithm REGISTERMULTI (see Materials and methods, Algorithm 8). 453, 393, 375, 378, 376, and 373 active components were

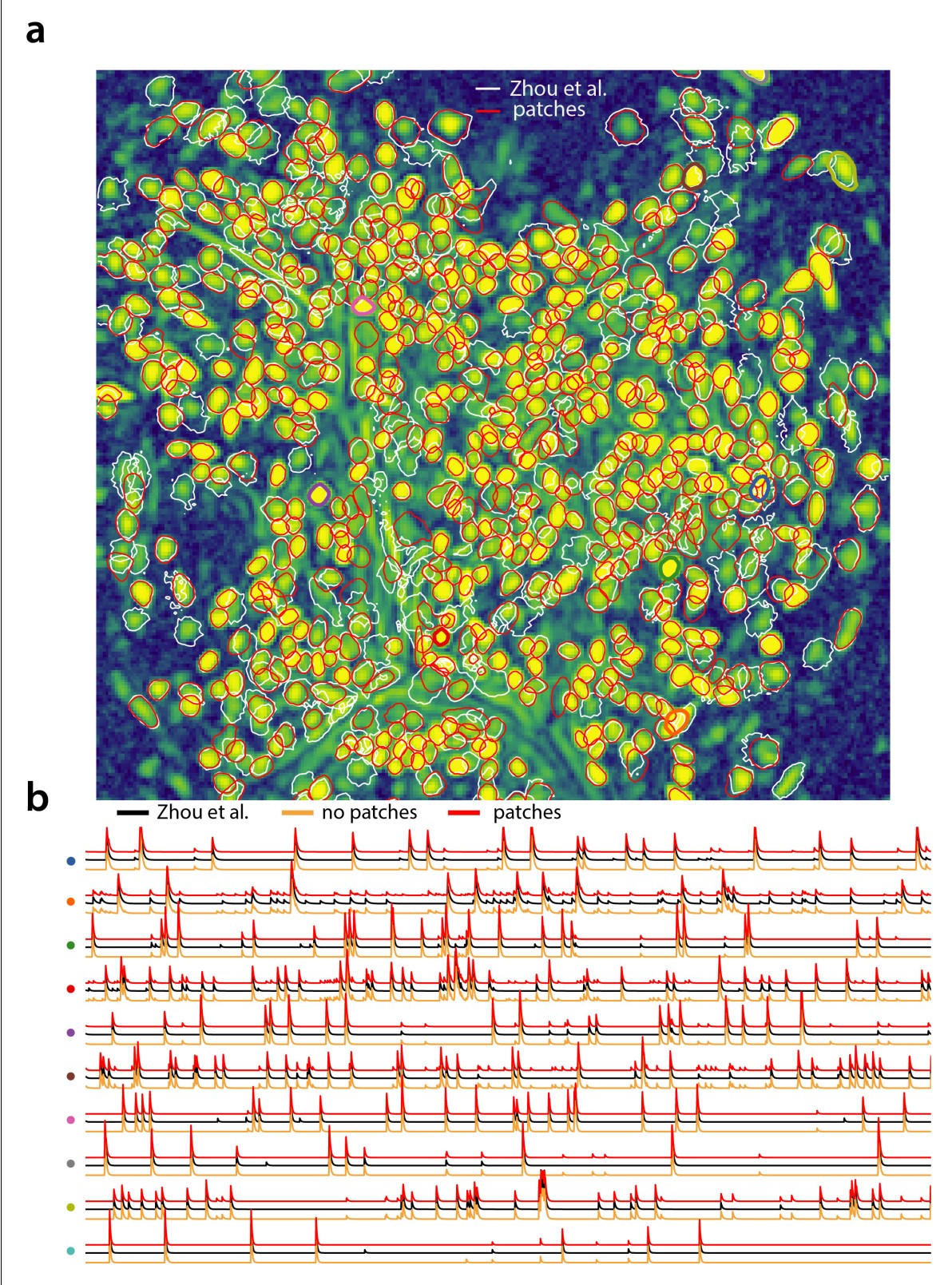

**Figure 7.** Analyzing microendoscopic 1 p data with the CNMF-E algorithm using CaImAn batch . (a) Contour plots of all neurons detected by the CNMF-E (white) implementation of *Zhou et al. (2018)* and CaImAn batch (red) using patches. Colors match the example traces shown in (b), which illustrate the temporal components of 10 example neurons detected by both implementations CaImAn batch . reproduces with reasonable fidelity the results of *Zhou et al. (2018)*.

*Figure 7 continued*

DOI: https://doi.org/10.7554/eLife.38173.018

found in the six sessions, respectively. Our tracking method detected a total of 686 distinct active components. Of these, 172, 108, 70, 92, 82, and 162 appeared in exactly 1, 2, 3, 4, 5, and all six sessions respectively. Contour plots of the 162 components that appeared in all sessions are shown in *Figure 9a*, and parts of the FOV are highlighted in *Figure 9d* showing that components can be tracked in the presence of non-rigid deformations of the FOV between the different sessions.

To test the stability of REGISTERMULTI for each subset of sessions, we repeated the same procedure running backwards in time starting from day 6 and ending at day 1, a process that also generated a total of 686 distinct active components. We identified the components present in *at least* a given subset of sessions when using the forward pass, and separately when using the backwards pass, and compared them against each other (*Figure 9b*) for all possible subsets. Results indicate a very high level of agreement between the two approaches with many of the disagreements arising near the boundaries (data not shown). Disagreements near the boundaries can arise because the forward pass aligns the union with the FOV of the last session, whereas the backwards pass with the FOV of the first session, potentially leading to loss of information near the boundaries.

A step by step demonstration of the tracking algorithm for the first three sessions is shown in *Figure 9—figure supplement 1*. Our approach allows for the comparison of two non-consecutive sessions through the union of components without the need of a direct pairwise registration (*Figure 9—figure supplement 1f*), where it is shown that registering sessions 1 and 3 directly and through the union leads to nearly identical results. *Figure 9c* compares the registrations for all pairs of sessions using the forward (red) or the backward (blue) approach, with the direct pairwise registrations. Again, the results indicate a very high level of agreement, indicating the stability and effectiveness of the proposed approach.

A different approach for multiple day registration was recently proposed by *Sheintuch et al. (2017)* (CELLREG). While a direct comparison of the two methods is not feasible in the absence of ground truth, we tested our method against the same publicly available datasets from the Allen Brain Observatory visual coding database. (http://observatory.brain-map.org/visualcoding). Similarly to *Sheintuch et al. (2017)* the same experiment performed over the course of different days produced very different populations of active neurons. To measure performance of REGISTERPAIR for pairwise registration, we computed the transitivity index proposed in *Sheintuch et al. (2017)*. The transitivity property requires that if cell 'a' from session one matches with cell 'b' from session 2, and cell 'b' from session two matches with cell 'c' from session 3, then cell 'a' from session one should match with cell 'c' from session 3 when sessions 1 and 3 are registered directly. For all ten tested datasets the transitivity index was very high, with values ranging from 0.976 to 1 ($0.992 \pm 0.006$, data not shown). A discussion between the similarities and differences of the two methods is given in Materials and methods.

## Discussion

### Reproducible and scalable analysis for the 99%

Significant advances in the reporting fidelity of fluorescent indicators, and the ability to simultaneously record and modulate neurons granted by progress in optical technology, have made calcium imaging one of the two most prominent experimental methods in systems neuroscience alongside electrophysiology recordings. Increasing adoption has led to an unprecedented wealth of imaging data which poses significant analysis challenges. CAIMAN is designed to provide the experimentalist with a complete suite of tools for analyzing this data in a formal, scalable, and reproducible way. The goal of this paper is to present the features of CAIMAN and examine its performance in detail. CAIMAN embeds existing methods for preprocessing calcium imaging data into a MapReduce framework and augments them with supervised learning algorithms and validation metrics. It builds on the CNMF algorithm of *Pnevmatikakis et al. (2016)* for source extraction and deconvolution, extending it along the lines of (i) reproducibility and performance improvement, by automating quality assessment through the use of unsupervised and supervised learning algorithms for component detection and classification, and (ii) scalability, by enabling fast large scale processing with standard computing

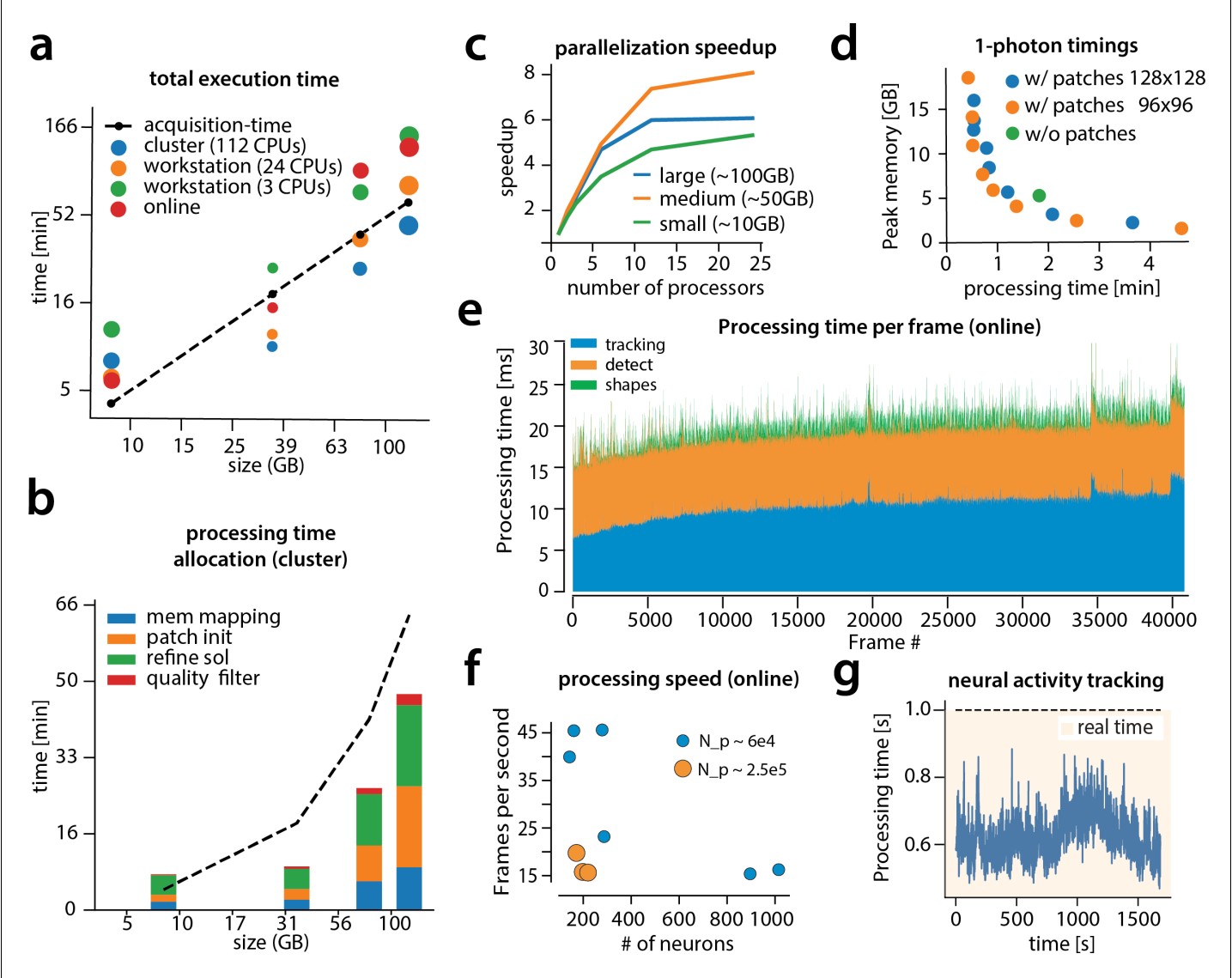

**Figure 8.** Time performance of CAIMAN BATCH and CAIMAN ONLINE for four of the analyzed datasets (small, medium, large and very large). (**a**) Log-log plot of total processing time as a function of data size for CAIMAN BATCH two-photon datasets using different processing infrastructures: (i) a desktop with three allocated CPUs (green), (ii) a desktop with 24 CPUs allocated (orange), and (iii) a HPC where 112 CPUs are allocated (blue). The results indicate a near linear scaling of the processing time with the size of dataset, with additional dependence on the number of found neurons (size of each point). Large datasets (>100 GB) can be seamlessly processed with moderately sized desktops or laptops, but access to a HPC enables processing with speeds faster than the acquisition time (considered 30 Hz for a 512×512 FOV here). However, for smaller datasets the advantages of adopting a cluster vanishes, because of the inherent overhead. The results of CAIMAN ONLINE using the laptop, using the 'strict' parameter setting (**Figure 4—figure supplement 1**), are also plotted in red indicating near real-time processing speed. (**b**) Break down of processing time for CAIMAN BATCH (excluding motion correction). Processing with CNMF in patches and refinement takes most of the time for CAIMAN BATCH. (**c**) Computational gains for CAIMAN BATCH due to parallelization for three datasets with different sizes. The parallelization gains are computed by using the same 24 CPU workstation and utilizing a different number of CPUs for each run. The different parts of the algorithm exhibit the same qualitative characteristics (data not shown). (**d**) Cost analysis of CNMF-E implementation for processing a 6000 frames long 1p dataset. Processing in patches in parallel induces a time/memory tradeoff and can lead to speed gains (patch size in legend). (**e**) Computational cost per frame for analyzing dataset J123 with CAIMAN ONLINE. Tracking existing activity and detecting new neurons are the most expensive steps, whereas udpating spatial footprints can be efficiently distributed among all frames. (**f**) Processing speed of CAIMAN ONLINE for all annotated datasets. Overall speed depends on the number of detected neurons and the size of the FOV ($N_p$ stands for number of pixels). Spatial downsampling can speed up processing. (**g**) Cost of neural activity online tracking for the whole brain zebrafish dataset (maximum time over all planes per volume). Tracking can be done in real-time using parallel processing.

DOI: https://doi.org/10.7554/eLife.38173.019

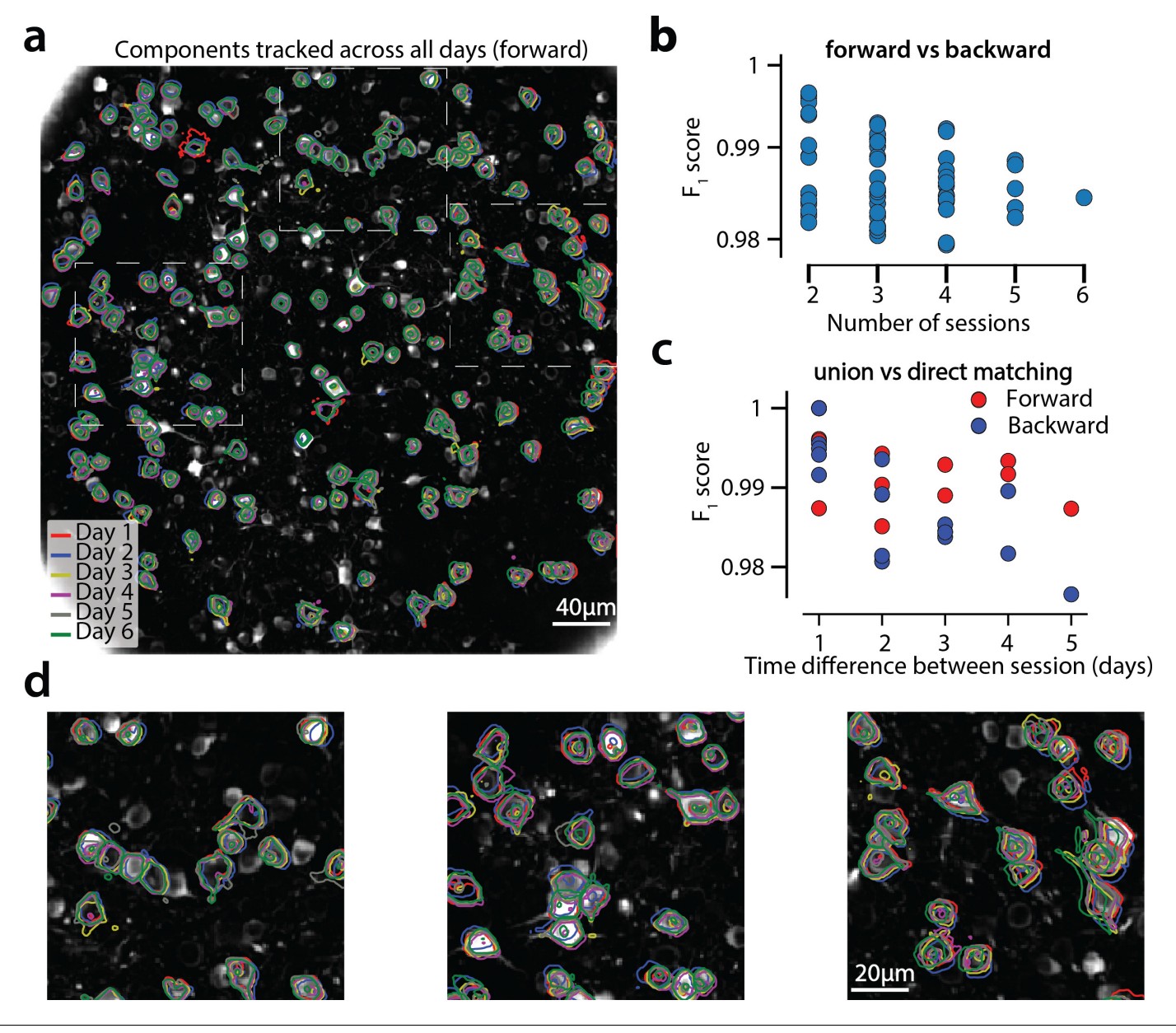

**Figure 9.** Components registered across six different sessions (days). (**a**) Contour plots of neurons that were detected to be active in all six imaging sessions overlaid on the correlation image of the sixth imaging session. Each color corresponds to a different session. (**b**) Stability of multiday registration method. Comparisons of forward and backward registrations in terms of $F_1$ scores for all possible subsets of sessions. The comparisons agree to a very high level, indicating the stability of the proposed approach. (**c**) Comparison (in terms of $F_1$ score) of pair-wise alignments using readouts from the union vs direct alignment. The comparison is performed for both the forward and the backwards alignment. For all pairs of sessions the alignment using the proposed method gives very similar results compared to direct pairwise alignment. (**d**) Magnified version of the tracked neurons corresponding to the squares marked in panel (**a**). Neurons in different parts of the FOV exhibit different shift patterns over the course of multiple days, but can nevertheless be tracked accurately by the proposed multiday registration method.

DOI: https://doi.org/10.7554/eLife.38173.020

The following figure supplement is available for figure 9:

**Figure supplement 1.** Tracking neurons across days; step-by-step description of multi session registration.

DOI: https://doi.org/10.7554/eLife.38173.021

infrastructure (e.g., a commodity laptop or workstation). Scalability is achieved by either using a MapReduce batch approach, which employs parallel processing of spatially overlapping, memory mapped, data patches; or by integrating the online processing framework of *Giovannucci et al., 2017* within our pipeline. Apart from computational gains both approaches also result in improved performance. Towards our goal of providing a single package for dealing with standard problems arising in analysis of imaging data, CaImAn also includes an implementation of the CNMF-E algorithm of *Zhou et al. (2018)* for the analysis of microendoscopic data, as well as a novel method for registering analysis results across multiple days.

## Towards surpassing human neuron detection performance

To evaluate the performance of CaImAn batch and CaImAn online, we used a number of distinct labelers to generate a corpus of nine annotated two-photon imaging datasets. The results indicated a surprising level of disagreement between individual labelers, highlighting both the difficulty of the problem, and the non-reproducibility of the laborious task of human annotation. CaImAn reached near-human performance with respect to this consensus annotation, by using the *same* parameters for all the datasets without dataset dependent parameter tweaking. Such tweaking can include setting the SNR threshold based on the noise level of the recording, the complexity of the neuropil signal based on the level of background activity, or specialized treatment around the boundaries of the FOV to compensate for eventual imaging artifacts, and as shown can significantly improve the results on individual datasets. As demonstrated in our results, optimal parameter setting for CaImAn online can also depend on the length of the experiment with stricter parameters being more suitable for longer datasets. We plan to investigate parameter schemes that increase in strictness over the course of an experiment.

CaImAn has higher precision than recall when run on most datasets. While more balanced results can be achieved by appropriately relaxing the relevant quality evaluation thresholds, we prefer to maintain a higher precision as we believe that the inclusion of false positive traces can be more detrimental in any downstream analysis compared to the exclusion of, typically weak, true positive traces. This is true especially in experiments with low task dimensionality where a good signal from few neurons can be sufficient for the desired hypothesis testing.

Apart from being used as a benchmarking tool, the set of manual annotations can also be used as labeled data for supervised learning algorithms. CaImAn uses two CNN based classifiers trained on (a subset of) this data, one for post processing component classification in CaImAn batch, and the other for detecting new neurons in residual images in the CaImAn online. The deployment of these classifiers resulted in significant gains in terms of performance, and we expect further advances in the future. The annotations are made freely available to the community for benchmarking and training purposes.

## CaImAn batch vs CaImAn online

Our results suggest similar performance between CaImAn batch and CaImAn onine when evaluated on the basis of processing speed and quality of results, with CaImAn online outperforming CaImAn batch on longer datasets in terms of neuron detection, possibly due to its inherent ability to adapt to non-stationarities arising during the course of a large experiment. By contrast, CaImAn batch extracts better traces compared to CaImAn online with respect to the traces derived from the consensus annotations. While multiple passes over the data with CaImAn online can mitigate these shortcomings, this still depends on good initialization with CaImAn batch, as the analysis of the whole brain zebrafish dataset indicates. In offline setups, CaImAn onine could also benefit from the post processing component evaluation tools used in batch mode. for example using the batch classifier for detecting false positive components at the end of the experiment.

CaImAn online differs from CaImAn batch in that the former has lower memory requirements and it can support novel types of closed-loop all-optical experiments (*Packer et al., 2015*; *Carrillo-Reid et al., 2017*). As discussed in *Giovannucci et al., 2017*, typical all-optical closed-loop experiments require the pre-determination of ROIs that are monitored/modulated. Indeed, CaImAn online allows identification and modulation of new neurons on the fly, greatly expanding the space of possible experiments. Even though our simulated online processing setup is not integrated with hardware to an optical experimental setup, our results indicate thatCaImAn online performed close to real-time

in most cases. Real time can be potentially achieved by using parallel computational streams for the three steps of frame processing (motion correction and tracking, detecting new neurons, updating shapes), since these steps can be largely run in an asynchronous mode independently. This suggests that large scale closed-loop experiments with single cell resolution are feasible by combining existing all-optical technology and our proposed analysis method.

### Future directions

While CAIMAN uses a highly scalable processing pipeline for two-photon datasets, processing of one-photon microendoscopic imaging data is less scalable due to the more complex background model that needs to be retained in memory during processing. Adapting CAIMAN ONLINE to the one-photon data processing algorithm of *Zhou et al. (2018)* is a promising way for scaling up efficient processing in this case. The continuing development and quality improvement of neural activity indicators has enabled direct imaging of neural processes (axons/dendrites), imaging of synaptic activity (*Xie et al., 2016*), or direct imaging of voltage activity in vivo conditions (*Piatkevich et al., 2018*). While the approach presented here is tuned for somatic imaging through the use of various assumptions (space localized activity, CNN classifiers trained on images of somatic activity), the technology of CAIMAN is largely transferable to these domains as well. We will pursue these extensions in future work.

## Materials and methods

### Memory mapping

CAIMAN BATCH uses memory mapping for efficient parallel data access. With memory mapped arrays, arithmetic operations can be performed on data residing on the hard drive without explicitly loading it to RAM, and slices of data can be indexed and accessed without loading the full file in memory, enabling out-of-core processing (*Toledo, 1999*). On modern computers tensors are stored in linear format, no matter the number of the array dimensions. Therefore, one has to decide which elements of an array are contiguous in memory: in *row-major order*, consecutive elements of a row (first-dimension) are next to each other, whereas in *column-major order* consecutive elements of a column (last dimension) are contiguous. Such decisions significantly affect the speed at which data is read or written on spinning disks (and to a lesser degree on solid state drives): in *column-major order* reading a full column is fast because memory is read in a single sequential block, whereas reading a row is inefficient since only one element can be read at a time and all the data needs to be accessed.

In the context of calcium imaging datasets, CAIMAN BATCH represents the datasets in a matrix form $Y$, where each row corresponds to a different imaged pixel, and each column to a different frame. As a result, a *column-major order* mmap file enables the fast access of individual frames at a given time, whereas a *row-major order* files enables the fast access of an individual pixel at all times. To facilitate processing in patches CAIMAN BATCH stores the data in *row-major order*. In practice, this is opposite to the order with which the data appears, one frame at a time. In order to reduce memory usage and speed up computation CAIMAN BATCH employs a MapReduce approach, where either multiple files or multiple chunks of a big file composing the original datasets are processed and saved in mmap format in parallel. This operation includes two phases, first the chunks/files are saved (possibly after motion correction, if required) in multiple row-major mmap format, and then chunks are simultaneously combined into a single large row-major mmap file.

### Mathematical model of the CNMF framework

The CNMF framework (*Figure 1d*) for calcium imaging data representation can be expressed in mathematical terms as (*Pnevmatikakis et al., 2016*)

$$Y = AC + B + E. \tag{1}$$

Here, $Y \in \mathbb{R}^{d \times T}$ denotes the observed data written in matrix form, where $d$ is the total number of observed pixels/voxels, and $T$ is the total number of observed timesteps (frames). $A \in \mathbb{R}^{d \times N}$ denotes the matrix of the $N$ spatial footprints, $A = [\mathbf{a}_1, \mathbf{a}_2, \ldots, \mathbf{a}_N]$, with $\mathbf{a}_i \in \mathbb{R}^{d \times 1}$ being the spatial footprint of component $i$. $C \in \mathbb{R}^{N \times T}$ denotes the matrix of temporal components, $C = [\mathbf{c}_1, \mathbf{c}_2, \ldots, \mathbf{c}_N]^\top$, with

**Table 3.** Cross-validated results of each labeler, where each labeler's performance is compared against the annotations of the rest of the labelers using a majority vote.

Results are given in the form $F_1$ score (precision, recall), and empty entries correspond to datasets not manually annotated by the specific labeler. The results indicate decreased performance compared to the consensus annotation annotations.

| Name | L1 | L2 | L3 | L4 | Mean |
|---|---|---|---|---|---|
| N.01.01 | 0.75 (0.73, 0.77) | 0.70 (0.58, 0.88) | 0.86 (0.81, 0.90) | 0.84 (0.92, 0.77) | 0.79 (0.76, 0.83) |
| N.03.00.t | X | 0.75 (0.69, 0.82) | 0.79 (0.67, 0.97) | 0.85 (0.76, 0.97) | 0.8 (0.71, 0.92) |
| N.00.00 | X | 0.87 (0.84, 0.90) | 0.82 (0.75, 0.91) | 0.72 (0.71, 0.97) | 0.83 (0.76, 0.93) |
| YST | 0.7 (0.93, 0.56) | 0.79 (0.7, 0.9) | 0.81 (0.76, 0.86) | 0.77 (0.75, 0.78) | 0.77 (0.78, 0.78) |
| N.04.00.t | X | 0.79 (0.76, 0.83) | 0.72 (0.60, 0.89) | 0.68 (0.53, 0.96) | 0.73 (0.63, 0.89) |
| N.02.00 | 0.84 (0.97, 0.75) | 0.88 (0.89, 0.87) | 0.86 (0.79, 0.94) | 0.81 (0.7, 0.95) | 0.85 (0.83, 0.88) |
| J123 | X | 0.9 (0.86, 0.93) | 0.89 (0.84, 0.93) | 0.77 (0.63, 0.96) | 0.87 (0.88, 0.88) |
| J115 | 0.85 (0.98, 0.76) | 0.87 (0.80, 0.97) | 0.88 (0.80, 0.97) | 0.87 (0.93, 0.82) | 0.85 (0.78, 0.94) |
| K53 | 0.86 (0.98, 0.77) | 0.9 (0.85, 0.96) | 0.88 (0.8, 0.96) | 0.89 (0.9, 0.88) | 0.88 (0.88, 0.89) |
| mean ± std | 0.8±0.06 (0.92±0.09, 0.72±0.08) | 0.83±0.07 (0.77±0.1, 0.9±0.05) | 0.83±0.05 (0.76±0.07, 0.92±0.04) | 0.81±0.06 (0.76±0.13, 0.9±0.08) | 0.82±0.06 (0.77±0.12, 0889±0.06) |

DOI: https://doi.org/10.7554/eLife.38173.022

$\mathbf{c}_i \in \mathbb{R}^{T \times 1}$ being the temporal trace of component $i$. $B$ is the background/neuropil activity matrix. For two-photon data it is modeled as a low rank matrix $B = \mathbf{bf}$, where $\mathbf{b} \in \mathbb{R}^{d \times n_b}, \mathbf{f} \in \mathbb{R}^{n_b \times T}$ correspond to the matrices of spatial and temporal background components, and $n_b$ is the number of background components. For the case of micro-endoscopic data the integration volume is much larger and the low rank model is inadequate. For this we use the CNMF-E algorithm of *Zhou et al. (2018)* where the background is modeled as

$$B = W(Y - AC), \tag{2}$$

where $W \in \mathbb{R}^{d \times d}$ is an appropriate weight matrix, where the $(i,j)$ entry models the influence of the neuropil signal of pixel $j$ to the neuropil signal at pixel $i$.

## Combining results from different patches

To combine results from different patches we first need to account for the overlap at the boundaries. Neurons lying close to the boundary between neighboring patches can appear multiple times and must be merged. With this goal, we optimized the merging approach used in *Pnevmatikakis et al. (2016)*: Groups of components with spatially overlapping footprints whose temporal traces are correlated above a threshold are replaced with a single component, that tries to explain as much of the variance already explained by the 'local' components (as opposed to the variance of the data as performed in *Pnevmatikakis et al. (2016)*). If $A_{\mathrm{old}}, C_{\mathrm{old}}$ are the matrices of components to be merged, then the merged component $\mathbf{a}_m, \mathbf{c}_m$ are given by the solution of the rank-1 NMF problem:

$$\min_{\mathbf{a}_m \geq 0, \mathbf{c}_m \geq 0} \|A_{\mathrm{old}} C_{\mathrm{old}} - \mathbf{a}_m \mathbf{c}_m^\top\|. \tag{3}$$

Prior to merging, the value of each component at each pixel is normalized by the number of patches that overlap in this pixel, to avoid counting the activity of each pixel multiple times.

We follow a similar procedure for the background/neuropil signals from the different patches. When working with two-photon data, the spatial background/neuropil components for each patch can be updated by keeping their spatial extent intact to retain a local neuropil structure, or they can

be merged when they are sufficiently correlated in time as described above to promote a more global structure. For the case of one-photon data, CNMF-E estimates the background using a local autoregressive process (see *Equation 2*) (*Zhou et al., 2018*), a setup that cannot be immediately propagated when combining the different patches. To combine backgrounds from the different patches, we first approximate the backgrounds $B^i$ from all the patches $i$ with a low rank matrix using non-negative matrix factorization of rank $g_b$ to obtain global spatial, and temporal background components.

$$[\mathbf{b}^i, \mathbf{f}^i] = \text{NNMF}(B^i, g_b). \tag{4}$$

The resulting components are embedded into a large matrix $B \in \mathbb{R}^{d \times T}$ that retains a low rank structure. After the components and backgrounds from all the patches have been combined, they are further refined by running CNMF iteration of updating spatial footprints, temporal traces, and neuropil activity. CAIMAN BATCH implements these steps in parallel (as also described in *Pnevmatikakis et al. (2016)*): Temporal traces whose corresponding spatial traces do not overlap can be updated in parallel. Similarly, the rows of the matrix of spatial footprints $A$ can also be updated in parallel (*Figure 2b*). The process is summarized in algorithmic format in Algorithms 1–2. When working with one-photon data, instead of producing a low-rank approximation of $B$ that would underfit the background, we increase patch overlap and run the full pipeline on each patch. In the final phase, when neurons overlap we retain only the variant with the highest quality rather than merging them.

## Initialization strategies

Source extraction using matrix factorization requires solving a bi-convex problem where initialization plays a critical role. The CNMF/CNMF-E algorithms use initialization methods that exploit the locality of the spatial footprints to efficiently identify the locations of candidate components (*Pnevmatikakis et al., 2016*; *Zhou et al., 2018*). CAIMAN incorporates these methods, extending them by using the temporal locality of the calcium transient events. The available initialization methods for CAIMAN BATCH include:

GREEDYROI: This approach, introduced in *Pnevmatikakis et al. (2016)*, first spatially smooths the data with a Gaussian kernel of size comparable to the average neuron radius, and then initializes candidate components around locations where maximum variance (of the smoothed data) is explained. This initialization strategy is fast but requires manual specification of the number of components by the user.

ROLLINGGREEDYROI: The approach, introduced in this paper, operates like GREEDYROI by spatially smoothing the data and looking for points of maximum variance. Instead of working across all the data, ROLLINGGREEDYROI looks for points of maximum variance on a rolling window of a fixed duration, for example 3 s, and initializes components by performing a rank one NMF on a local spatial neighborhood. By focusing into smaller rolling windows, ROLLINGGREEDYROI can better isolate single transient events, and as a result detect better neurons with sparse activity. ROLLINGGREEDYROI is the default choice for processing of 2-photon data.

GREEDYCORR: This approach, introduced in *Zhou et al. (2018)*, initializes candidate components around locations that correspond to the local maxima of an image formed by the pointwise product between the correlation image and the peak signal-to-noise ratio image. A threshold for acceptance of candidate neurons is used, making it unnecessary to pre-specify the neuron count. This comes at the expense of a higher computational cost. GREEDYCORR is the default choice for processing of one-photon data.

SPARSENMF: Sparse NMF approaches, when ran in small patches, can be effective for quickly uncovering spatial structure in the imaging data, especially for neural processes (axons/dendrites) whose shape cannot be easily parametrized and/or localized.

SEEDEDINITIALIZATION: Often locations of components are known either from manual annotation or from labeled data obtained in a different way, such as data from a static structural channel recorded concurrently with the functional indicator. CAIMAN can be seeded with binary (or real valued) masks for the spatial footprints. Apart from $A$, these masks can be used to initialize all the other relevant matrices $C$ and $B$ as well. This is performed by (i) first estimating the temporal background components $\mathbf{f}$ using only data from parts of the FOV not covered by any masks and, (ii) then estimating the

spatial background components $\mathbf{b}$, and then estimating $A, C$ (with $A$ restricted to be non-zero only at the locations of the binary masks), using a simple NMF approach. Details are given in Algorithm 3.

## Details of quality assessment tests

Here we present the unsupervised and supervised quality assessment tests in more detail (*Figure 2*).

### Matching spatial footprints to the raw data

Let $\mathbf{a}_i, \mathbf{c}_i$ denote the spatial footprint and temporal trace of component $i$, and the let $A_{\backslash i}, C_{\backslash i}$ denote the matrices $A, C$ when the component $i$ has been removed. Similarly, let $Y_i = Y - A_{\backslash i}C_{\backslash i} - B$ denote the entire dataset when the background and the contribution of all components except $i$ have been removed. If component $i$ is real then $Y_i$ and $\mathbf{a}_i\mathbf{c}_i^\top$ will look similar during the time intervals when the component $i$ is active. As a first test CaImAn finds the first $N_p$ local peaks of $c_i$ (e.g., $N_p = 5$), constructs intervals around these peaks, (e.g., 50 ms in the past and 300 ms in the future, to cover the main part of a possible calcium transient around that point), and then averages $Y_i$ across time over the union of these intervals to obtain a spatial image $<Y_i>$ (*Figure 2c*). The Pearson's correlation over space between $<Y_i>$ and $\mathbf{a}_i$ (both restricted on a small neighborhood around the centroid of $\mathbf{a}_i$) is then computed, and component $i$ is rejected if the correlation coefficient is below a threshold value $\theta_{\mathrm{sp}}$, (e.g., $\theta_{\mathrm{sp}} < 0.5$). Note that a similar test is used in the online approach of *Giovannucci et al., 2017* to accept for possible new components.

### Detecting fluorescence traces with high SNR

For a candidate component to correspond to an active neuron its trace must exhibit dynamics reminiscent of the calcium indicator's transient. A criterion for this can be obtained by requiring the average SNR of trace $\mathbf{c}_i$ over the course a transient to be above a certain threshold $\theta_{\mathrm{SNR}}$, for example $\theta_{\mathrm{SNR}} = 2$, (*Figure 2d*). The average SNR can be seen as a measure of how unlikely it is for the transients of $\mathbf{c}_i$ (after some appropriate z-scoring) to have been the result of a white noise process.

To compute the SNR of a trace, let $R = Y - AC - B$ be the residual spatiotemporal signal. We can obtain the residual signal for each component $i$, $\mathbf{r}_i$, by projecting $R$ into the spatial footprint $\mathbf{a}^i$:

$$\mathbf{r}_i = 1\|\mathbf{a}_i\|^2 R^\top \mathbf{a}_i \tag{5}$$

Then the trace $\mathbf{c}_i + \mathbf{r}_i$ corresponds to the non-denoised trace of component $i$. To calculate its SNR we first compute a type of z-score:

$$\mathbf{z}_i = \frac{\mathbf{c}_i + \mathbf{r}_i - \mathrm{Baseline}\,(\mathbf{c}_i + \mathbf{r}_i)}{\mathrm{Noise}\,(\mathbf{c}_i + \mathbf{r}_i)}. \tag{6}$$

The $\mathrm{Baseline}(\cdot)$ function determines the baseline of the trace, which can be varying in the case of long datasets exhibiting baseline trends, for example due to bleaching. The function $\mathrm{Noise}(\cdot)$ estimates the noise level of the trace. Since calcium transients around the baseline can only be positive, we estimate the noise level by restricting our attention only to the points $t_n$ where $\mathbf{c}_i + \mathbf{r}_i$ is below the baseline value, that is $t_n = \{t : \mathbf{c}_i(t) + \mathbf{r}_i(t) \leq \mathrm{Baseline}\,(\mathbf{c}_i + \mathbf{r}_i)\}$, and compute the noise level as the scale parameter of a half-normal distribution (*Figure 2b*):

$$\mathrm{Noise}\,(\mathbf{c}_i + \mathbf{r}_i) = \mathrm{std}\,([\mathbf{c}_i + \mathbf{r}_i](t_n))/\sqrt{1 - \frac{2}{\pi}}. \tag{7}$$

We then determine how likely is that the positive excursions of $\mathbf{z}_i$ can be attributed just to noise. We compute the probabilities $\mathbf{p}_i(t) = \Phi(-\mathbf{z}_i(t))$, where $\Phi(\cdot)$ denotes the cumulative distribution function of a standard normal distribution, and compute the most unlikely excursion over a window of $N_s$ timesteps that corresponds to the length of a typical transient, for example $N_s = \lceil 0.4s \times F \rceil$, where 0.4s could correspond to the typical length of a GCaMP6f transient, and $F$ is the imaging rate.

$$p^i_{\min} = \min_t \left( \prod_{j=0}^{N_s-1} \mathbf{p}_i(t+j) \right)^{1/N_s}. \tag{8}$$

The (averaged peak) SNR of component $i$ can then be defined as

$$\text{SNR}_i = \Phi^{-1}(1 - p^i_{\min}) = -\Phi^{-1}(p^i_{\min}), \tag{9}$$

where $\Phi^{-1}$ is the quantile function for the standard normal distribution (logit function) and a component is accepted if $\text{SNR}_i \geq \theta_{\text{SNR}}$. Note that for numerical stability we compute $p^i_{\min}$ in the logarithmic domain and check the condition $p^i_{\min} \leq \Phi(-\theta_{\text{SNR}})$.

We can also use a similar test for the significance of the time traces in the spike domain after performing deconvolution. In this case, traces are considered to spike if their maximum height due to a spike transient exceeds a threshold. If we assume that the shape of each calcium transient has been normalized to have maximum amplitude 1, then this corresponds to testing $\|\mathbf{s}_i\|_\infty \geq \theta_{\text{SNR}} \sigma_i$, where $\mathbf{s}_i$ represents the deconvolved activity trace for component $i$, and $\theta_{\text{SNR}}$ is again an appropriate SNR threshold, for example $\theta_{\text{SNR}} = 2$, and $\sigma_i$ is the noise level for trace $i$.

## Classification through convolutional neural networks (CNNs)

The tests described above are unsupervised but require fine-tuning of two threshold parameters $(\theta_{sp}, \theta_{\text{SNR}})$ that might be dataset dependent and might be sensitive to strong non-stationarities. As a third test we trained a 4-layer CNN to classify the spatial footprints into true or false components, where a true component here corresponds to a spatial footprint that resembles a neuron soma (See *Figure 2e* and section *Classification through convolutional networks* for details). A simple threshold $\theta_{\text{CNN}}$ can be used to tune the classifier (e.g., $\theta_{\text{CNN}} = 0.5$).

### Collection of manual annotations and consensus

We collected manual annotations from four independent labelers who were instructed to find round or donut shaped neurons of similar size using the ImageJ Cell Magic Wand tool (*Walker, 2014*). We focused on manually annotating only cells that were active within each dataset and for that reason the labelers were provided with two summary statistics: (i) A movie obtained by removing a running 20th percentile (as a crude background approximation) and downsampling in time by a factor of 10, and (ii) the max-correlation image. The correlation image (CI) at every pixel is equal to the average temporal correlation coefficient between that pixel and its neighbors (*Smith and Häusser, 2010*) (eight neighbors were used for our analysis). The max-correlation image is obtained by computing the CI for each batch of 33 s (1000 frames for a 30 Hz acquisition rate), and then taking the maximum over all these images (*Figure 3—figure supplement 1a*). Neurons that are inactive during the course of the dataset will be suppressed both from the baseline removed video (since their activity will always be around their baseline), and from the max-correlation image since the variation around this baseline will mostly be due to noise leading to practically uncorrelated neighboring pixels (*Figure 3—figure supplement 1a*). Nine different mouse in vivo datasets were used from various brain areas and labs. A description is given in *Table 2*. To create the final consensus, the labelers were asked to jointly resolve the inconsistencies between their annotations. To this end, every ROI selected by at least one but not all labelers was re-considered by a group of at least two labelers that decided whether it corresponds to an active neuron or not.

The annotation procedure provides a binary mask per selected component. On the other hand, the output of for each component is a non-negatively valued vector over the FOV (a real-valued mask). The two sets of masks differ not only in their variable type but also in their general shape: Manual annotation through the Cell Magic Wand tool tends to produce circular shapes, whereas the output of CaImAn will try to accurately estimate the shape of each active component (*Figure 3—figure supplement 1b*). To construct the consensus components that can be directly used for comparison, the binary masks from the manual annotations were used to seed the CNMF algorithm (Algorithm 3). This produced a set of real valued components with spatial footprints restricted to the areas provided by the annotations, and a corresponding set of temporal components that can be used to evaluate the performance of CaImAn (*Figure 4*). Registration was performed using the REGISTERPAIR algorithm (Algorithm 7) and match was counted as a true positive when the (modified)

Jaccard distance (*Equation 11*) was below 0.7. Details of the registration procedure are given below (see *Component registration*).

## Cross-Validation analysis of manual annotations

As mentioned in the results section, comparing each manual annotation with the consensus annotation can create slightly biased results in favor of individual annotators since the consensus annotation is chosen from the union of individual annotations. To correct for this we performed a cross-validation analysis where the annotations of each labeler were compared against an automatically generated combination of the rest of the labelers. To create the combined annotations we first used the REGISTERMULTI procedure to construct the union of each subset of $N-1$ labelers (where $N$ is the total number of labelers for each dataset). When $N = 4$ then the combined annotation consisted of the components that were selected by at least two labelers. When $N = 3$ a stricter intersection approach was used; the combined annotation consisted of the components that were selected by both remaining labelers. The procedure was repeated for all subsets of labelers and all datasets. The results are shown in *Table 3* While individual scores for specific annotators and datasets vary significantly compared to using the consensus annotation as ground truth (*Table 1*), the decrease in average performance was modest indicating a low bias level.

## Classification through convolutional neural networks (CNNs)

CAIMAN uses two CNN classifiers; one for post processing component screening in CAIMAN BATCH, and a different one for screening candidate components in CAIMAN ONLINE. In both cases a four layer CNN was used, with architecture as described in *Figure 2e*. The first two convolutional layers consist of 32 $3{\times}3$ filters each, whereas each of the latter two layers consist of 64 $3 \times 3$ filters, all followed by a rectifier linear unit (ReLU). Every two layers a $2 \times 2$ max-pool filter is included. A two layer dense network with 512 hidden units is used to compute the predictions (*Figure 2e*).

### CAIMAN BATCH classifier for post processing classification

The purpose of the batch classifier is to classify the components detected by CAIMAN BATCH into neuron somas or other shapes, by examining their spatial footprints. Only three annotated datasets (NF.03.00.t, NF.04.00.t, NF.02.00) were used to train the batch classifier. The set of estimated footprints from running CAIMAN BATCH initialized with the consensus annotation was matched to the set of consensus footprints. Footprints matched to consensus components were considered positive examples, whereas the remaining components were labeled as negatives. The two sets were enriched using data augmentation (rotations, reflections, contrast manipulation etc.) through the Keras library (keras.io) and the CNN was trained on 60% of the data, leaving 20% for validation and 20% for testing. The CNN classifier reached an accuracy of 97% on test data; this generalized to the rest of the datasets (*Figure 2e*) without any parameter change.

### Online classifier for new component detection

The purpose of the CAIMAN ONLINE classifier is to detect new components based on their spatial footprints by looking at the mean across time of the residual buffer. To construct the labeled data for the online classifier, CAIMAN BATCH was run on the first five annotated datasets seeded with the masks obtained through the manual annotations. Subsequently the activity of random subsets of found components and the background was removed from contiguous frames of the raw datasets to construct residual buffers, which were averaged across time. From the resulting images patches were extracted corresponding to positive examples (patches around a neuron that was active during the buffer) and negative examples (patches around other positions within the FOV). A neuron was considered active if its trace attained an average peak-SNR value of 4 or higher during the buffer interval. Similarly to the batch classifier, the two sets were augmented and split into training, validation and testing sets. The resulting classifier reached a 98% accuracy on the testing set, and also generalized well when applied to different datasets.

### Differences between the two classifiers

Although both classifiers examine the spatial footprints of candidate components, their required performance characteristics are different which led us to train them separately. Firstly, the two

classifiers are trained on separate data: The batch classifier is trained on spatial footprints extracted from CAIMAN BATCH, whereas the online classifier is trained on residual signals that are generated as CAIMAN ONLINE operates. The batch classifier examines each component as a post-processing step to determine whether its shape corresponds to a neural cell body. As such, false positive and false negative examples are treated equally and possible mis-classifications do not directly affect the traces of the other components. By contrast, the online classifier operates as part of the online processing pipeline. In this case, a new component that is not detected in a residual buffer is likely to be detected later should it become more active. On the other hand, a component that is falsely detected and incorporated in the online processing pipeline will continue to affect the future buffer residuals and the detection of future components. As such the online algorithm is more sensitive to false positives than false negatives. To ensure a small number of false positive examples under testing conditions, only components with average peak-SNR value at least four were considered as positive examples during training of the online classifier.

## Distributed update of spatial footprints

To efficiently distribute the cost of updating shapes across all frames we derived a simple algorithm that (i) ensures that every spatial footprint is updated at least once every $T_u$ steps, where $T_u$ is a user defined parameter, for example $T_u = 200$, and (ii) no spatial component is updated during a step when new components were added. The latter property is used to compensate for the additional computational cost that comes with introducing new components. Whenever a new component is added the algorithm collects the components with overlapping spatial footprints and makes sure they are updated at the next frame. This property ensures that the footprints of all required components adapt quickly whenever a new neighbor is introduced. The procedure is described in algorithmic form in Algorithm 6.

## Component registration

Fluorescence microscopy methods enable imaging the same brain region across different sessions that can span multiple days or weeks. While the microscope can visit the same location in the brain with reasonably high precision, the FOV might might not precisely match due to misalignments or deformations in the brain medium. CAIMAN provides routines for FOV alignment and component registration across multiple sessions/days. Let $\mathbf{a}_1^1, \mathbf{a}_2^1, \ldots, \mathbf{a}_{N_1}^1$ and $\mathbf{a}_1^2, \mathbf{a}_2^2, \ldots, \mathbf{a}_{N_2}^2$ the sets of spatial components from sessions 1 and 2 respectively, where $N_1$ and $N_2$ denote the total number of components from each session. We first compute the FOV displacement by aligning some summary images from the two sessions (e.g., mean or correlation image), using a non-rigid registration method, for example NoRMCoRRE (*Pnevmatikakis and Giovannucci, 2017*). We apply the estimated displacement field to the components of $A_1$ to align them with the FOV of session 2. To perform the registration, we construct a pairwise distance matrix $D \in \mathbb{R}^{N_1 \times N_2}$ with $D(i,j) = d(\mathbf{a}_i^1, \mathbf{a}_j^2)$, where $d(\cdot, \cdot)$ denotes a distance metric between two components. The chosen distance corresponds to the Jaccard distance between the binarized versions of the components. A real valued component $\mathbf{a}$ is converted into its binary version $m(\mathbf{x})$ by setting to one only the values of $\mathbf{a}$ that are above the maximum value of $\mathbf{a}$ times a threshold $\theta_b$, for example $\theta_b = 0.2$:

$$m(\mathbf{a})((x)) = \begin{cases} 1, & \mathbf{a}((x)) \geq \theta_b \|\mathbf{a}\|_\infty \\ 0, & \text{otherwise} \end{cases}. \tag{10}$$

To compute the distance between two binary masks $m_1, m_2$, we use the Jaccard index (intersection over union) which is defined as

$$J(m_1, m_2) = \frac{|m_1 \cap m_2|}{|m_1 \cup m_2|}, \tag{11}$$

and use it to define the distance metric as

$$d(\mathbf{a}_i^1, \mathbf{a}_j^2) = \begin{cases} 1 - J(m(\mathbf{a}_i^1), m(\mathbf{a}_j^2)) & 1 - J(m(\mathbf{a}_i^1), m(\mathbf{a}_j^2)) \leq \theta_d \\ 0 & (m(\mathbf{a}_i^1) \subseteq m(\mathbf{a}_j^2)) \text{ OR } (m(\mathbf{a}_j^2) \subseteq m(\mathbf{a}_i^1)), \\ \infty & \text{otherwise.} \end{cases} \tag{12}$$

where $\theta_d$ is a distance threshold, for example 0.5 above which two components are considered non-matching and their distance is set to infinity. This is done to prevent false matchings between only marginally overlapping components.

After the distance matrix $D$ has been completed, an optimal matching between the components of the two sessions is computed using the Hungarian algorithm to solve the linear assignment problem. As infinite distances are allowed, it is possible to have components from both sessions that are not matched with any other component, preventing false assignments and enabling the registration of sessions with different number of neurons. Moreover, the use of infinite distances speeds up the Hungarian algorithm as it significantly restricts the space of possible pairings. This process of registering components across two sessions (REGISTERPAIR) is summarized in Algorithm 7.

To register components across multiple sessions, we first order the sessions chronologically and register session 1 against session 2. From this registration we construct the union of the distinct components between the two sessions by keeping the matched components from session two as well as the non-matched components from both sessions aligned to the FOV of session 2. We then register this union of components to the components of session three and repeat the procedure until all sessions are have been registered. This process of multi session registration (REGISTERMULTI) is summarized in Algorithm 8. At the end of the process the algorithm produces a list of matches between the components of each session and the union of all active distinct components, allowing for efficient tracking of components across multiple days (*Figure 9*), and the comparison of non-consecutive sessions through the union without the need of direct pairwise registration (*Figure 9—figure supplement 1*). An alternative approach to the problem of multiple session registration (CellReg) was presented recently by *Sheintuch et al. (2017)* where the authors register neurons across multiple days by first constructing a similar union set of all the components which is then refined using a clustering procedure. REGISTERMULTI differs from the CellReg method of *Sheintuch et al. (2017)* along the following dimensions:

- REGISTERMULTI uses a simple intersection over union metric to estimate the distance between two neighboring neurons after the FOV alignment. Cells that have a distance above a given threshold are considered distinct by default and are not tested for matching. This parameter has an intuitive interpretation and can be set a-priori for each dataset. By contrast, CELLREG uses a probabilistic framework based on the joint probability distribution between the distance of two cells and the correlation of their shapes. Such choice makes specific parametric assumptions about the distributions of centroid distances between the same and different cells, as well as their shape correlations. This model must be re-evaluated for every different set of sessions to be registered and can require considerable data to learn the appropriate distance metric.
- REGISTERMULTI uses the Hungarian algorithm to register two different set of components, solving the linear assignment problem optimally under the assumed distance function. In contrast CELLREG uses a greedy method for initializing the assignment of cells to the union superset relying on the following clustering step to refine these estimates, adding extra computational burden to the registration procedure.

## Implementation details for CAIMAN BATCH

Each dataset was processed using the same set of parameters, apart from the expected size of neurons (estimated by inspecting the correlation image), the size of patches and expected number of neurons per patch (estimated by inspecting the correlation image). For the dataset N.01.01, where optical modulation was induced, the threshold for merging neurons was slightly higher (the stimulation caused clustered synchronous activity). For shorter datasets, rigid motion correction was sufficient; for longer datasets K53, J115 we applied non-rigid motion correction. Parameters for the automatic selection of components were optimized using a grid search approach.

The global default parameters for all datasets were obtained by performing a grid search on the nine datasets over the following values: trace peak SNR threshold on the set $\{1.75, 2, 2.25, 2.5\}$, spatial correlation threshold on the set $\{0.75, 0.8, 0.85\}$, lower threshold on CNN classifier (reject if prediction is below a certain value) on the set $\{0.05, 0.1, 0.15\}$, and upper threshold on classifier (accept if prediction is above a certain value) on the set $\{0.9, .95, 0.99, 1\}$. The best overall parameters (used for the results reported in *Table 1*) were given for the choice $(2, 0.85, 0.1, 0.99)$. For all datasets the background neuropil activity was modeled as a rank two matrix, and calcium dynamics were

modeled as a first order autoregressive process. The remaining parameters were optimized so that all the datasets could be run on a machine with less than 128 GB RAM.

## Implementation details for CᴀʟᴍAɴ ᴏɴʟɪɴᴇ

Datasets were processed for two epochs with the exception of the longer datasets `J115, K53` where only one pass of the data was performed to limit computational cost. For all datasets the background neuropil activity was modeled as a rank two matrix, and calcium dynamics were modeled as a first order autoregressive process. For each dataset, CᴀʟᴍAɴ ᴏɴʟɪɴᴇ was initialized on the first 200 frames, using the BᴀʀᴇIɴɪᴛɪᴀʟɪᴢᴀᴛɪᴏɴ on the entire FOV with only two neurons, so in practice all the neurons were detected during the online mode. We did not post-process the results (which could have further enhanced performance) in order to demonstrate performance levels with fully online practices. As in batch processing, the expected size of neurons was chosen separately for each dataset after inspecting the correlation image. Several datasets (`N.03.00.t, N.02.00, J123, J115, K53`) were spatially decimated by a factor of 2 to enhance processing speed, a step that did not lead to changes in detection performance.

To select global parameters for all datasets we performed a grid search on all nine datasets by varying the following parameters: The peak SNR threshold for accepting a candidate component on the set $\{0.6, 0.8, 1, 1.2, 1.4, 1.6, 1.8, 2\}$, the online CNN classifier threshold for accepting candidate components on the set $\{0.5, 0.55, 0.6, 0.65, 0.7, 0.75\}$, and the number of candidate components per frame on the set $\{5, 7, 10, 14\}$. The best overall parameters (reported in *Table 1*) were given for the choice $(1.2, 0.65, 10)$. This parameter choice was also the best when conditioning on the shorter six datasets. For the three longer datasets, the best parameter choice was $(2, 0.6, 5)$, corresponding to a stricter set of parameters with less candidate components per frame (*Figure 4—figure supplement 1*).

For the analysis of the whole brain zebrafish dataset, CᴀʟᴍAɴ ᴏɴʟɪɴᴇ was run for one epoch with the same parameters as above, with only differences appearing in the number of neurons during initialization (600 vs 2), and the value of the threshold for the online CNN classifier (0.75 vs 0.5). The former decision was motivated by the goal of retrieving with a single pass neurons from a preparation with a denser level of activity over a larger FOV in this short dataset (1885 frames). To this end, the number of candidate neurons at each timestep was set to 10 (per plane). The threshold choice was motivated by the fact that the classifier was trained on mouse data only, and thus a higher threshold choice would help diminish potential false positive components. Rigid motion correction was applied online to each plane.

## Comparison with Suite2p

To compare CᴀʟᴍAɴ with Suite2p we used the MATLAB version of the Suite2p package (*Pachitariu et al., 2017*). To select parameters for Suite2p we used a small grid search around the default values for the variables nSVDforROI, NavgFramesSVD, and sig. The classifier used by Suite2p was not re-trained for each dataset but used with the default values. For each case (with the classifier and without the classifier), the values that give the best $F_1$ score in average are reported in *Figure 4—figure supplement 2*. The dataset `J123` was excluded from the comparison since (due its low SNR) Suite2p did not converge and kept adding a large number of neurons in each iteration. Use of the classifier improved the results on average, from $F_1$ score $0.51 \pm 0.12$ without the classifier to $0.55 \pm 0.12$, however the use of the classifier improved only four of the eight tested datasets in terms of the $F_1$ score. As with CᴀʟᴍAɴ it is possible that dataset specific parameter choice can lead to improved results.

## Performance quantification as a function of SNR

To quantify performance as a function of SNR we construct the consensus traces by running CᴀʟᴍAɴ ʙᴀᴛᴄʜ on the datasets seeded with the 'consensus' binary masks obtained from the manual annotators. Afterwards we obtain the average peak-SNR of a trace **c** with corresponding residual signal **r** (*Equation 5)* is obtained as

$$\mathrm{SNR}(\mathbf{z}) = -\Phi^{-1}(p_{\min}), \tag{13}$$

where $\Phi^{-1}(\cdot)$ denotes the probit function (quantile function for the standard Gaussian distribution), $\mathbf{z}$ is the z-scored version of $\mathbf{c} + \mathbf{r}$ (*Equation 6*) and $p_{\min}$ is given by *Equation 8*.

Let $c_1^{\mathrm{gt}}, c_2^{\mathrm{gt}}, \ldots, c_N^{\mathrm{gt}}$ be the consensus traces and $c_1^{\mathrm{cm}}, c_2^{\mathrm{cm}}, \ldots, c_N^{\mathrm{cm}}$ be their corresponding CAIMAN inferred traces. Here we assume that false positive and false negative components are matched with trivial components that have 0 SNR. Let also $\mathrm{SNR}^{\mathrm{gt}}{}_i = \mathrm{SNR}(c_i^{\mathrm{gt}})$ and $\mathrm{SNR}^{\mathrm{cm}}{}_i = \mathrm{SNR}(c_i^{\mathrm{cm}})$, respectively. After we compute the SNR for both consensus and inferred traces the performance algorithm can be quantified in multiple ways as a function of a SNR thresholds $\theta_{\mathrm{SNR}}$:

**Precision**: Precision at level $\theta_{\mathrm{SNR}}$, can be computed as the fraction of detected components with $\mathrm{SNR}^{\mathrm{cm}} > \theta\,\mathrm{SNR}$ that are matched with consensus components. It quantifies the certainty that a component detected with a given SNR or above corresponds to a true component.

$$\mathrm{PREC}(\theta_{\mathrm{SNR}}) = \frac{|\{i : (\mathrm{SNR}^{\mathrm{cm}}{}_i > \theta_{\mathrm{SNR}}) \ \& \ (\mathrm{SNR}^{\mathrm{gt}}{}_i > 0)\}|}{|\{i : (\mathrm{SNR}^{\mathrm{cm}}{}_i > \theta_{\mathrm{SNR}})\}|}$$

**Recall**: Recall at level $\theta_{\mathrm{SNR}}$, can be computed as the fraction of consensus components with $\mathrm{SNR}^{\mathrm{gt}} > \theta\,\mathrm{SNR}$ that are detected by the algorithm. It quantifies the certainty that a consensus component with a given SNR or above is detected.

$$\mathrm{RECALL}(\theta_{\mathrm{SNR}}) = \frac{|\{i : (\mathrm{SNR}^{\mathrm{gt}}{}_i > \theta_{\mathrm{SNR}}) \ \& \ (\mathrm{SNR}^{\mathrm{cm}}{}_i > 0)\}|}{|\{i : (\mathrm{SNR}^{\mathrm{gt}}{}_i > \theta_{\mathrm{SNR}})\}|}$$

$F_1$ **Score**: An overall $F_1$ score at level $\theta_{\mathrm{SNR}}$, can be obtained by computing the harmonic mean between precision and recall

$$\mathrm{F}_1(\theta_{\mathrm{SNR}}) = 2 \frac{\mathrm{PREC}(\theta_{\mathrm{SNR}}) \times \mathrm{RECALL}(\theta_{\mathrm{SNR}})}{\mathrm{PREC}(\theta_{\mathrm{SNR}}) + \mathrm{RECALL}(\theta_{\mathrm{SNR}})}$$

The cautious reader will observe that the precision and recall quantities described above are not computed in the same set of components. This can be remedied by recomputing the quantities in two different ways:

**AND framework**: Here we consider a match only if *both* traces have SNR above the given threshold:

$$\mathrm{PREC}_{\mathrm{AND}}(\theta_{\mathrm{SNR}}) = \frac{|\{i : (\mathrm{SNR}^{\mathrm{cm}}{}_i > \theta_{\mathrm{SNR}}) \, \& \, (\mathrm{SNR}^{\mathrm{gt}}{}_i > \theta_{\mathrm{SNR}})\}|}{|\{i : (\mathrm{SNR}^{\mathrm{cm}}{}_i > \theta_{\mathrm{SNR}})\}|}$$

$$\mathrm{RECALL}_{\mathrm{AND}}(\theta_{\mathrm{SNR}}) = \frac{|\{i : (\mathrm{SNR}^{\mathrm{gt}}{}_i > \theta_{\mathrm{SNR}}) \, \& \, (\mathrm{SNR}^{\mathrm{cm}}{}_i > \theta_{\mathrm{SNR}})\}|}{|\{i : (\mathrm{SNR}^{\mathrm{gt}}{}_i > \theta_{\mathrm{SNR}})\}|}$$

**OR framework**: Here we consider a match if *either* trace has SNR above the given threshold and its match has SNR above 0.

$$\mathrm{RECALL}_{\mathrm{OR}}(\theta_{\mathrm{SNR}}) = \frac{|\{i : (\max(\mathrm{SNR}^{\mathrm{gt}}{}_i, \mathrm{SNR}^{\mathrm{cm}}{}_i) > \theta_{\mathrm{SNR}}) \, \& \, (\min(\mathrm{SNR}^{\mathrm{gt}}{}_i, \mathrm{SNR}^{\mathrm{cm}}{}_i) > 0)\}|}{|\{i : (\mathrm{SNR}^{\mathrm{cm}}{}_i > 0)\}|}$$

$$\mathrm{RECALL}_{\mathrm{OR}}(\theta_{\mathrm{SNR}}) = \frac{|\{i : (\max(\mathrm{SNR}^{\mathrm{gt}}{}_i, \mathrm{SNR}^{\mathrm{cm}}{}_i) > \theta_{\mathrm{SNR}}) \, \& \, (\min(\mathrm{SNR}^{\mathrm{gt}}{}_i, \mathrm{SNR}^{\mathrm{cm}}{}_i) > 0)\}|}{|\{i : (\mathrm{SNR}^{\mathrm{gt}}{}_i > 0)\}|}$$

It is easy to show that

$$\mathrm{PREC}_{\mathrm{AND}}(\theta_{\mathrm{SNR}}) \leq \mathrm{PREC}(\theta_{\mathrm{SNR}}) \leq \mathrm{PREC}_{\mathrm{OR}}(\theta_{\mathrm{SNR}})$$
$$\mathrm{RECALL}_{\mathrm{AND}}(\theta_{\mathrm{SNR}}) \leq \mathrm{RECALL}(\theta_{\mathrm{SNR}}) \leq \mathrm{RECALL}_{\mathrm{OR}}(\theta_{\mathrm{SNR}})$$
$$\mathrm{F}_{1\,\mathrm{AND}}(\theta_{\mathrm{SNR}}) \leq \mathrm{F}_1(\theta_{\mathrm{SNR}}) \leq \mathrm{F}_{1\,\mathrm{OR}}(\theta_{\mathrm{SNR}}),$$

with equality holding for $\theta_{\mathrm{SNR}} = 0$. As demonstrated in *Figure 4d*, these bounds are tight.

## Additional features of CAIMAN

CAIMAN contains a number of additional features that are not presented in the results section for reasons of brevity. These include:

## Volumetric data processing

Apart from planar 2D data, CaImAn batch is also applicable to 3D volumetric data collected via dense raster scanning methods or from direct volume imaging methods such as light field microscopy (*Prevedel et al., 2014*; *Grosenick et al., 2017*).

## Segmentation of structural indicator data

Structural indicators expressed in the nucleus and functional indicators expressed in the cytoplasm can facilitate source extraction and help identify silent or specific subpopulations of neurons (e.g., inhibitory). CaImAn provides a simple adaptive thresholding filtering method for segmenting summary images of the structural channel (e.g., mean image). The obtained results can be used for seeding source extraction from the functional channel in CaImAn batch or CaImAn online as already discussed.

## Duplicate detection

The annotations obtained through the consensus process were screened for possible duplicate selections. To detect for duplicate components we define the degree of spatial overlap matrix $O$ as

$$O_{ij} = \begin{cases} 0, & i=j \\ \frac{|m(\mathbf{a}^i) \cap m(\mathbf{a}^j)|}{|m(\mathbf{a}^i)|}, & i \neq j \end{cases}, \tag{14}$$

that defines the fraction of component $i$ that overlap with component $j$, where $m(\cdot)$ is the thresholding function defined in *Equation 10*. Any entry of $O$ that is above a threshold $\theta_o$ (e.g., $\theta_o = 0.7$ used here) indicates a pair of duplicate components. To decide which of the two components should be removed, we use predictions of the CaImAn batch CNN classifier, removing the component with the lowest score.

## Extraction of $\Delta F/F$

The fluorescence trace $\mathbf{f}_i$ of component $i$ can be written as

$$\mathbf{f}_i = \|\mathbf{a}_i\|^2 (\mathbf{c}_i + \mathbf{r}_i). \tag{15}$$

The fluorescence due to the component's transients overlaps with a background fluorescence due to baseline fluorescence of the component and neuropil activity, that can be expressed as

$$\mathbf{f}_{0,i} = \text{Baseline}\,(\mathbf{f}_i + B^\top \mathbf{a}_i), \tag{16}$$

where $\text{Baseline} : \mathbb{R}^T \mapsto \mathbb{R}^T$ is a baseline extraction function, and $B$ is the estimated background signal. Examples of the baseline extraction function are a percentile function (e.g., 10th percentile), or a for longer traces, a running percentile function, for example 10th percentile over a window of a hundred seconds (computing the exact running percentile function can be computationally intensive. To reduce the complexity we compute the running percentile with a stride of $W$, where $W$ is equal or smaller to the length of the window, and then linearly interpolate the values). To determine the optimal percentile level an empirical histogram of the trace (or parts of it in case of long traces) is computed using a diffusion kernel density estimator (*Botev et al., 2010*), and the mode of this density is used to define the baseline and its corresponding percentile level. The $\Delta F/F$ activity of component $i$ can then be written as

$$\mathbf{c}_i^{\Delta F/F} = \frac{\mathbf{f}_i - \text{Baseline}\,(\mathbf{f}_i)}{\mathbf{f}_{0,i}} \tag{17}$$

The approach we propose here is conceptually similar to practical approaches where the $\Delta F/F$ is computed by averaging over the spatial extent of an ROI (*Jia et al., 2011*) with some differences: (i) instead of averaging with a binary mask we use the a weighed average with the shape of each component, (ii) signal due to overlapping components is removed from the calculation of the background fluorescence, and (iii) the traces have been extracted through the CNMF process prior to the $\Delta F/F$ extraction. Note that the same approach can also be performed to the trace $\|\mathbf{a}_i\|^2 \mathbf{c}_i$ that does not include the residual traces for each component. In practice it can be beneficial to extract $\Delta F/F$ traces prior to deconvolution, since the $\Delta F/F$ transformation can alleviate the effects of drifting

baselines, for example due to bleaching. For the non-deconvolved traces $\mathbf{f}_i$ some temporal smoothing can also be applied to obtain more smooth $\Delta F/F$ traces.

## Algorithmic details

In the following we present in pseudocode form several of the routines introduced and used by CaImAn. Note that the pseudocode descriptions do not aim to present a complete picture and may refer to other work for some of the steps.

**Algorithm 1: ProcessInPatches**

**Require:** Input data matrix $Y$, patch size, overlap size, initialization method, rest of parameters.

| | | |
|---|---|---|
| 1: | $Y^{(1)}, \ldots, Y^{(N_p)} = \text{ConstructPatches}(Y, p_s, o_s)$ | Break data into memory mapped patches. |
| 2: | **for** $i = 1, \ldots, N_p$ **do** | Process each patch |
| 3: | $[A^{(i)}, C^{(i)}, \mathbf{b}^{(i)}, \mathbf{f}^{(i)}] = \text{Cnmf}(Y^{(i)}, \text{options})$ | Run CNMF on each patch |
| 4: | **end for** | |
| 5: | $[A, C] = \text{MergeComponents}[\{A^{(i)}, C^{(i)}\}_{i=1,\ldots,N}]$ | Merge components |
| 6: | $[\mathbf{b}, \mathbf{f}] = \text{MergeBackgrounds}[\{\mathbf{b}^{(i)}, \mathbf{f}^{(i)}\}_{i=1,\ldots,N}]$ | Merge background components |
| 7: | $M \leftarrow (A > 0)$. | Find masks of spatial footprints. |
| 8: | **repeat** | Optionally keep updating $A, C, \mathbf{b}, \mathbf{f}$ using HALS (*Cichocki et al., 2007*). |
| 9: | $\quad [\mathbf{b}, \mathbf{f}] \leftarrow \text{Nnmf}(Y - AC, n_b)$ | |
| 10: | $\quad C \leftarrow \arg\min_{C \geq 0} \|Y - \mathbf{b}\mathbf{f} - AC\|$ | |
| 11: | $\quad A \leftarrow \arg\min_{A \geq 0, A(\sim M) == 0} \|Y - \mathbf{b}\mathbf{f} - AC\|$ | |
| 12: | **until** Convergence | |
| 13: | **return** $A, C, \mathbf{b}, \mathbf{f}$ | |

**Algorithm 2 CaImAn batch**

**Require:** Input data matrix $Y$, rest of parameters.

| | | |
|---|---|---|
| 1: | $Y \leftarrow \text{NormCorre}(Y, \text{params})$ | Motion Correction (*Pnevmatikakis and Giovannucci, 2017*) |
| 2: | $A, C, \mathbf{b}, \mathbf{f} = \text{ProcessInPatches}(Y, \text{params})$ | Run CNMF in patches Algorithm 1 |
| 3: | $J \leftarrow \text{EatimateQuality}(Y, A, C, \mathbf{b}, \mathbf{f}, \text{params})$ | Get indeces of accepted components |
| 4: | $A \leftarrow A[:, J], \quad C \leftarrow C[J, :]$ | Disregard rejected components |
| 5: | $[\mathbf{b}, \mathbf{f}] \leftarrow \text{Nnmf}(Y - AC, n_b)$ | |
| 6: | $C \leftarrow \arg\min_{C \geq 0} \|Y - \mathbf{b}\mathbf{f} - AC\|$ | |
| 7: | $A \leftarrow \arg\min_{A \geq 0, A(\sim M) == 0} \|Y - \mathbf{b}\mathbf{f} - AC\|$ | Refit (optional) |
| 8: | **return** $A, C, \mathbf{b}, \mathbf{f}$ | |

**Algorithm 3: SeededInitialization**

**Require:** Input data matrix $Y$, matrix of binary masks $M$, number of background components $n_b$.

| | | |
|---|---|---|
| 1: | $\mathbf{p} = \text{find}(M\mathbf{1} == 0)$ | Find the pixels not covered by any component. |
| 2: | $[\sim, \mathbf{f}] \leftarrow \text{Nnmf}(Y[\mathbf{p}, :], n_b)$ | Run NMF on these pixels just to get temporal backgrounds $\mathbf{f}$ |
| 3: | $\mathbf{b} \leftarrow \arg\min_{\mathbf{b} \geq 0} \|Y - \mathbf{b}\mathbf{f}\|$ | Obtain spatial background $\mathbf{b}$. |
| 4: | $C \leftarrow \max\left((M^\top M)^{-1} M^\top (Y - \mathbf{b}\mathbf{f}), 0\right)$ | Initialize temporal traces. |
| 5: | $A \leftarrow \arg\min_{A \geq 0, A(\sim M) == 0} \|Y - \mathbf{b}\mathbf{f} - AC\|$. | Initialize spatial footprints constrained within the masks. |
| 6: | **repeat** | Optionally keep updating $A, C, \mathbf{b}, \mathbf{f}$ using HALS |
| 7: | $\quad [\mathbf{b}, \mathbf{f}] \leftarrow \text{Nnmf}(Y - AC, n_b)$ | |
| 8: | $\quad C \leftarrow \arg\min_{C \geq 0} \|Y - \mathbf{b}\mathbf{f} - AC\|$ | |

*Continued on next page*

*Continued*

**Algorithm 3: SEEDEDINITIALIZATION**

| | |
|---|---|
| 9: | $A \leftarrow \arg\min_{A \geq 0, A(\sim M)==0} \|Y - \mathbf{b}\mathbf{f} - AC\|$ |
| 10: | **until** Convergence |
| 11: | **return** $A, C, \mathbf{b}, \mathbf{f}$ |

**Algorithm 4 CAIMAN ONLINE (See *Giovannucci et al., 2017* for explanation of routines)**

**Require:** Data matrix $Y$, initial estimates $A, \mathbf{b}, C, \mathbf{f}, S$, current number of components $K$, current timestep $t'$, rest of parameters.

| | | |
|---|---|---|
| 1: | $W = Y[:, 1 : t']C^\top / t'$ | |
| 2: | $M = CC^\top / t'$ | Initialize sufficient statistics (*Giovannucci et al., 2017*) |
| 3: | $\mathcal{G} = \text{DETERMINEGROUPS}([A, \mathbf{b}], K)$ | (*Giovannucci et al., 2017*, Algorithm S1-S2) |
| 4: | $R_{buf} = [Y - [A, \mathbf{b}][C; \mathbf{f}]][:, t' - l_b + 1 : t']$ | Initialize residual buffer |
| 5: | $t = t'$ | |
| 6: | **for** $i = 1, \ldots, N_{\text{epochs}}$ **do** | |
| 7: | **While** there is more data **do** | |
| 8: | $t \leftarrow t + 1$ | |
| 9: | $\mathbf{y}_t \leftarrow \text{MOTIONCORRECT}(\mathbf{y}_t, \mathbf{b}\mathbf{f}_{t-1})$ | (*Pnevmatikakis and Giovannucci, 2017*) |
| 10: | $[\mathbf{c}_t; \mathbf{f}_t] \leftarrow \text{UPDATETRACES}([A, \mathbf{b}], [\mathbf{c}_{t-1}; \mathbf{f}_{t-1}], \mathbf{y}_t, \mathcal{G})$ | (*Giovannucci et al., 2017*, Algorithm S3) |
| 11: | $C, S \leftarrow \text{OASIS}(C, \gamma, s_{min}, \lambda)$ | (*Friedrich et al., 2017b*) |
| 12: | $A_{\text{new}}, C_{\text{new}} \leftarrow \text{FINDNEWCOMPONENTS}(R_{buf}, N_{comp})$ | Algorithm 5 |
| 13: | $[A, \mathbf{b}], [C, \mathbf{f}], K, \mathcal{G}, R_{buf}, W, M \leftarrow \text{INTEGRATENEWCOMPONENTS}($ | |
| 14: | $[A, \mathbf{b}], [C, \mathbf{f}], K, \mathcal{G}, A_{\text{new}}, C_{\text{new}}, R_{buf}, \mathbf{y}_t, W, M)$ | (*Giovannucci et al., 2017*, Algorithm S4) |
| 15: | $R_{buf} \leftarrow [R_{buf}[:, 2 : l_b], \mathbf{y}_t - A\mathbf{c}_t - \mathbf{b}\mathbf{f}_t]$ | Update residual buffer |
| 16: | $W, M \leftarrow \text{UPDATESUFFSTATISTICS}(W, M, \mathbf{y}_t, [\mathbf{c}_t; \mathbf{f}_t])$ | |
| 17: | $I_u \leftarrow \text{SHAPEUPDATEINDECES}(A, I_{\text{new}})$ | Indeces of components to get updated, Algorithm S6 |
| 18: | $[A, \mathbf{b}] \leftarrow \text{UPDATESHAPES}[W, M, [A, \mathbf{b}], I_u]$ | (*Giovannucci et al., 2017*, Algorithm S5) |
| 19: | **end while** | |
| 20: | $t \leftarrow 0$ | |
| 21: | **end for** | |
| 22: | **return** $A, \mathbf{b}, C, \mathbf{f}, S$ | |

**Algorithm 5: FINDNEWCOMPONENTS**

**Require:** Residual buffer $R_{buf}$, number of new candidate components $N_{comp}$, neuron radius $r$.

| | | |
|---|---|---|
| 1: | $E \leftarrow \sum_t \max(R_{\text{buf}(t)}, 0)^2$ | |
| 2: | $E \leftarrow \text{HIGHPASSFILTER}(E)$ | Spatial high pass filtering for contrast enhancement. |
| 3: | $P = \text{FINDLOCALPEAKS}(E, N_{\text{comp}}, r)$ | Find local maxima at least $2r$ apart. |
| 4: | $A_{\text{test}} \leftarrow \emptyset$ | |
| 5: | **for** $p \in P$ **do** | |
| 6: | $N_p = \{(x, y) : |x - p_x| \leq r, |y - p_y| \leq r\}$ | Define a neighborhood around $p$ |
| 7: | $A_{\text{test}} \leftarrow A_{\text{test}} \cup \text{MEAN}(R_{\text{buf}})$ | |
| 8: | **end for** | |
| 9: | $I_{\text{accept}} \leftarrow \text{ONLINECNNCCLASSIFIER}(A_{\text{test}})$ | Find indeces of accepted components |
| 10: | $A_{\text{new}} \leftarrow \emptyset, C_{\text{new}} \leftarrow \emptyset$ | |
| 11; | **for** $i \in I_{\text{accept}}$ **do** | |

*Continued on next page*

*Continued*

**Algorithm 5:** FINDNEWCOMPONENTS

| | |
|---|---|
| 12: | $[\mathbf{a}, \mathbf{c}] \leftarrow \text{NNMF}(R_{\text{buf}}[N_{p^i}, :], 1)$ |
| 13: | $A_{\text{new}} \leftarrow A_{\text{new}} \cup \mathbf{a}$ |
| 14: | $C_{\text{new}} \leftarrow C_{\text{new}} \cup \mathbf{c}$ |
| 15: | **end for** |
| 16: | **return** $A_{\text{new}}, C_{\text{new}}$ |

**Algorithm 6:** SHAPEUPDATEINDECES

**Require:** Set of spatial footprints $A$, indeces of newly added components $J$, update vector $\mathbf{q}$, update period $T_u$, current step in online mode $t$.

| | | |
|---|---|---|
| 1: | **if** t = 0 **then** | Initialize vector at the beginning of online mode. |
| 2: | $q \leftarrow 2^{[1,2,\dots,|A|]/|A|}$ | Values logarithmically spaced between 1 and 2. |
| 3: | **end if** | |
| 4: | $\mathbf{q} \leftarrow \mathbf{q} \times 0.5^{1/T_u}$ | |
| 5: | **if** $J = \emptyset$ **then** | |
| 6: | $I_u \leftarrow \{i : q_i \leq 1\}$ | Indeces of components to get updated. |
| 7: | $\mathbf{q}(I_u) \leftarrow \mathbf{q}(I_u) + 1$ | |
| 8: | **else** | Do not update shapes if new components are added. |
| 9: | $I_o = \emptyset$ | |
| 10: | **for** $j \in J$ **do** | |
| 11: | $I_o \leftarrow I_o \cup \{i : A[:, i]^\top A[:, j] > 0\}$ | Find overlapping components. |
| 12: | **end for** | |
| 13: | $\mathbf{q}(I_o) \leftarrow 0$ | Make sure these components get updated at the next step. |
| 14: | $I_u \leftarrow \emptyset$ | |
| 15: | **end if** | |
| 16: | **return** Indeces of components to get updated $I_u$, update counter vector $\mathbf{q}$. | |

**Algorithm 7:** REGISTERPAIR

**Require:** Spatial footprint matrices $A_1, A_2$, field of view templates $I_1, I_2$, thresholds for binarization $\theta_b$ and matching $\theta_m$.

| | | |
|---|---|---|
| 1: | $S = \text{COMPUTEMOTIONFIELD}(I_1, I_2)$ | Compute motion field between the templates. |
| 2: | $A_1 \leftarrow \text{APPLYMOTIONFIELD}(A_1, S)$ | Align $A_1$ to the template $I_2$ |
| 3: | $[M_1, M_2] = \text{BINARIZE}([A_1, A_2], \theta_b)$ | Turn components into binary masks. |
| 4: | $D = \text{COMPUTEDISTANCEMATRIX}(M_1, M_2, \theta_D)$ | Compute distance matrix. |
| 5: | $P_1, P_2, L_1, L_2 = \text{HUNGARIAN}(D)$ | Match using the Hungarian algorithm. |
| 6: | **return** Matched components $P_1, P_2$, non-matched components $L_1, L_2$ and aligned components from first session $A_1$. | |

**Algorithm 8:** REGISTERMULTI

**Require:** List of Spatial footprint matrices $A_1, A_2, \dots, A_N$, field of view templates $I_1, I_2, \dots, I_N$, thresholds for binarization $\theta_b$ and matching $\theta_m$.

| | | |
|---|---|---|
| 1: | **for** $i = 1, \dots, N$ **do** | |
| 2: | $K_i = \text{SIZE}(A_i, 2)$ | Number of components in each session. |
| 3: | **end for** | |
| 4: | $A_u \leftarrow A_1$ | Initialize $A_u$ matrix |
| 5: | $m[1] = [1, 2, \dots, K_1]$ | Initialize matchings list |

*Continued on next page*

*Continued*

**Algorithm 8: REGISTERMULTI**

| | | |
|---|---|---|
| 6: | $K_{tot} \leftarrow K_1$ | Total # of distinct components so far. |
| 7: | **for** $i = 2, \ldots, N$ **do** | |
| 8: | $P_u, P_i, L_u, L_i, A_u = \text{REGISTERPAIR}(A_u, A_i, I_{i-1}, I_i, \theta_b, \theta_m)$ | Register $A_u$ to session $i$. |
| 9: | $A_u[:, P_u] \leftarrow A_i[:, P_i]$ | Keep the matched components from session $i$. |
| 10: | $A_u \leftarrow [A_u, A_i[:, L_i]]$ | Include the non-matched components from session $i$. |
| 11: | $m[i][P_i] = P_u$ | $m[i][j] = k$ if component $j$ from session $i$ is mapped to component $k$ in Optionally keep updating $A_u$. |
| 12: | $m[i][L_i] = [K_{tot} + 1, K_{tot} + 2, \ldots, K_{tot} + |L_i|]$ | Include newly added components. |
| 13: | $K_{tot} \leftarrow K_{tot} + |L_i|$ | Update total number of distinct components. |
| 14: | **end for** | |
| 15: | **return** Union of all distinct components $A_u$, and list of matchings $m$. | |

## Acknowledgements

We thank B Cohen, L Myers, N Roumelioti, and S Villani for providing us with manual annotations. We thank V Staneva and B Deverett for contributing to the early stages of CAIMAN, M Schachter for his insight and contributions, and L Paninski for numerous useful discussions. We thank N Carriero, I Fisk, and D Simon from the Flatiron Institute (Simons Foundation) for useful discussions and suggestions to optimize High Performance Computing code. We thank T Kawashima and M Ahrens for sharing the whole brain zebrafish dataset. Last but not least, we thank the active community of users for their great help in terms of code/method contributions, bug reporting, code testing and suggestions that have led to the growth of into a widely used open source package. A partial list of contributors (in the form of GitHub usernames) can be found in https://github.com/flatironinstitute/CalmAn/graphs/contributors (Python) and https://github.com/flatironinstitute/CalmAn-MATLAB/graphs/contributors (MATLAB). The authors acknowledge support from following funding sources: AG, EAP, JF, PG (Simons Foundation, internal funding). JG, SAK, DWT (NIH NRSA F32NS077840-01,5U01NS090541, 1U19NS104648 and Simons Foundation SCGB), PZ (NIH NIBIB R01EB022913, NSF NeuroNex DBI-1707398, Gatsby Foundation), JT (NIH R01-MH101198), FN (MURI, Simons Collaboration on the Global Brain and Pew Foundation).

## Additional information

### Funding

| Funder | Grant reference number | Author |
|---|---|---|
| National Institutes of Health | F32NS077840-01 | Jeffrey L Gauthier |
| Simons Foundation | FI-CCB | Andrea Giovannucci<br>Johannes Friedrich<br>Pat Gunn<br>Jeremie Kalfon<br>Dmitri B Chklovskii<br>Eftychios A Pnevmatikakis |
| Simons Foundation | SCGB | Sue Ann Koay<br>Farzaneh Najafi<br>Jeffrey L Gauthier<br>David W Tank |
| National Institutes of Health | 5U01NS090541 | Sue Ann Koay<br>Jeffrey L Gauthier<br>David W Tank |

| National Institutes of Health | 1U19NS104648 | Sue Ann Koay<br>Jeffrey L Gauthier<br>David W Tank |
| National Institutes of Health | NIBIB R01EB022913 | Pengcheng Zhou |
| National Science Foundation | NeuroNex DBI-1707398 | Pengcheng Zhou |
| Gatsby Charitable Foundation | | Pengcheng Zhou |
| National Institutes of Health | R01-MH101198 | Jiannis Taxidis |
| Pew Charitable Trusts | | Farzaneh Najafi |

The funders had no role in study design, data collection and interpretation, or the decision to submit the work for publication.

## Author contributions

Andrea Giovannucci, Conceptualization, Data curation, Software, Formal analysis, Supervision, Validation, Investigation, Visualization, Methodology, Writing—review and editing; Johannes Friedrich, Conceptualization, Software, Formal analysis, Validation, Investigation, Visualization, Methodology, Writing—review and editing; Pat Gunn, Software, Validation, Writing—review and editing; Jérémie Kalfon, Software, Visualization; Brandon L Brown, Software; Sue Ann Koay, Data curation, Formal analysis, Validation, Writing—review and editing; Jiannis Taxidis, Data curation, Software; Farzaneh Najafi, Software, Validation; Jeffrey L Gauthier, Conceptualization, Data curation; Pengcheng Zhou, Software, Validation, Methodology, Writing—review and editing; Baljit S Khakh, Resources, Funding acquisition; David W Tank, Resources, Data curation; Dmitri B Chklovskii, Conceptualization, Resources, Data curation, Software, Formal analysis, Supervision, Project administration, Writing—review and editing; Eftychios A Pnevmatikakis, Conceptualization, Data curation, Software, Formal analysis, Supervision, Validation, Visualization, Methodology, Writing—original draft

## Author ORCIDs

Andrea Giovannucci [ID] http://orcid.org/0000-0002-7850-444X
Johannes Friedrich [ID] https://orcid.org/0000-0002-1321-5866
Jérémie Kalfon [ID] https://orcid.org/0000-0002-2818-9728
Pengcheng Zhou [ID] http://orcid.org/0000-0003-1237-3931
David W Tank [ID] http://orcid.org/0000-0002-9423-4267
Dmitri B Chklovskii [ID] http://orcid.org/0000-0002-4781-2546
Eftychios A Pnevmatikakis [ID] http://orcid.org/0000-0003-1509-6394

## Decision letter and Author response

Decision letter https://doi.org/10.7554/eLife.38173.027
Author response https://doi.org/10.7554/eLife.38173.028

## Additional files

### Supplementary files

• Transparent reporting form
DOI: https://doi.org/10.7554/eLife.38173.031

### Data availability

All input data used to generate most figures, along with the necessary scripts, is available via Zenodo (https://zenodo.org/record/1659149#.XC_Wcs9Ki9s). The original (pre-non-rigid-motion correction) NF datasets listed in Table 2 are publicly available via https://github.com/CodeNeuro/neurofinder. They were originally shared by the Hausser, Losonczy, Svoboda, and Harvey labs.

The following previously published datasets were used:

| Author(s) | Year | Dataset title | Dataset URL | Database and Identifier |
|---|---|---|---|---|
| Andrea Giovannuc- | 2018 | Datasets | https://doi.org/10.5281/ | Zenodo, 10.5281/zenodo.1659149 |

ci, Johannes Frie-
drich, Pat Gunn,
Brandon L Brown,
Sue Ann Koay, Jian-
nis Taxidis, Farza-
neh Najafi, Jeffrey L
Gauthier, Peng-
cheng Zhou, Baljit S
Khakh, David W
Tank, Dmitri B
Chklovskii, Eftychios
A Pnevmatikakis

Generated: Data
from CalmAn, an
open source tool
for scalable
Calcium Imaging
data Analysis

zenodo.1659149

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
