## [Decision Letter]

Thank you for submitting your article "CaImAn: An open source tool for scalable Calcium Imaging data Analysis" for consideration by *eLife*.

Your manuscript has been thoroughly reviewed by three outstanding reviewers and your software extensively tested by one of the groups. Your technical contribution, while admittedly introducing no novel approaches per se, is seen as a timely and carefully constructed software package that will help a large community of neuroscientists. However, before we can pass final judgment, we ask you to address all of the comments of the reviewers; this extensive request is based on the pointed nature of the comments and the importance of having your software accepted by the largest possible fraction of the imaging community. Please pay particular attention to queries on implementation.

A few key points to attend to include:

It is essential to judge the results with CaImAn against existing pipelines, e.g. the very popular Suite2P (Pachitariu et al.,) There is no expectation that CaImAn will shine in every dimension of use. Yet the authors simply must supply comparative benchmarks.

The trial data sets need to be completely described and, of course, made publicly and fully available if and when the manuscript is accepted.

It should be stated up front that there is no "ground truth" in the sense of simultaneous electrical and calcium measurements, only a comparison with a consensus view (note "tyranny of the majority" in one interpretation). Reviewer three makes a clear suggestion for cross-validation that should be followed.

The authors note that fine-tuning of parameters to individual datasets 'can significantly increase performance'. This must be detailed in light of the experience gained with nine different datasets.

Please augment the analysis of Figure 4 to show how precision and recall separately change with signal-to-noise ratio.

Reviewer #1:

Giovannucci et al., present a new open-source software package for analyzing calcium imaging data collected by two-photon and 1-photon microscopes. While the key algorithms used in the pipeline have been developed over many years and largely overlap with previously published work, new features, e.g. memory mapping, CNN-based classifiers, cross-day registration, have been added to improve the performance and further extend the functionalities of the software. The authors tested the package using various in vivo imaging datasets and compared the results with 'consensus ground truth'. The software is written in Python and is open-source, well-organized, and requires minimal user intervention.

The main advance of the manuscript is to combine a plethora of powerful algorithms that have been developed and improved over almost a decade into a single software package that allows batch processing as well as real-time processing of calcium imaging data. Most of the algorithms have been optimized for speed and performance to allow their use in real-time experiments such as closed-loop and all-optical experiments. Having a software package for calcium imaging data analysis that is freely available, well-documented and makes use of the most recent algorithms for the multiple steps involved in calcium imaging processing is extremely valuable for the field, as it will improve the quality and the efficiency of calcium data analysis throughout the neuroscience community, and also open up many new experimental avenues. I therefore strongly support publication.

1) My main concern is the lack of rigorous comparison of the performance of the package with existing solutions. The manuscript only contains internal comparisons but does not show comparisons to existing pipelines (e.g. Suite2P by Pachitariu et al.,) or earlier versions described in other manuscripts (e.g. CNMF in Pnevmatikakis et al., 2016), or state-of-the-art databases such as Neurofinder. This greatly increases the difficulty of judging the overall performance of the software package in its current form. Although the effort the authors went through to generate an improved ground truth data set is laudable, this does not allow one to judge the performance of the package with respect to existing packages/algorithms. For example, a test of how the package performs on the Neurofinder data and how existing packages (like e.g. Suite2P) perform on their newly generated ground truth data would help one to assess the performance of the ROI detection, which is one of the core features of the pipeline. While some of the comparisons have been performed in the respective publications, an overview of the performance of the pipeline with respect to its modules in their current form would be extremely informative.

2) Following on from the previous point, in the Discussion, the authors claim that 'apart from computational gains both approaches [MapReduce batch and OnACID online] also result in improved performance'. Is this with respect to CNMF-based approaches or other existing methods, for example, Suite2pl? If the latter, then the caiman software was never directly compared to any other approaches commonly used (discussed in the Related work section) to validate if it outperforms them in terms of speed or accuracy (only shown similar number of cells detected to Zhou et al., (2018)). Some of the datasets used for ground truth testing could be easily analyzed using different algorithms to provide the comparison and demonstrate the advantages of Caiman. Pachitariu et al., (2018) showed Suite2p detects more cells (in particular, cells with low baseline firing rate) than CNMF-based method does. It would be helpful to know in which way Caiman outperforms Suite2P.

3) The authors claim that CaImAn Online outperforms Batch in terms of neuron detection for longer recordings, while CaImAn Batch is better suited for shorter recordings (Table 1 and also in the Discussion section). What is this claim based on, or what is the performance measure considered? Assuming longer recordings as > 3000 frames and by inspecting Table 2, there is no clear distinction in terms of F1 scores or precision/recall scores (Table 1) based on the file length. Additionally, it is currently hard to inspect this claim as the information to be compared is stored in two separate tables.

4) The manuscript states that fine-tuning of parameters to individual datasets 'can significantly increase performance' (subsection “CAIMAN Batch and CAIMAN online detect neurons with near-human accuracy”) but no evidence is provided for this rather strong statement. It would be interesting to see the results of one of the videos analyzed in this section when the parameters were fine-tuned to show if such adjustment could lead to more reliable results.

5) In Figure 4A, why are the consensus ground truth components different in the Caiman batch panels and the caiman online panels? Were they constructed using the same SEEDINITIALIZATION procedure?

a) In the same figure, there are clearly some components that do not look like cells, in both Caiman batch/online detected components, and even in the consensus ROIs. The authors should add two panels to this figure, one showing the raw FOV image (preferably just the average image of the video, without further processing) with the manual selected ROI contours marked on it – which should look similar to the last panel in Figure 3A; and another showing the 'consensus ground truth' components given by the SEEDINITIALIZATION on top of the average image.

b) A worry is that the initialization procedure might add error to the ground truth and bias it towards the same direction as caiman does, and therefore masking error from benchmarking. For example, if one cell is mistakenly split into several components in both consensus data and caiman data, and both methods detected them, the performance metrics will be artifactually higher. It is important to directly compare the manually-picked ROIs in Figure 3 with the caiman-detected ROIs in Figure 4A (yellow), as the authors claim in the abstract that the software 'achieves near-human performance in detecting locations of active neurons'. It would be good to quantify how well the spatial footprints of the manually drawn ROIs and the caiman detected ROIs overlap. At least please plot the number of ROIs detected by humans against that by the software.

6. Figure 4B shows that the F1 score of Caiman online is higher than that of Caiman batch. And also, in text, the recall value of Caiman online is higher than Caiman batch. In the Materials and methods section, the authors state Caiman online uses a more strict CNN classifier to avoid false positives – I would expect this to result in a higher precision and lower recall, compared to the results given by Caiman batch – but it turns out to be the opposite. Is it a result of comparing the results with different consensus ground truth data?

7) The authors analysed how the F1 score of Caiman Batch depends on SNR, but do not directly address the question of why precision is much higher than recall. It would be helpful if the authors would should show how precision and recall change with SNR separately in addition to the current Figure 4D.

8) Motion correction algorithms and CNN classifier based neuronal filtering are discussed for the online purposes while the computational performance section does not touch on their speed. Running of the demo files in the GitHub link provided by the authors shows that they both introduce additional delays. The CNN classifier (evaluate_components_CNN function) run every few hundred frames is especially computationally expensive (> a few hundred ms on my device). In Figure 8C, is the CNN classifier used? If not, how much time delay will that induce? The computational cost of motion correction and CNN classifier should be reported to verify that.

9) In Figure 8C, does the 'processing time' in the left panel include the time for 'updating shapes'? Judging from the right panel, if the 'updating shapes' is enabled, it could take larger than 15 seconds to complete processing one frame, which means the software could not function in real-time. Subsection “Computational performance of CAIMAN” say that the shape update functionality can be distributed over several frames or done in parallel to allow for real-time processing. Was this already implemented into the online software? Also, how does the shape update functionality impact the activity traces, and does it bring any substantial advantages? This is not shown (Giovannucci et al., (2017) also does not quantify the differences) but can be useful to know for online implementation of the algorithm into experimental setups. It would be helpful if the authors would quantify how much improvement the 'updating shapes' step actually brings to the fidelity of calcium traces. If the improvement is minimal, then the user might skip this step if speed is the priority.

10) Figure 3, Figure 7, Figure 9, Appendix 0—figure 10, Appendix 0—figure 11: how were these FOV images generated? Are they the average frames of the motion-corrected videos or are the intensities of pixels within ROIs enhanced somehow? Please clarify. Please show the average image of the videos without enhancement.

Reviewer #2:

Let me start this review by acknowledging that CaImAn is unequivocally the state of the art toolbox for calcium imaging, and is an impressive piece of work.

However, this manuscript a bit confusing. It reads like the combination of an advertisement and documentation, without ever referencing (quantitatively) the gains/performance of the tools in caiman relative to any other tools in the literature. In other words, it is a description and benchmarking of a given tool.

Such a document is very useful, but also a bit confusing because that is not what peer-reviewed manuscripts typically look like. *eLife* is an interesting journal/venue, so perhaps they are interested in publishing something like this, I won't comment on its suitability for publishing in its current form. I will, however, describe what I believe to be the most important novel contributions that are valuable to the field that are in this manuscript, with some suggested modifications to clarify/improve on some points.

1) Nine new manually labeled datasets, with four "expert" labelers. In spike sorting, the most important paper ever (imho) was the one that generate "ground truth" data. this manuscript does not do that; one could image a "ground truth" channel but this manuscript does basically provide an upper bound on accuracy for any algorithm on these calcium imaging datasets, as defined by the consensus. This is potentially incredibly valuable. however, the details of the datasets are merely listed in a table in the Methods section, but not quantitative description / evaluation of them. As a resource, showing some images of sample traces, sample frames, summary statistics so that potential users could evaluate and compare their data with these data would be very valuable and interesting. comparing them, how hard is each for the experts, and what about the machines, is the hardness correlated, etc.? And this would be a highlight of this manuscript.

2) The parallelization of the standard method in the field, as in Figure 8, is a new result, and interesting but the metrics are not "standard" in the parallel programming literature. I'd recommend a few relevant notions of scaling, including strong scaling and weak scaling (https://en.wikipedia.org/wiki/Scalability#Weak_versus_strong_scaling), as well as "scale up" and "scale out" (https://en.wikipedia.org/wiki/Scalability).

I think strong, weak, scale up, and scale out would be the appropriate quantities to plot for Figure 8, as well as a reference of "optimal scaling", which is available, e.g., from Amdahl's law (https://en.wikipedia.org/wiki/Amdahl%27s_law). Perhaps users also want the quantities that are plotted but, in terms of demonstrating high quality parallelization procedures, the scaling plots are more standard and informative.

3) Registration of components across days. It seems one pub previously addressed this, but caiman has a new strategy, employing the (basically standard) approach to aligning points, the Hungarian algorithm. note that Hungarian scales with n^3 naively, though faster implementations are available for sparse data, etc. MATLAB actually has much better implementations than Python last I checked. In any case, I believe this approach is better than the previously publish one just based on first principles, but no evidence is reported, just words. I also believe it is faster when n is small, but when n is big, I suspect it would be slower. A quantitative comparison of accuracy and speed would be desirable. Also, Hungarian typically requires the *same* number of points, but there is no guarantee that this will be the case. I did not quite understand the comment about infinite distances.

4) Caiman batch and online have a number of minor features/additions/tweaks/bug-fixes relative to previous implementations. however, the resulting improvements in performance is not documented. perhaps that is unnecessary. but a clear list of novel contributions would be very instructive. Getting a tool from "kind of working in our lab" to "actually working in lots of other labs" is really hard, and really important. But the level of effort that went in to doing that is obscured in the text. perhaps this is the most important point of the manuscript.

5 It is very impressive that all the results were obtained using the same parameters. however, the caveat to that is that all the results were based on using the parameters jointly chosen on *these datasets*, meaning that one would expect the performance on essentially any new dataset to be worse. Held out datasets can provide evidence that performance does not drop too much. With more information on each datasets (see point 1), and similar knowledge of held-out data, one could use these results to predict how well these particular parameters would work on a new dataset that a different lab might generate. Note that this is the same problem as the very popular benchmarking problems in machine vision that are very popular, which doesn't stop those papers from getting published in top journals.

6) In the end, it remains unclear precisely what you recommend to do and when. Specifically, the manuscript mentions many functions/options/setting/parameters and several pre-processing stages. I also understand that the jupyter notebooks provide some examples. I think some concrete guidance, given that you've now analyzed nine different datasets spanning labs, sensors, etc., you have a wealth of knowledge about the heterogeneity of these datasets that would be valuable for the rest of the community, but is not quite conveyed in the manuscript.

Reviewer #3:

Giovannucci et al., present a software package to process calcium imaging data. As calcium imaging is becoming a dominant methodology to record neural activity, a standardized and open-source analytical platform is of importance. CaImAn presents one such effort. The manuscript is well written, and the software seems state-of-the-art and properly validated. I believe *eLife* Tools and Resources is a perfect outlet for such a commendable effort and enthusiastically support the publication of the study. Below are some comments that I believe the authors can address relatively easily prior to publication.

Subsection “Batch processing of large scale datasets on standalone machines” (parallelization). First, it should be mentioned that the difficulty of parallelization arises from this particular motion correction algorithm in which correction of a frame depends on the previous frames. If each frame is processed independently, like in some other motion correction methods, it would be trivial to parallelize. Second, is there any mechanism to ensure that there is no jump between the temporal chunks? In other words, is the beginning of chunk X adjusted to match the end of chunk X-1? This seems critical for subsequent analysis.

Subsection “CAIMAN BATCH and CAIMAN ONLINE detect neurons with near-human accuracy” (ground truth).The authors mention that the human performance is overestimated because the ground truth is based on the agreement among the scorers. This effect can be eliminated, and the true human performance can be estimated by cross-validation. For example, in a dataset annotated by four annotators, one can test the performance of one annotator against a putative ground truth defined by the other three annotators. This can be repeated for all the four annotators and the average will give a better estimate of human performance than the current method.

Subsection “Computational performance of CAIMAN”. The details of the systems are not described in the Materials and methods section. For a Macbook Pro, it has 8 logical CPU cores and 4 physical CPU cores (in 1 CPU). Please specify the CPU model names.

Subsection “Computational performance of CAIMAN” (employing more processing power results in faster processing). From the figure, a 112 CPU cluster processes the data about three times as fast as a 8 CPU laptop. The performance gain does not seem to be proportional to the number of CPUs available even considering the overhead. Please discuss the overhead of parallelization and the bottleneck when processed in the cluster.

Subsection “Computational performance of CAIMAN” (the performance scales linearly). Please clarify whether this (linearity) is a visual observation of the results, or is this based on a complexity analysis of the factorization algorithm?

Subsection “Computational performance of CAIMAN” (enabling real-time processing). Please clarify how often such an update of a footprint is required in order to support real-time processing without parallelization.

Reviewing editor:

Please consider adding a reference under "Related Work" to the use of supervised learning ("Adaboost") to segment active neurons, i.e. Automatic identification of fluorescently labeled brain cells for rapid functional imaging. I. Valmianski, A. Y. Shih, J. D. Driscoll, D. M. Matthews, Y. Freund and D. Kleinfeld, Journal of Neurophysiology (2010) 104:1803-1811.

---

## [Author Response]

Your manuscript has been thoroughly reviewed by three outstanding reviewers and your software extensively tested by one of the groups. Your technical contribution, while admittedly introducing no novel approaches per se, is seen as a timely and carefully constructed software package that will help a large community of neuroscientists. However, before we can pass final judgment, we ask you to address all of the comments of the reviewers; this extensive request is based on the pointed nature of the comments and the importance of having your software accepted by the largest possible fraction of the imaging community. Please pay particular attention to queries on implementation.

We would like to thank the reviewers and the editor for their thorough and thoughtful reviews. CaImAn is a big project and the reviews touched upon almost all of its aspects leading to a large revision of our paper. We believe that our current submission addresses all the issues raised by reviewers. In summary the revised version of our paper includes (among many other improvements):

A link to a (password protected) website that contains all the datasets, their characteristics and the manual annotations.

A comparison of CaImAn with the popular package Suite2p.

A more systematic exploration of the parameter space for both CaImAn batch and CaImAn online, to better distinguish between “average” and “best” cases, and insight for parameter choosing.

A systematic study on the computational speed for both CaImAn batch and CaImAn online, and the effect of parallelization.

Concurrently with this submission we also released a new version of CaImAn, that presents a simplified way to run the various algorithms and pass the parameters.

Please find our detailed responses below. Since our paper is long, we copied excerpts from the paper (text, figures etc.) that directly address the reviewers’ concerns.A few key points to attend to include:It is essential to judge the results with CaImAn against existing pipelines, e.g. the very popular Suite2P (Pachitariu et al.) There is no expectation that CaImAn will shine in every dimension of use. Yet the authors simply must supply comparative benchmarks.

With our revised submission we provide a comparison between Suite2p and CaImAn. The results are described in the text and appear in a new Figure 4—figure supplement 2.

To compare against Suite2p we used a grid search for some parameters around the defaults provided by its developers and report the combination that gave the best results on average. The software and scripts were cloned from the master branch of the GitHub repository (https://github.com/cortex-lab/Suite2P) on August 27th 2018. We modified the make_db_example.mand master_file_example.mscripts to accommodate for the grid parameter search. One file was excluded from the comparison (J123) because Suite2p would run a significantly large number of iterations and add tens of thousands of components (J123 has very low SNR, it is possible that some ad hoc parameter needed to be set in Suite2p). Suite2p comes with a trainable classifier; to keep the comparison fully automated this classifier was not retrained for each dataset, yet it was used with its default and results were reported both using the default general suite2P classifier or without any classifier. In both conditions we selected the set of parameters providing the highest average f1_score over all the datasets. In all cases CaImAn outperformed Suite2p with varying degrees of difference. Suite2p includes a supervised learning based classifier that is used to evaluate components. The *MEAN +/- STD* F1_score obtained with the default classifier was 0.59 +/-.12 (optimal parameters {1500, 2000,0.25}), whereas without classifier we obtained an F1_score of 0.55 +/- 0.12 ({500,2000,0.25}). It is expected that separate training on each dataset will yield better results (as is also the case with CaImAn). As we stated in our initial cover letter, comparisons like this are not definitive and Suite2p in the hands of a more experienced user might yield better results than the results reported here. We in fact contacted the main developer of Suite2p prior trying out the comparison on our own but received no response. As such we choose to present these results in the supplement and avoid overselling them.

The trial data sets need to be completely described and, of course, made publicly and fully available if and when the manuscript is accepted.

We have deposited the raw data and manual and consensus annotations to Zenodo.

It should be stated up front that there is no "ground truth" in the sense of simultaneous electrical and calcium measurements, only a comparison with a consensus view (note "tyranny of the majority" in one interpretation). Reviewer three makes a clear suggestion for cross-validation that should be followed.

We agree. In the revised manuscript we changed our phrasing to use the term “consensus annotation” as opposed to “consensus ground truth” to make clear your point that there is no well defined ground truth. We also followed the cross-validation approach suggested by reviewer #3 to better understand the variability of human annotators. While the “ground truth” (and thus individual scores) could vary significantly in this case, on average we observed only mild differences. See also our detailed response to the relevant comment from reviewer #3.

The authors note that fine-tuning of parameters to individual datasets 'can significantly increase performance'. This must be detailed in light of the experience gained with nine different datasets.

In the revised version of the paper we made a more systematic search of the parameter space for both CaImAn batch and CaImAn online. We performed a small grid search on some threshold parameters for the quality evaluation tests. For both algorithms we found the set of parameters that give the best results on average for all datasets,as well as the parameters that individually maximize the performance on each dataset. The set of parameters yielding the highest average F1 score was selected as the default and reported in Figure 4B (batch and online) and Table 1. We also checked which choice of parameters maximized the performance for each dataset (batch max and online maxin Figure 4B). The average F1 score are 0.77 +/- 0.03 for batch max, 0.78 +/- 0.04 for online max, 0.75 +/- 0.03 for batch, and 0.76 +/- 0.05 for online. In both cases the average total F1 score can be raised by about 0.02.

We offer more details the Results section and Materials and methods section (Implementation details of CaImAn batch/online).

For CaImAn batch: “The global default parameters for all datasets were obtained by performing a grid search on the 9 datasets over the following values: trace peak SNR threshold {1.75, 2, 2.25, 2.5}, spatial correlation threshold {0.75, 0.8, 0.85}, lower threshold on CNN classifier (reject if prediction is below, {0.05, 0.1, 0.15}) and upper threshold on classifier (accept if prediction is above, {0.9,.95, 0.99, 1}). The best overall parameters (used for the results reported in Table 1) were given for the choice (2, 0.85, 0.1, 0.99).”

For CaImAn online: “To select global parameters for all datasets we performed a grid search on all 9 datasets by varying the following parameters: The peak SNR threshold for accepting a candidate component on the set {0.6, 0.8, 1, 1.2, 1.4, 1.6, 1.8, 2}, the online CNN classifier threshold for accepting candidate components on the set {0.5, 0.55, 0.6, 0.65, 0.7, 0.75}, and the number of candidate components per frame on the set {5, 7, 10, 14}. The best overall parameters (reported in Table 1) were given for the choice $(1.2, 0.65, 10).”

Please augment the analysis of Figure 4 to show how precision and recall separately change with signal-to-noise ratio.

We have followed this suggestion and also show how precision and recall change as a function of the SNR for each file in Figure 4E. The same trend is followed, i.e., performance is higher for neurons with higher SNR traces.

Reviewer #1:Giovannucci et al., present a new open-source software package for analyzing calcium imaging data collected by two-photon and 1-photon microscopes. While the key algorithms used in the pipeline have been developed over many years and largely overlap with previously published work, new features, e.g. memory mapping, CNN-based classifiers, cross-day registration, have been added to improve the performance and further extend the functionalities of the software. The authors tested the package using various in vivo imaging datasets and compared the results with 'consensus ground truth'. The software is written in Python and is open-source, well-organized, and requires minimal user intervention.The main advance of the manuscript is to combine a plethora of powerful algorithms that have been developed and improved over almost a decade into a single software package that allows batch processing as well as real-time processing of calcium imaging data. Most of the algorithms have been optimized for speed and performance to allow their use in real-time experiments such as closed-loop and all-optical experiments. Having a software package for calcium imaging data analysis that is freely available, well-documented and makes use of the most recent algorithms for the multiple steps involved in calcium imaging processing is extremely valuable for the field, as it will improve the quality and the efficiency of calcium data analysis throughout the neuroscience community, and also open up many new experimental avenues. I therefore strongly support publication.1) My main concern is the lack of rigorous comparison of the performance of the package with existing solutions. The manuscript only contains internal comparisons but does not show comparisons to existing pipelines (e.g. Suite2P by Pachitariu et al.,) or earlier versions described in other manuscripts (e.g. CNMF in Pnevmatikakis et al., 2016), or state-of-the-art databases such as Neurofinder. This greatly increases the difficulty of judging the overall performance of the software package in its current form. Although the effort the authors went through to generate an improved ground truth data set is laudable, this does not allow one to judge the performance of the package with respect to existing packages/algorithms. For example, a test of how the package performs on the Neurofinder data and how existing packages (like e.g. Suite2P) perform on their newly generated ground truth data would help one to assess the performance of the ROI detection, which is one of the core features of the pipeline. While some of the comparisons have been performed in the respective publications, an overview of the performance of the pipeline with respect to its modules in their current form would be extremely informative.

We added a detailed comparison of the performance of CaImAn batch to Suite2P. Please refer to our response to the editor comments for details. Besides that, we also analysed all the neurofinder test files and generated the results to be submitted but unfortunately the neurofinder website appears to be down at the moment, therefore submission is impossible (September/October 2018). We have nevertheless added the scripts and results of the performed benchmark in the folder shared with the reviewers. The original CNMF algorithm (Pnevmatikakis et al., 2016) can be considered a subset of the CaImAn batch algorithm. CNMF essentially corresponds to applying CaImAn batch without using patches and without using automated component quality testing. In contrast, CNMF sorted the various components and required from the human to select a cut-off threshold. As such, we chose not to do a formal comparison. For instance, it is not possible to run J123, J115 and K53 on a 128 GB RAM workstation because of the memory load (data not shown).

2) Following on from the previous point, in the Discussion, the authors claim that 'apart from computational gains both approaches [MapReduce batch and OnACID online] also result in improved performance'. Is this with respect to CNMF-based approaches or other existing methods, for example, Suite2pl? If the latter, then the caiman software was never directly compared to any other approaches commonly used (discussed in the Related work section) to validate if it outperforms them in terms of speed or accuracy (only shown similar number of cells detected to Zhou et al., (2018)). Some of the datasets used for ground truth testing could be easily analyzed using different algorithms to provide the comparison and demonstrate the advantages of Caiman. Pachitariu et al., (2018) showed Suite2p detects more cells (in particular, cells with low baseline firing rate) than CNMF-based method does. It would be helpful to know in which way Caiman outperforms Suite2P.

The introduction of patch processing with map reduce, component quality testing, as well as a modified initialization algorithm (RollingGreedyROI vs GreedyROI in CNMF) has improved the results of CaImAn over CNMF in terms of scalability, performance, but also with respect to automation. As we also noted above, the original CNMF algorithm was not fully automated and required the user to order the components and select a cut-off point. As such, we believe a comparison of the two methods will not be particularly instructive. We also do not want to comment on qualitative differences between Suite2p and CaImAn. In its core Suite2p uses a very similar matrix factorization model to the original CNMF algorithm, with the main difference being the background/neuropil model. Other differences arise in the implementation details as well as post-processing methods, are hard to compare qualitatively. Our results directly contradict the statements of Pachitariu et al., (2018) as we demonstrate a qualitatively much better performance of CaImAn than what is reported there.

Just to clarify, for microendoscopic 1p data, CaImAn does not present an alternative algorithm to CNMF-E (Zhou et al., (2018)). It merely just ports the algorithm in Python and endows it with the map-reduce and component quality testing capabilities. As such, we expect the results to be similar (but not identical).

3) The authors claim that CaImAn Online outperforms Batch in terms of neuron detection for longer recordings, while CaImAn Batch is better suited for shorter recordings (Table 1 and also in the Discussion section). What is this claim based on, or what is the performance measure considered? Assuming longer recordings as > 3000 frames and by inspecting Table 2, there is no clear distinction in terms of F1 scores or precision/recall scores (Table 1) based on the file length. Additionally, it is currently hard to inspect this claim as the information to be compared is stored in two separate tables.

The reviewer is right that our statement was vague and not well supported. Here by longer recordings we mean recordings with length > 20000 or, better stated, experiments long enough where the spatial footprints could change during the course of the experiment due to various non-stationarities. In our analysis we observed that in the three longest datasets (J115, J123, and K53) the difference between CaImAn batch and online was the highest, with online achieving better results as quantified by our precision/recall framework. CaImAn online looks at a local window of the data (current frame plus the residual buffer) to update the activity of each neuron and identify new ones. On the other hand, the batch algorithm looks at the all data at once and tries to express the spatio-temporal activity of each neuron as a simple rank one matrix. We believe that this important difference endows the online algorithm with more robustness to various non-stationarities that can arise in long experimental sessions. To ease the presentation, we have re-ordered the datasets in all the Figures and Tables in an increasing order according to their number of frames and included the number of frames on Table 1. We have also rephrased our wording and included additional discussion to better express our views:

“[…]While the two algorithms performed similarly on average, CaImAn online tends to perform better for longer datasets (e.g., datasets J115, J123, K53 that all have more than 40000 frames). CaImAn batch operates on the dataset at once, representing each neuron as constant in time spatial footprint. In contrast, CaImAn online operates at a local level looking at a short window over time to detect new components, while adaptively changing their shape based on the new data. This enables CaImAn online to adapt to slow non-stationarities that can appear in long experiments.”

4) The manuscript states that fine-tuning of parameters to individual datasets 'can significantly increase performance' (subsection “CAIMAN Batch and CAIMAN online detect neurons with near-human accuracy”) but no evidence is provided for this rather strong statement. It would be interesting to see the results of one of the videos analyzed in this section when the parameters were fine-tuned to show if such adjustment could lead to more reliable results.

We thank the reviewer for this suggestion. To quantify this comment, we repeated the analysis on the nine labeled datasets using a small grid search over several parameters. The results show an average improvement of 0.02 in terms of F1 score and also highlight strategies for parameter choice. We present these results in Figure 4B and note in the text (see also relevant answer to the editor for parameter details):

“[…] In general by choosing a parameter combination that maximizes the value for each dataset, the performance increases across the datasets with F_1 scores in the range 0.72-0.85 and average performance 0.78 +/- 0.05 (see Figure 4B (orange) and Figure 4—figure supplement 1 (magenta)). This analysis also shows that in general a strategy of testing a large number of components per timestep, but with stricter criteria, achieves better results than testing fewer components with looser criteria (at the expense of increased computational cost). The results also indicate different strategies for parameter choice depending on the length of a dataset: Lower threshold values and/or larger number of candidate components (Figure 4—figure supplement 1(red)), lead to better values for shorter datasets, but can decrease precision and overall performance for longer datasets. The opposite also holds for higher threshold values and/or smaller number of candidate components (Figure 4—figure supplement 1(blue)), where CaImAn online for shorter datasets can suffer from lower recall values, whereas in longer datasets CaImAn online can add neurons over a longer period of time while maintaining high precision values and thus achieve better performance.”

5) In Figure 4A, why are the consensus ground truth components different in the Caiman batch panels and the caiman online panels? Were they constructed using the same SEEDEDINITIALIZATION procedure?

We thank the reviewer for the careful inspection and for pointing out this issue. We apologize for our oversight, it should not have happened in the first place. The mismatch stemmed from two different thresholds used to binarize the ground truth. It is now fixed in Figure 4A.

a) In the same figure, there are clearly some components that do not look like cells, in both Caiman batch/online detected components, and even in the consensus ROIs. The authors should add two panels to this figure, one showing the raw FOV image (preferably just the average image of the video, without further processing) with the manual selected ROI contours marked on it – which should look similar to the last panel in Figure 3A; and another showing the 'consensus ground truth' components given by the SEEDEDINITIALIZATION on top of the average image.

We added Figure 3—figure supplement 1 comparing the original masks and the ones after seeded initialization for the dataset. Panel b shows that SEEDEDINITIALIZATION allows the original elliptical annotations to adapt to the actual footprint of each component as shown in the FOV. Figure 3—figure supplement 1C also demonstrates some examples of the thresholding process. The main goal of this process is to obtain the most prominent (visible) parts of the neuron, a step that facilitates the comparison process.

We have also modified the text to make this point more clear:

“[…]To compare CaImAn against the consensus annotation, the manual annotations were used as binary masks to construct the consensus spatial and temporal components, using the SeededInitialization procedure (Algorithm 3) of CaImAn Batch. This step is necessary in order to adapt the manual annotations to the shapes of the actual spatial footprints of each neuron in the FOV (Figure 3—figure supplement 1), since manual annotations had in general an elliptical shape.”

b) A worry is that the initialization procedure might add error to the ground truth and bias it towards the same direction as caiman does, and therefore masking error from benchmarking. For example, if one cell is mistakenly split into several components in both consensus data and caiman data, and both methods detected them, the performance metrics will be artifactually higher. It is important to directly compare the manually-picked ROIs in Figure 3 with the caiman-detected ROIs in Figure 4A (yellow), as the authors claim in the abstract that the software 'achieves near-human performance in detecting locations of active neurons'. It would be good to quantify how well the spatial footprints of the manually drawn ROIs and the caiman detected ROIs overlap. At least please plot the number of ROIs detected by humans against that by the software.

The SEEDEDINITIALIZATION procedure cannot split (or merge) any selection. Essentially what it does is a simple non-negative matrix factorization where the spatial footprints of each components are constrained to be subsets of the input annotations, together with additional components for background. As noted in our previous response, this is necessary to ensure a comparison among similar ROIs. While this can introduce a bias in terms of the extracted traces, we believe that this bias is significantly less compared to non-model based approaches. For example, simply computing the average over a selected ROI which would introduce contamination from neuropil or components with overlapping spatial footprints. We have updated Table 1 to also include the number of neurons selected by each labeller and our algorithms.

6. Figure 4B shows that the F1 score of Caiman online is higher than that of Caiman batch. And also, in text, the recall value of Caiman online is higher than Caiman batch. In the Materials and methods section, the authors state Caiman online uses a more strict CNN classifier to avoid false positives – I would expect this to result in a higher precision and lower recall, compared to the results given by Caiman batch – but it turns out to be the opposite. Is it a result of comparing the results with different consensus ground truth data?

There is an important difference between the two classifiers. The CNN classifier for the CaImAn batch algorithm is applied only once for every component that has been selected from the CNMF algorithm to test its quality. If it does not pass the test, then it is excluded from the final list of active components. However, the CNN classifier for the CaImAn online algorithm is applied to every candidate component considered at each frame. If a component does not get selected at some point from the online CNN classifier then it can be considered again for inclusion in the future once more signal has been integrated. As a result, it can be included at a later point during the experiment. This explains our choice to train the online CNN classifier to be stricter. This salient point is explained in subsection “Differences between the two classifiers”.

“Although both classifiers examine the spatial footprints of candidate components, their required performance characteristics are different which led us to train them separately. First of all, the two classifiers are trained on separate data: The batch classifier is trained on spatial footprints extracted from CaImAn batch, whereas the online classifier is trained on residual signals that are generated as CaImAn online operates. The batch classifier examines each component as a post-processing step to determine whether its shape corresponds to a neural cell body. As such, false positive and false negative examples are treated equally and possible mis-classifications do not directly affect the traces of the other components. By contrast, the online classifier operates as part of the online processing pipeline. In this case, a new component that is not detected in a residual buffer is likely to be detected later should it become more active. On the other hand, a component that is falsely detected and incorporated in the online processing pipeline will continue to affect the future buffer residuals and the detection of future components. As such the online algorithm is more sensitive to false positives than false negatives. To ensure a small number of false positive examples under testing conditions, only components with average peak-SNR value at least 4 were considered as positive examples during training of the online classifier**”**

7) The authors analysed how the F1 score of Caiman Batch depends on SNR, but do not directly address the question of why precision is much higher than recall. It would be helpful if the authors would should show how precision and recall change with SNR separately in addition to the current Figure 4D.

We updated Figure 4D to show how precision and recall change as a function of the SNR, per the reviewer’s instructions (see also our response to the editor). The main reason for the discrepancy between precision and recall is the existence of the quality assessment step in CaImAn batch and the similar tests that are used during CaImAn online. These tests aim to filter out false positive components a step that will increase precision but do not introduce any missed out components, a step that would increase recall. In fact, if a true positive component is filtered out the recall will actually fall. Appropriately changing the thresholds of the various tests can lead to increased recall at the price of reduced precision. However, we believe that having a higher precision is desirable. As we note in the Discussion section:

“[…] The performance of CaImAn (especially in its batch version) indicates a considerably higher precision than recall in most datasets. While more balanced results can be achieved by appropriately relaxing the relevant quality evaluation thresholds, we prefer to maintain a higher precision as we believe that the inclusion false positive traces can be more detrimental in any downstream analysis compared to the exclusion of, typically weak, true positive traces. This statement is true especially in experiments with low task dimensionality where a good signal from few neurons can be sufficient for the required hypothesis testing.”

8) Motion correction algorithms and CNN classifier based neuronal filtering are discussed for the online purposes while the computational performance section does not touch on their speed. Running of the demo files in the GitHub link provided by the authors shows that they both introduce additional delays. The CNN classifier (evaluate_components_CNN function) run every few hundred frames is especially computationally expensive (> a few hundred ms on my device). In Figure 8C, is the CNN classifier used? If not, how much time delay will that induce? The computational cost of motion correction and CNN classifier should be reported to verify that.

These are both valid observations. Motion correction adds a computational cost to the online pipeline, especially if non-rigid motion correction is required. In our case, motion correction was done beforehand to ensure that the FOV where CaImAn online operates is perfectly aligned to the FOV that the annotators used. As we note in the text:

“[…] The analysis here excludes the cost of motion correction, because the files where motion corrected before hand to ensure that manual annotations and the algorithms where operating on the same FOV. This cost depends on whether rigid or pw-rigid motion correction is being used. Rigid motion correction taking on average 3-5ms per frame for a 512 x 512 pixel FOV, whereas pw-rigid motion correction with patch size 128 x 128 pixel is typically 3-4 times slower.”

The online CNN classifier is run on every step to evaluate the spatial footprints of the candidate components. This also adds a high computational overload (>=10ms) which we found to depend critically on the computing infrastructure. For example, utilizing a GPU, which is not the default mode of installing CaImAn, can help speed up this process significantly. Note that the cost of the CNN classifier mostly occurs from calling the underlying neural network at every frame and less so on the number of components that it needs to check at every point. We discuss these issues further in the paper and have modified Figure 8 to include a breakdown of the computational cost per frame for one dataset (J123).

“The cost of detecting and incorporating new components remains approximately constant across time and is dependent on the number of candidate components at each timestep. In this example 5 candidate components were used per frame resulting in a relatively low cost (~7ms per frame). As discussed earlier, a higher number of candidate components can lead to higher recall in shorter datasets but at a computational cost. This step can benefit by the use of a GPU for running the online CNN on the footprints of the candidate components. Finally, as also noted in [Giovannucci et al., 2017], the cost of tracking components can be kept low, and shows a mild increase over time as more components are being added by the algorithm.”

The cost of running the CNN classifier for the zebrafish example is not substantially higher and is significantly smaller than the intervolume time of 1s. An architecture where the CNN classifier is run on a parallel stream can significantly help there although we have not implemented such an architecture yet.

9) In Figure 8C, does the 'processing time' in the left panel include the time for 'updating shapes'? Judging from the right panel, if the 'updating shapes' is enabled, it could take larger than 15 seconds to complete processing one frame, which means the software could not function in real-time. Subsection “Computational performance of CAIMAN” say that the shape update functionality can be distributed over several frames or done in parallel to allow for real-time processing. Was this already implemented into the online software? Also, how does the shape update functionality impact the activity traces, and does it bring any substantial advantages? This is not shown (Giovannucci et al., (2017) also does not quantify the differences) but can be useful to know for online implementation of the algorithm into experimental setups. It would be helpful if the authors would quantify how much improvement the 'updating shapes' step actually brings to the fidelity of calcium traces. If the improvement is minimal, then the user might skip this step if speed is the priority.

Periodically updating the shapes is an important step in the online pipeline: As the experiment proceeds and a neuron keeps firing spikes the algorithm can accumulate more information and produce refined estimates about the neuron’s spatial footprint. This is important for both estimating its future activity more accurately and for distinguishing it from neighboring neurons. Moreover, the spatial footprint of a neuron can slowly vary with time especially in long experiments. The online algorithm can natively adapt to changes like this unlike batch approaches. We have modified our code so that now the shape update is indeed distributed among all the frames and happens only in frames where no new neurons are added to further distribute the cost evenly. We developed a simple algorithm that ensures (i) that every neuron gets updated every N frames (where N is a user defined parameter, default value 200) and (ii) if a neuron is added, then the spatial footprints of all neighboring neurons are updated to adapt to the presence of their new neighbor. We describe this process more analytically in the Materials and methods section (see: distributed shape update). Based on this approach we modified Figure 8 to reflect this new approach and show how the computational cost per frame is allocated to each step of the online algorithm.

“As discussed in Giovannucci et al., (2017) processing time of CaImAn online depends primarily on i) the computational cost of tracking the temporal activity of discovered neurons, ii) the cost of detecting and incorporating new neurons, and iii) the cost of periodic updates of spatial footprints. Figure 8e shows the cost of each of these steps for each frame, for one epoch of processing of the dataset J123. Distributing the spatial footprint update more uniformly among all frames removes the computational bottleneck appearing in Giovannucci et al., (2017), where all the footprints where updated periodically at the same frame.”

10) Figure 3, Figure 7, Figure 9, Appendix 0—figure 10, Appendix 0—figure 11: how were these FOV images generated? Are they the average frames of the motion-corrected videos or are the intensities of pixels within ROIs enhanced somehow? Please clarify. Please show the average image of the videos without enhancement.

The figure that we generally choose as a background is the so-called correlation image (or for longer datasets the max-correlation image). The value of the correlation image at each pixel corresponds to the average of the correlation coefficients between the trace of a pixel and its neighbors (in an already motion corrected movie). We included this definition in the Materials and methods section but now also refer to it in footnote 3 in the text:

“[…] The value of the correlation image for each pixel represent the average correlation (across time) between the pixel and its neighbors. This summarization can enhance active neurons and suppress neuropil for two photon datasets (Figure 10A). See Materials and methods section (Collection of manual annotations)”.

We choose this image because it demonstrates invariance with respect to expression levels and it has the property of attaining high values in pixels that are part of active neurons and lower values for pixels that are not. Since our methods focus on the detection of active neurons, their performance can be better assessed visually by plotting contours against the correlation image as opposed to e.g., the mean. Note that the use of the correlation image is very common in practice and has also been used for source extraction algorithms (as we detail in the related work). To make this point more clear we included the new Figure 3—figure supplement 1A that shows the median (equivalent to mean) and correlation images for three datasets and demonstrates the utility of the correlation image. Median and correlation images overlaid to all the manual labels are shown in the supporting website that contains all our data.

Reviewer #2:Let me start this review by acknowledging that CaImAn is unequivocally the state of the art toolbox for calcium imaging, and is an impressive piece of work.However, this manuscript a bit confusing. It reads like the combination of an advertisement and documentation, without ever referencing (quantitatively) the gains/performance of the tools in caiman relative to any other tools in the literature. In other words, it is a description and benchmarking of a given tool.Such a document is very useful, but also a bit confusing because that is not what peer-reviewed manuscripts typically look like. eLife is an interesting journal/venue, so perhaps they are interested in publishing something like this, I won't comment on its suitability for publishing in its current form. I will, however, describe what I believe to be the most important novel contributions that are valuable to the field that are in this manuscript, with some suggested modifications to clarify/improve on some points.

We thank the reviewer for the useful comments. We added a formal comparison with a state-of-the-art package for calcium imaging data (Suite2P) and demonstrate that CaImAn is competitive when benchmarked against consensus annotations (see answer to editor for details).

1) Nine new manually labeled datasets, with four "expert" labelers. In spike sorting, the most important paper ever (imho) was the one that generate "ground truth" data. this manuscript does not do that; one could image a "ground truth" channel but this manuscript does basically provide an upper bound on accuracy for any algorithm on these calcium imaging datasets, as defined by the consensus. This is potentially incredibly valuable. however, the details of the datasets are merely listed in a table in the methods, but not quantitative description / evaluation of them. As a resource, showing some images of sample traces, sample frames, summary statistics so that potential users could evaluate and compare their data with these data would be very valuable and interesting. comparing them, how hard is each for the experts, and what about the machines, is the hardness correlated, etc.? And this would be a highlight of this manuscript.

We have created a website that can be used to download the raw data and manual and consensus annotations. For each dataset, the website also depicts the labels generated by each labelers and the consensus overlaid on the correlation image, as well as example spatial and temporal components extracted by CaImAn. The site can be found at the url https://users.flatironinstitute.org/~neuro/caiman_paper/

To log-in use the username “reviewers” and password “island”. We kindly ask the reviewers to not share these credentials; we will make the site publicly accessible should the paper gets accepted.

2) The parallelization of the standard method in the field, as in Figure 8, is a new result, and interesting but the metrics are not "standard" in the parallel programming literature. I'd recommend a few relevant notions of scaling, including strong scaling and weak scaling (https://en.wikipedia.org/wiki/Scalability#Weak_versus_strong_scaling), as well as "scale up" and "scale out" (https://en.wikipedia.org/wiki/Scalability).I think strong, weak, scale up, and scale out would be the appropriate quantities to plot for Figure 8, as well as a reference of "optimal scaling", which is available, eg, from Amdahl's law (https://en.wikipedia.org/wiki/Amdahl%27s_law). Perhaps users also want the quantities that are plotted but, in terms of demonstrating high quality parallelization procedures, the scaling plots are more standard and informative.

We thank the reviewer for this useful suggestion. We have included a new panel in Figure 8 (panel c) that shows the scaling of CaImAn batch when processing a dataset in the same machine but using a different number of CPUs. Even though it would be desirable to plot this Figure all the way up to 112 CPUs (or more) we found that the usage of a computing cluster was leading to variable results due to variable speeds for reading the files over network drives, thus hindering our conclusions. While the language of strong vs weak scaling can help demonstrate the properties of the algorithm, we believe that it can be highly technical for a neuroscience audience and refrained from using it to describe our results. Instead we note:

“[…] To the study the effects of parallelization we ran CaImAn batch on the same computing architecture (24CPUs) utilizing a different number of CPUs at a time (Figure 8c). In all cases significant speedup factors can be gained by utilizing parallel processing, with the gains being similar for all stages of processing (patch processing, refinement, and quality testing, data not shown). The results show better scaling for medium sized datasets (J123, ~50GB). For the largest datasets (J115, ~100GB), the speedup gains saturate due to limited RAM, whereas for small datasets (~5GB) the speedup factor can be limited by the increased fraction of communications cost overhead (an indication of weak scaling in the language of high performance computing).”

3) Registration of components across days. It seems one pub previously addressed this, but caiman has a new strategy, employing the (basically standard) approach to aligning points, the Hungarian algorithm. note that Hungarian scales with n^3 naively, though faster implementations are available for sparse data, etc. MATLAB actually has much better implementations than Python last I checked. In any case, I believe this approach is better than the previously publish one just based on first principles, but no evidence is reported, just words. I also believe it is faster when n is small, but when n is big, I suspect it would be slower. A quantitative comparison of accuracy and speed would be desirable. Also, Hungarian typically requires the *same* number of points, but there is no guarantee that this will be the case. I did not quite understand the comment about infinite distances.

Our response to this comment can be separated in three parts:

A) It is true that the Hungarian algorithm is not the most computationally efficient solution for the linear assignment problem. However, for the population sizes that we encounter the cost of solving the linear assignment problem with the Hungarian problem is only a small fraction of the total cost. Moreover, the cubic cost refers to the general case of a dense unstructured affinity matrix. In our case, the matrix is sparse since most pairs of neurons are far from each other and are not considered for registration. This is the reason why we assign infinite distances (i.e., zero affinity) in this case. The sparse matrix leads to faster matrix vector operations and eventually a faster solution.

B) The infinite distances also permit to have neurons that are unmatched (i.e., have infinite distance from all the neurons in the other session) and enable registering sessions with unequal number of neurons. Every neuron that is matched in the other session with infinite distance is considered as unmatched.

C) Comparison with the method of Sheintuch et al., is not easy because of the absence of ground truth information. The metric proposed in Sheintuch et al., is tailored for their approach, since it uses the confidence in the assignment that comes from their probabilistic approach. To better compare the two approaches, we applied our method to the same publicly available Allen Brain datasets and computed the transitivity index. For all the datasets considered our transitivity index was very high (>0.99). A similar analysis already appeared in our initial submission in Figure 8C where we compared the approach of registering components through union vs direct registration.

We modified the paper to include the additional analysis and clarifications.

"A different approach for multiple day registration was recently proposed by Sheintuch et al., (2017) (CellReg). While a direct comparison of the two methods is not feasible in the absence of ground truth, we tested our method against the same publicly available datasets from the Allen Brain Observatory visual coding database. (http://observatory.brain-map.org/visualcoding). Similarly, to Sheintuch et al., (2017) the same experiment performed over the course of different days produced very different populations of active neurons. To measure performance of RegisterPair for pairwise registration, we computed the transitivity index proposed in Sheintuch et al., (2017). The transitivity property requires that if cell "a" from session 1 matches with cell "b" from session 2, and cell "b" from session 2 matches with cell "c" from session 3, then cell "a" from session 1 should match with cell "c" from session 3 when sessions 1 and 3 are registered directly. For all ten tested datasets the transitivity index was very high, with values ranging from 0.976 to 1 (0.992 +/- 0.006, data not shown). A discussion between the similarities and differences of the two methods is given in Methods section and Materials and methods section.”

4) Caiman batch and online have a number of minor features/additions/tweaks/bug-fixes relative to previous implementations. however, the resulting improvements in performance is not documented. perhaps that is unnecessary. but a clear list of novel contributions would be very instructive. Getting a tool from "kind of working in our lab" to "actually working in lots of other labs" is really hard, and really important. But the level of effort that went in to doing that is obscured in the text. perhaps this is the most important point of the manuscript.

We have tried to make a list of the contributions summarized in this work in the Introduction (see: Contributions). We elaborate more on them in the subsequent Methods section and Materials and methods section. These contributions refer more to algorithmic developments as well as the release of the labeled datasets. With respect to software contributions (e.g., bug fixes) we believe that providing a list would be rather daunting and without clear benefits. Our public code repository is a better source for that information and we invite interested parties to follow (or even participate in) the development of CaImAn there.

5 It is very impressive that all the results were obtained using the same parameters. however, the caveat to that is that all the results were based on using the parameters jointly chosen on *these datasets*, meaning that one would expect the performance on essentially any new dataset to be worse. Held out datasets can provide evidence that performance does not drop too much. With more information on each datasets (see point 1), and similar knowledge of held-out data, one could use these results to predict how well these particular parameters would work on a new dataset that a different lab might generate. Note that this is the same problem as the very popular benchmarking problems in machine vision that are very popular, which doesn't stop those papers from getting published in top journals.

The supervised learning tools that we use in CaImAn pertain to the two CNN classifiers. From these the batch classifier was trained only on the first three datasets, whereas the online classifier was trained on the first five datasets, thus leaving a significant amount of held out data for testing. This is stated in subsection “Classification through CNNs”). For the rest of the pipeline CaImAn mostly uses tools from unsupervised learning (e.g., matrix factorization, tests of correlation coefficients and SNR levels etc). As such it is not exactly trained on a set of data to define a set of parameters which could be subject to overfitting. Thus, we believe that the parameters we pick here for the diverse set of nine datasets offer a sufficient amount of generalization to other datasets.

6) In the end, it remains unclear precisely what you recommend to do and when. Specifically, the manuscript mentions many functions/options/setting/parameters and several pre-processing stages. I also understand that the jupyter notebooks provide some examples. I think some concrete guidance, given that you've now analyzed nine different datasets spanning labs, sensors, etc., you have a wealth of knowledge about the heterogeneity of these datasets that would be valuable for the rest of the community, but is not quite conveyed in the manuscript.

This is a helpful suggestion. We have created an entry on the wiki of our GitHub repo with several tips on using CaImAn. https://github.com/flatironinstitute/CaImAn/wiki/CaImAn-Tips

We plan to continuously update this entry as our code and experience evolves, which explains our decision to not include this discussion in the main part of the paper. We link to this page from the paper when talking about our software. Please also see our response to reviewer #1, point 4 on parameter choice strategies for CaimAn online.

Reviewer #3:Giovannucci et al., present a software package to process calcium imaging data. As calcium imaging is becoming a dominant methodology to record neural activity, a standardized and open-source analytical platform is of importance. CaImAn presents one such effort. The manuscript is well written, and the software seems state-of-the-art and properly validated. I believe eLife Tools and Resources is a perfect outlet for such a commendable effort and enthusiastically support the publication of the study. Below are some comments that I believe the authors can address relatively easily prior to publication.Subsection “Batch processing of large scale datasets on standalone machines” (parallelization). First, it should be mentioned that the difficulty of parallelization arises from this particular motion correction algorithm in which correction of a frame depends on the previous frames. If each frame is processed independently, like in some other motion correction methods, it would be trivial to parallelize. Second, is there any mechanism to ensure that there is no jump between the temporal chunks? In other words, is the beginning of chunk X adjusted to match the end of chunk X-1? This seems critical for subsequent analysis.

Jumps between consecutive chunks are avoided by ensuring that eventually all the chunks are registered with the same template. At the beginning each chunk gets its own template but then the templates of all the chunks get a template of their own (a template of templates so to speak) which is used to register each individual frame. We’ve made this more clear in the document (subsection “Batch processing of large scale datasets on standalone machines”):

“[…] Naive implementations of motion correction algorithms need to either load in memory the full dataset or are constrained to process one frame at a time, therefore preventing parallelization. Motion correction is parallelized in CaImAn batch without significant memory overhead by processing temporal chunks of movie data on different CPUs. First, each chunk is registered with its own template and a new template is formed by the registered data of each chunk. CaImAn batch then broadcasts to each CPU ameta-template, obtained as the median between all templates, which is used to align all the frames in each chunk. Each process writes in parallel to the target file containing motion-corrected data, which is stored as a memory mapped array.”

Subsection “CAIMAN BATCH and CAIMAN ONLINE detect neurons with near-human accuracy” (ground truth).The authors mention that the human performance is overestimated because the ground truth is based on the agreement among the scorers. This effect can be eliminated, and the true human performance can be estimated by cross-validation. For example, in a dataset annotated by four annotators, one can test the performance of one annotator against a putative ground truth defined by the other three annotators. This can be repeated for all the four annotators and the average will give a better estimate of human performance than the current method.

Thanks for this suggestion. We followed this approach by comparing the results of each annotator to the combined results of the other annotators. We report these results in Table 3, (see: subsection “Cross-Validation analysis of manual annotations”). Perhaps surprisingly, the overall results were only modestly decreased, although the scores of individual annotators varied more resulting in more cases where CaImAn achieved a higher F1 score than individual annotators. While this comparison is unbiased (in the sense that it does not favor the manual annotations) it resulted in a set of 3 (or 4) different “ground truth” labels for each dataset and we felt that a direct comparison with the results of CaImAn was not warranted. As such, we do not present this analysis in the main Results section.

“[…] As mentioned in the Results section, comparing each manual annotation with the consensus can create slightly biased results in favor of individual labelers since the consensus is chosen from the union of individual annotations. To correct for this, we performed a cross-validation analysis where the annotations of each labeler where compared against an automatically generated combinations of the rest of the labelers. To create the combined annotations we first used the RegisterMulti procedure to construct the union of each subset of N-1 labelers (where N is the total number of labelers for each dataset). When N=4 then the combined annotation consisted of the components that were selected by at least two labelers. When N=3 a stricter intersection approach was used, i.e., the combined annotation consisted of the components that were selected by both remaining labelers. The procedure was repeated for all subsets of labelers and all datasets. The results are shown in Table 3. While individual scores for specific annotators and datasets vary significantly compared to using the consensus annotation as ground truth (Table 1), the decrease in average performance was modest indicating a low bias level.”

Subsection “Computational performance of CAIMAN”. The details of the systems are not described in the Materials and methods section. For a Macbook Pro, it has 8 logical CPU cores and 4 physical CPU cores (in 1 CPU). Please specify the CPU model names.

We have included this information in the revision:

“[…] each dataset was analyzed using three different computing architectures: (i) a single laptop (MacBook Pro) with 8 CPUs (Intel Core i7) and 16GB of RAM (blue in Figure 8a), (ii) a linux-based workstation (CentOS) with 24 CPUs (Intel Xeon CPU E5-263 v3 at 3.40GHz) and 128GB of RAM (magenta), and (iii) a linux-based HPC cluster (CentOS) where 112 CPUs (Intel Xeon Gold 6148 at 2.40GHz, 4 nodes, 28 CPUs each) were allocated for the processing task (yellow).”

Subsection “Computational performance of CAIMAN” (employing more processing power results in faster processing). From the figure, a 112 CPU cluster processes the data about 3 times as fast as a 8 CPU laptop. The performance gain does not seem to be proportional to the number of CPUs available even considering the overhead. Please discuss the overhead of parallelization and the bottleneck when processed in the cluster.

This is a valid observation. This phenomenon is mainly due to the fact that the data was processed on two different machines. The 112 CPU cluster was reading the data over network drives, a process that was both slower and somewhat variable. To examine the parallelization properties (see also our response to Reviewer #2, point #2) we more rigorously tested the processing speed when using the *same* machine but different number of cores:

“[…] To the study the effects of parallelization we ran CaImAn batch on the same computing architecture (24CPUs) utilizing a different number of CPUs at a time (Figure 8c). In all cases significant speedup factors can be gained by utilizing parallel processing, with the gains being similar for all stages of processing (patch processing, refinement, and quality testing, data not shown). The results show better scaling for medium sized datasets (J123, ~50GB). For the largest datasets (J115, ~100GB), the speedup gains saturate due to limited RAM, whereas for small datasets (~5GB) the speedup factor can be limited by the increased fraction of communications cost overhead (an indication of weak scaling in the language of high performance computing).”

Subsection “Computational performance of CAIMAN” (the performance scales linearly). Please clarify whether this (linearity) is a visual observation of the results, or is this based on a complexity analysis of the factorization algorithm?

The linear scaling with respect to frames is based on the complexity of the algorithm. A rank-K factorization of a MxN matrix has typically complexity O(MNK) and all the other analysis steps have linear scaling.

Subsection “Computational performance of CAIMAN” (enabling real-time processing). Please clarify how often such an update of a footprint is required in order to support real-time processing without parallelization.

(Please see also our response to point #9 from reviewer #1 who raised a similar question). We have modified our code so that the shape update is distributed among all the frames and happens only in frames where no new neurons are added to further distribute the cost evenly. We developed a simple algorithm that ensures that (i) every neuron is updated every N frames (where N is a user defined parameter, default value 200) and (ii) if a neuron is added, then the spatial footprints of all neighboring neurons are updated to adapt to the presence of their new neighbor. We describe this process more analytically in the Materials and methods section (see: distributed shape update). Our results indicate that updating all shapes at least once every 500 frames (in a distributed fashion) leads to similar results as in the original case where all the shapes were being updated at once at specific frames.

“[…] To efficiently distribute the cost of updating shapes across all frames we derived a simple algorithm that i) ensures that every spatial footprint gets updated at least once every T_u steps, where T_u is a user defined parameter, e.g., T_u=200, and ii) no spatial component gets updated during a step when new components were added. The latter property is used to compensate for the additional computational cost that comes with introducing new components. Moreover, whenever a new component gets added the algorithm collects the components with overlapping spatial footprints and makes sure they get updated at the next frame. This property ensures that the footprints of all required components adapt quickly whenever a new neighbor is introduced. The procedure is described in algorithmic form in Algorithm 6.”

Reviewing editor:Please consider adding a reference under "Related Work" to the use of supervised learning ("Adaboost") to segment active neurons, i.e. Automatic identification of fluorescently labeled brain cells for rapid functional imaging. I. Valmianski, A. Y. Shih, J. D. Driscoll, D. M. Matthews, Y. Freund and D. Kleinfeld, Journal of Neurophysiology (2010) 104:1803-1811.

Thank you for this suggestion, we have included this reference in our discussion of related methods.